# A lamprey neural cell type atlas illuminates the origins of the vertebrate brain

Francesco Lamanna [1,10] ✉, Francisca Hervas-Sotomayor [1,10] ✉,
A. Phillip Oel[1,2], David Jandzik [3,4], Daniel Sobrido-Cameán [5],
Gabriel N. Santos-Durán[5], Megan L. Martik [6,8], Jan Stundl [6],
Stephen A. Green[6], Thoomke Brüning[1], Katharina Mößinger[1], Julia Schmidt[1],
Celine Schneider[1], Mari Sepp [1], Florent Murat [1,9], Jeremiah J. Smith [7],
Marianne E. Bronner[6], María Celina Rodicio[5], Antón Barreiro-Iglesias [5],
Daniel M. Medeiros [3], Detlev Arendt [2] & Henrik Kaessmann [1]✉

The vertebrate brain emerged more than ~500 million years ago in common evolutionary ancestors. To systematically trace its cellular and molecular origins, we established a spatially resolved cell type atlas of the entire brain of the sea lamprey—a jawless species whose phylogenetic position affords the reconstruction of ancestral vertebrate traits—based on extensive single-cell RNA-seq and in situ sequencing data. Comparisons of this atlas to neural data from the mouse and other jawed vertebrates unveiled various shared features that enabled the reconstruction of cell types, tissue structures and gene expression programs of the ancestral vertebrate brain. However, our analyses also revealed key tissues and cell types that arose later in evolution. For example, the ancestral brain was probably devoid of cerebellar cell types and oligodendrocytes (myelinating cells); our data suggest that the latter emerged from astrocyte-like evolutionary precursors in the jawed vertebrate lineage. Altogether, our work illuminates the cellular and molecular architecture of the ancestral vertebrate brain and provides a foundation for exploring its diversification during evolution.

The vertebrate brain is a structurally complex and preeminent organ because of its central functions in the body. Its most fundamental divisions are the forebrain (prosencephalon, traditionally divided into the telencephalon and diencephalon), the midbrain (mesencephalon) and the hindbrain (rhombencephalon) (Fig. 1a). This regionalization is shared across all extant jawed vertebrates and is present even in jawless vertebrates (that is, the extant cyclostomes: lampreys and hagfishes),

the sister lineage of jawed vertebrates (gnathostomes)[1] (Fig. 1a), which have overall less complex brains than jawed vertebrates[2]. While a basic molecular regionalization has been described for the substantially simpler central nervous systems (CNSs) of the closest evolutionary relatives of vertebrates (urochordates and cephalochordates)[3–5], the anatomical complexity of the four major divisions of the vertebrate brain evolved in common vertebrate ancestors ~515–645 million years

[1]Center for Molecular Biology of Heidelberg University (ZMBH), DKFZ-ZMBH Alliance, Heidelberg, Germany. [2]Developmental Biology Unit, European Molecular Biology Laboratory, Heidelberg, Germany. [3]Department of Ecology and Evolutionary Biology, University of Colorado Boulder, Boulder, CO, USA. [4]Department of Zoology, Comenius University, Bratislava, Slovakia. [5]Department of Functional Biology, CIBUS, Faculty of Biology, Universidade de Santiago de Compostela, Santiago de Compostela, Spain. [6]Division of Biology and Biological Engineering, California Institute of Technology, Pasadena, CA, USA. [7]Department of Biology, University of Kentucky, Lexington, KY, USA. [8]Present address: Department of Molecular and Cell Biology, University of California Berkeley, Berkeley, CA, USA. [9]Present address: INRAE, LPGP, Rennes, France. [10]These authors contributed equally: Francesco Lamanna, Francisca Hervas-Sotomayor. ✉e-mail: f.lamanna@zmbh.uni-heidelberg.de; f.hervas@zmbh.uni-heidelberg.de; h.kaessmann@zmbh.uni-heidelberg.de

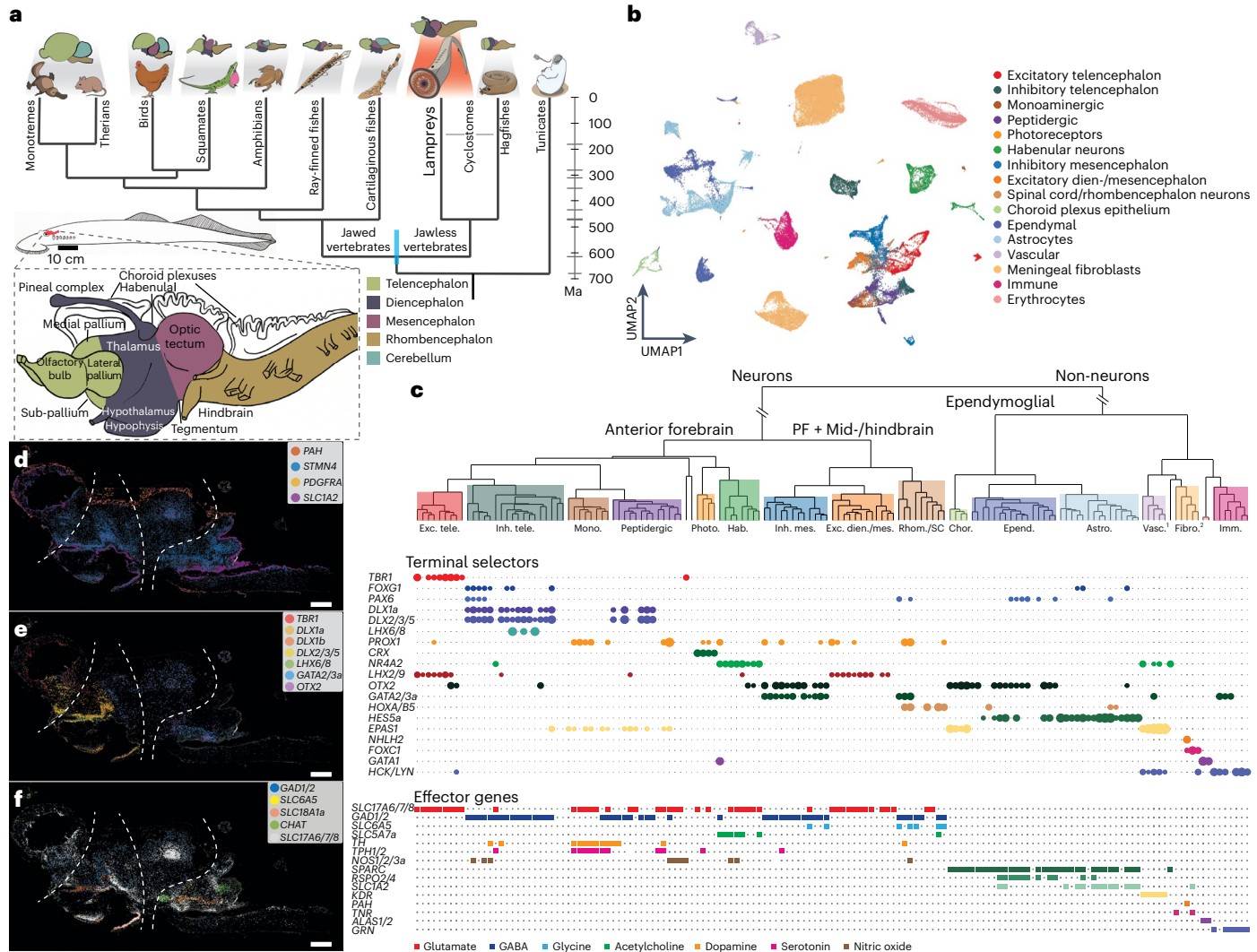

**Fig. 1 | Adult brain atlas overview. a**, Top: phylogenetic tree displaying the main vertebrate lineages and their approximate brain anatomies; the blue bar indicates the estimated confidence interval for the divergence time of cyclostomes and gnathostomes[6]. Bottom: schematic of the adult sea lamprey brain showing the different regions dissected for this study. **b**, UMAP of brain cells (all scRNA-seq data combined) coloured according to their corresponding cell type groups. **c**, Dendrogram describing the relationships between the identified cell types. The coloured boxes correspond to the highlighted cell type groups in **b**. Top: expression of terminal selector marker genes within each cell type; the circle sizes are proportional to the number of cells expressing the gene. Bottom: binary expression (presence/absence, based on whether a given gene is differentially expressed in the corresponding cell type; Methods) of effector

genes (neurotransmitters for neuronal types). PF, posterior forebrain; SC, spinal cord; 1, PNS glia; 2, erythrocytes. **d**–**f**, Sagittal sections (same orientation as in **a**) of the adult brain showing ISS maps of genes marking neurons (*STMN4*), ependymoglia (*SLC1A2*) and meningeal fibroblasts (*PAH* and *PDGFRA*) (**d**); anterior forebrain versus posterior forebrain and midbrain neuronal factors (**e**); and neurotransmitter genes (**f**). The dashed lines separate the main four brain regions illustrated in **a**. See Supplementary Fig. 2 for the ISS section schemes. Scale bars, 500 μm. The lamprey gene symbols throughout this study are based on the corresponding mouse orthologue names. When a lamprey gene corresponds to multiple mouse genes (one-to-many orthologous relationships), both gene names are indicated, using a slash (/) for separation.

ago (Ma)[6] (Fig. 1a), probably as part of the cephalic expansion that commenced around the emergence of this animal lineage (the 'new head' hypothesis)[7].

Previous anatomical and molecular studies of the vertebrate brain have yielded intriguing insights and hypotheses pertaining to its structural and functional evolution[8,9]. However, its ancestral cellular composition and underlying gene expression programs, as well as its subsequent diversification, have not been systematically explored.

To fill this critical gap, we generated a comprehensive cell type atlas of the adult and larval (ammocoete) brain of the sea lamprey (*Petromyzon marinus*), based on extensive transcriptomic and spatial expression data at single-cell resolution (https://lampreybrain.kaessmannlab.org/). Integrated comparative analyses of this atlas

unveiled details of the cell type repertoire and molecular architecture of the ancestral vertebrate brain but also revealed distinct cell types, gene expression programs and tissue structures that emerged during the evolution of the brain in jawed and jawless vertebrates.

## Cellular and molecular organization of the lamprey brain

We generated single-cell RNA-sequencing (scRNA-seq) data (21 libraries in total) for whole adult and ammocoete brains, as well as separately for their four major anatomical regions (telencephalon, diencephalon, mesencephalon and rhombencephalon), to facilitate cell type assignments (Fig. 1a and Supplementary Tables 1 and 2). To ensure optimal

scRNA-seq read mapping, we substantially refined and extended previous annotations of the lamprey germline genome[10] (Extended Data Fig. 1 and Supplementary Data 1) on the basis of 63 deeply sequenced RNA-seq libraries covering six major organs, including different brain regions (Supplementary Tables 1 and 2 and Methods). After quality control and data filtering (Methods), we obtained transcriptomes for a total of 159,381 high-quality cells (72,810 for adults and 86,571 for ammocoetes). Using a detailed clustering approach and an iterative marker-gene-based annotation procedure (Methods), we identified 151 (95 neuronal) distinct cell types in the adult dataset and 120 (92 neuronal) in the larval dataset (Supplementary Table 3; see the online atlas). To spatially localize cell types across the brain, we generated in situ sequencing (ISS)[11] data for 93 selected marker genes in both lamprey life stages and single-molecule RNA fluorescence in situ hybridization (smRNA-FISH) images for four genes in the larval stage (Supplementary Table 4 and Supplementary Data 2).

Overall, neural cell type compositions are similar between the two stages (Extended Data Figs. 2–4a). However, we noted a generally higher cell type specificity of gene expression patterns in adults than in ammocoetes (Extended Data Fig. 4c)—a result that is robust to controls for technical differences between datasets (Extended Data Fig. 4b–d and Methods).

A cell type tree derived from the datasets for the adult lamprey, which is thought to be better suited for the inference of ancestral vertebrate traits than ammocoetes[12], reflects cell type relationships based on gene expression distances (Fig. 1b,c). This tree unveils the hierarchical organization of cell types in the lamprey brain (Fig. 1c). The primary division is between neuronal and non-neuronal cell types, which are in turn split into ependymoglial cells (that is, neural-tube-derived glia) and other cells (that is, vascular cells, meningeal fibroblasts, blood cells and glial cells from the peripheral nervous system (PNS)). Our spatial ISS data illustrate that these three major cell type classes occupy very distinct areas of the brain (Fig. 1d).

At a secondary hierarchical level, non-neuronal cells are organized according to their cell class identity (for example, astrocytes, ependymal cells, erythrocytes and immune cells), in agreement with their molecular phenotype (Fig. 1c). By contrast, the organization of neuronal types primarily reflects their anatomical origin. A first separation is thus evident between telencephalic, anterior diencephalic (that is, hypothalamus and pre-thalamus), pineal and habenular neurons on one side of the neuronal clade, and posterior diencephalic (that is, thalamus and pre-tectum), mesencephalic and rhombencephalic/spinal cord neurons on the other. Within each developmental subdivision, neurons are organized according to their neurotransmitter phenotype (Fig. 1c).

The overall hierarchical cell type organization of the lamprey brain is supported by the expression patterns of terminal selectors (that is, sets of transcription factors (TFs) that determine and maintain cell type identity[13,14]) and effector genes (that is, sets of genes that characterize the molecular phenotype of cells) (Fig. 1c). Inhibitory neurons, for instance, are regulated mainly by *DLX* genes in the anterior forebrain[15] but by *GATA2/3*, *OTX2* and *TAL* genes in the posterior forebrain, midbrain, hindbrain and spinal cord[16,17] (Fig. 1c and Extended Data Fig. 5a; gene names are based on the respective names of the mouse orthologue(s)—see Methods for details regarding the gene nomenclature used in this study). Our ISS data confirm this strict compartmentalization of neuronal regulators (Fig. 1e and Extended Data Fig. 5h,i,l,m). Conversely, neurotransmitter-related genes are expressed across different brain regions (Fig. 1c,f).

The hierarchical relationship of cell types in the lamprey brain is very similar to that observed for a reference mammalian brain atlas (that is, that of the mouse[18]), which suggests that all vertebrates share a common general cellular and molecular organization of neural tissues that was established during the evolution of the vertebrate stem lineage.

## Vertebrate cell type families

To illuminate the cell type composition and molecular architecture of the ancestral vertebrate brain and to uncover differences between the CNSs of cyclostomes and gnathostomes, we performed detailed comparative analyses of our adult lamprey atlas with a corresponding atlas established for the mouse[18]. The neuronal and non-neuronal cells of the two atlases were contrasted separately using a dedicated method for homologous cell type detection (self-assembling manifold mapping (SAMap))[19] and a correlation-based analysis of gene expression that also considers paralogous genes and was adapted from a previous approach[20] (Methods).

The SAMap results show a great degree of correspondence between the two species for groups of cell types belonging to the same class (for example, vascular cells, astrocytes and excitatory neurons of the telencephalon), as indicated by the uniform manifold approximation and projection (UMAP) of the inter-species manifold (Fig. 2a,b) and the distribution of mapping scores between the two atlases (Fig. 2e,f, Extended Data Fig. 6 and Supplementary Table 5). This high-level similarity is confirmed by cross-species dendrograms based on the correlation approach applied to orthologous TF genes (Fig. 2c,d and Extended Data Fig. 7). These observations suggest that many of the corresponding cell classes share evolutionarily related gene expression programs (Supplementary Fig. 1). We propose that the matching groups of cell types uncovered in these analyses might constitute homologous cell type 'families' (ref. 21) that were already present in the brain of the last common ancestor of jawless and jawed vertebrates more than ~515–645 Ma[6].

## Blood, vascular and PNS cells

The blood cells found in the lamprey brain can be classified into erythrocytes, characterized by the massive expression of haemoglobin and haeme-related genes (for example, *ALAS1/2*), and immune cells, which are mainly composed of microglia/macrophages and lymphocytes (Extended Data Fig. 8a). The microglia/macrophage cell types are highly correlated to mammalian perivascular macrophages and microglia (Fig. 2c,e and Supplementary Table 5) and express genes that are typically related to the non-specific immune response (for example, *GRN*, *CSF1R* and *HCK/LYN*; Extended Data Fig. 8a–c) both outside (macrophages) and inside (microglia) the brain (Extended Data Fig. 8d). We also identified a lymphocytic cell population (type: Lympho2) expressing one of the two known cyclostome-specific variable lymphocyte receptor genes (*VLRA*; Extended Data Fig. 8a), which is part of a distinct adaptive immune system that emerged in the cyclostome lineage in parallel to that of gnathostomes[22].

We identified several vascular cell types, corresponding to endothelial cells/pericytes, which express typical vascular markers (for example, *EPAS1* and *KDR*; Extended Data Fig. 8a–c) and are principally localized at the innermost meningeal layer (Extended Data Fig. 8d), forming the perineural vascular plexus[23]. The inner and outer leptomeningeal layers are populated by fibroblast-like cells (type: Fibro1) that are probably homologous to the meningeal vascular fibroblasts described in the mouse brain[18,24], given their high respective homology mapping scores (Fig. 2c,e and Supplementary Table 5) and the expression of key orthologous marker genes (for example, *PDGFRA*, *FOXC1* and *LUM*; Extended Data Fig. 8a–d). A second fibroblast type (Fibro2) occupies the space between the leptomeningeal boundaries (Extended Data Fig. 8d) and is characterized by the expression of genes involved in the metabolism of glucose (*G6PC*)[25], fatty acids (*FABP3*), cholesterol (*SOAT1/2*) and aromatic amino acids (*PAH*) (Extended Data Fig. 8a) This cell type might correspond to meningeal round cells, which form a metabolically active tissue typical of lamprey that is not present in the meninges of other vertebrates[23,25].

PNS glia are represented by a small cluster (*n* = 53) expressing the orthologues of the mouse TF genes *Sox10* and *Sox9* (denominated *SOXE2* and *SOXE3* in lamprey, respectively[26]); they co-localize with

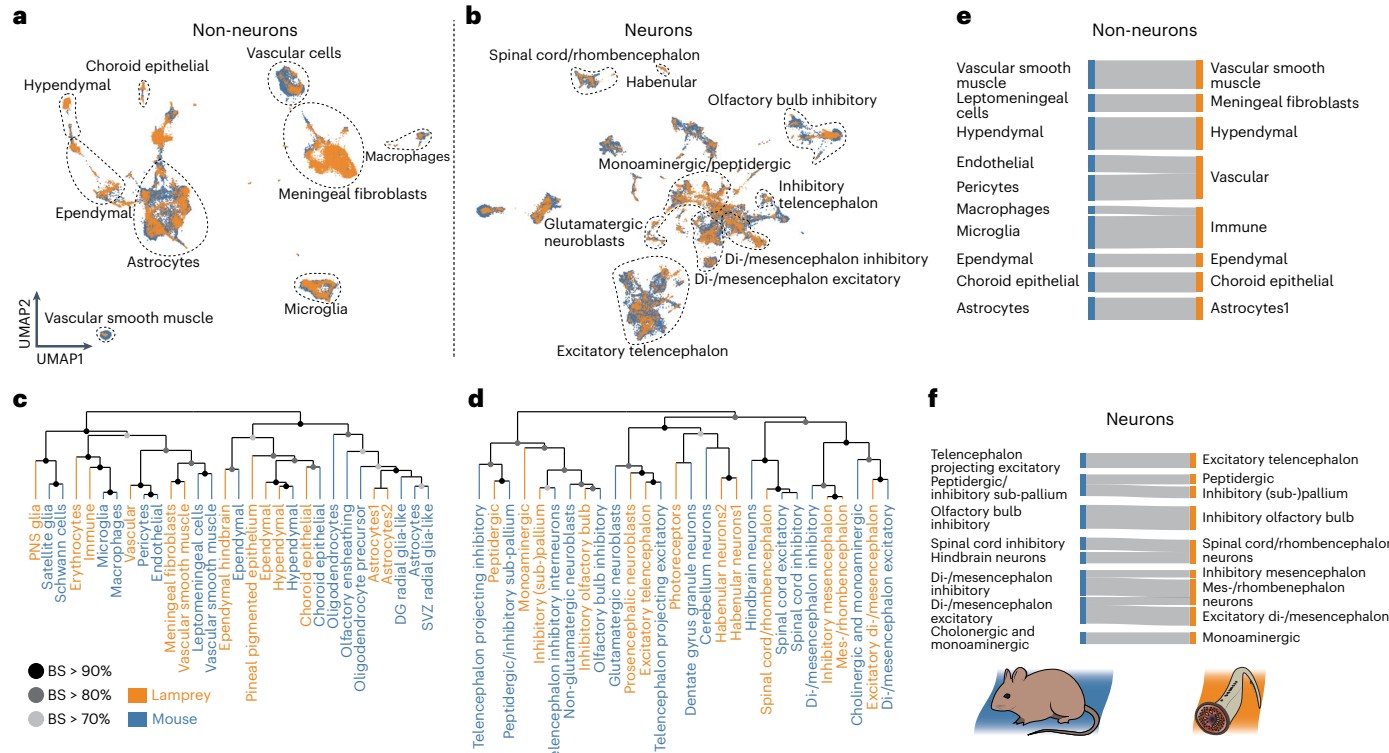

**Fig. 2 | Comparisons between lamprey and mouse brain atlases. a,b**, SAMap results displaying UMAPs of non-neuronal (**a**) and neuronal (**b**) cells from both species. Erythrocytes and oligodendrocytes were removed from the lamprey and mouse datasets, respectively. **c,d**, Dendrograms reporting gene expression distance (Pearson's *r*) of TF genes of non-neuronal (**c**) and neuronal (**d**) cell type groups from the two species. BS, bootstrap support (*n* = 1,000). **e,f**, Sankey diagrams relating non-neuronal (**e**) and neuronal (**f**) cell type groups between the two species based on SAMap mapping scores (min = 0.1; max = 0.65). The link width is proportional to the mapping score. DG, dentate gyrus; SVZ, sub-ventricular zone.

cranial nerve roots (Extended Data Fig. 8a,e). This group of cells, which most likely corresponds to the previously described peripheral ensheathing glia[27], expresses some markers whose mouse orthologues are characteristic of satellite glia (*SOXE2*) and Schwann cells and their precursors (*EGR2/3/4* and *PMP22/EMP3*) (Extended Data Fig. 8a). However, they lack the expression of key peripheral myelin constituent genes such as *MPZ* and *PMP2*, confirming the absence of actual myelin in the lamprey PNS[28]. Together with the co-clustering of this cell type with mouse satellite glia and Schwann cells (Fig. 2c), our observations strongly support and extend the hypothesis that lamprey PNS ensheathing glia are homologous to mammalian Schwann cells/ precursors. The co-localization of this cell type with meningeal fibroblasts and vascular smooth muscle cells in the cell type tree (Extended Data Fig. 8a) probably reflects their common developmental origin from the neural crest[18].

## Ependymoglial cells and the origin of myelination

Our analyses revealed that ependymoglial cells (that is, CNS glia) in lamprey are divided into two main, developmentally related, cell classes: ependymal-like cells and astrocyte-like cells, referred to as 'ependymal' and 'astrocytes' hereafter, given the observations described below. Ependymal cells are ciliated, epithelial-like cells that populate the ventricular system of the brain, the circumventricular organs[29] and the choroid plexuses[30] and are characterized by the expression of the ciliogenesis-related TF *FOXJ1* and the extracellular matrix component *CCN2/3/5* (Extended Data Fig. 9a,e,f,i). We identified two types of specialized secretory ependymal types in the lamprey brain: choroid plexus epithelial cells (*OTX2*+), responsible for the production of cerebrospinal fluid (CSF), and hypendymal cells of the sub-commissural organ (SCO), which massively express the main Reissner's fibres component

SCO-spondin (*SSPO*)[31] (Extended Data Fig. 9a,b,g,i). Two additional types of specialized ependymal cells are the pigmented pineal epithelial cells, defined by markers that are common to the retina pigment epithelium (for example, *RPE65* and *RRH*; Extended Data Fig. 9a), and the *KERA*-expressing ependymal cells of the hindbrain and spinal cord (types: ReEpen1 and ReEpen3; Extended Data Fig. 9a,c,d). The large number of detected ependymal cells and cell types in the adult dataset (Extended Data Fig. 3b) probably reflects the large relative sizes of the ventricles and choroid plexuses of the lamprey brain (Extended Data Fig. 9i)[32].

Notably, lamprey astrocytes are highly comparable to those from mouse in terms of their overall transcriptome signature (Fig. 2c,e). They share key marker genes that are fundamental for the development and function of astrocytes, such as *SOXE3* (*Sox9*), *HES5* and *SLC1A2* (Fig. 3a and Extended Data Fig. 9a). However, like in other anamniotes (for example, fishes and amphibians), lamprey astrocytes are mainly localized around the ventricles (Fig. 3 and Extended Data Fig. 9h), forming the so-called ependymo-radial glia[33].

Like in the PNS, lamprey CNS axons are not myelinated[28], consistent with the absence of key master regulators of oligodendrocyte identity (*OLIG1* and *OPALIN*) and myelin-specific genes (*MOBP* and *TSPAN2*) in its genome. Other myelin-related genes are present in the genome, but they are not expressed in glial cells (for example, *PDGFRA* and *NKX6-1/2* are expressed in meningeal fibroblasts; Extended Data Fig. 8a). Notably, despite the lack of myelination, lamprey astrocytes express several oligodendrocyte-specific genes, such as the TFs *NKX2-2* and *SOXE2* (*Sox10*)[34] (Fig. 3b and Extended Data Fig. 9a), the proteolipid gene *PLP1/GPM6B* (orthologous to the myelin components *Plp1* and *Gpm6b*) (Fig. 3b and Extended Data Fig. 9a), and the extracellular matrix glycoproteins *TNR* and *HEPACAM* (Extended Data Fig. 9a,j,k). Given the

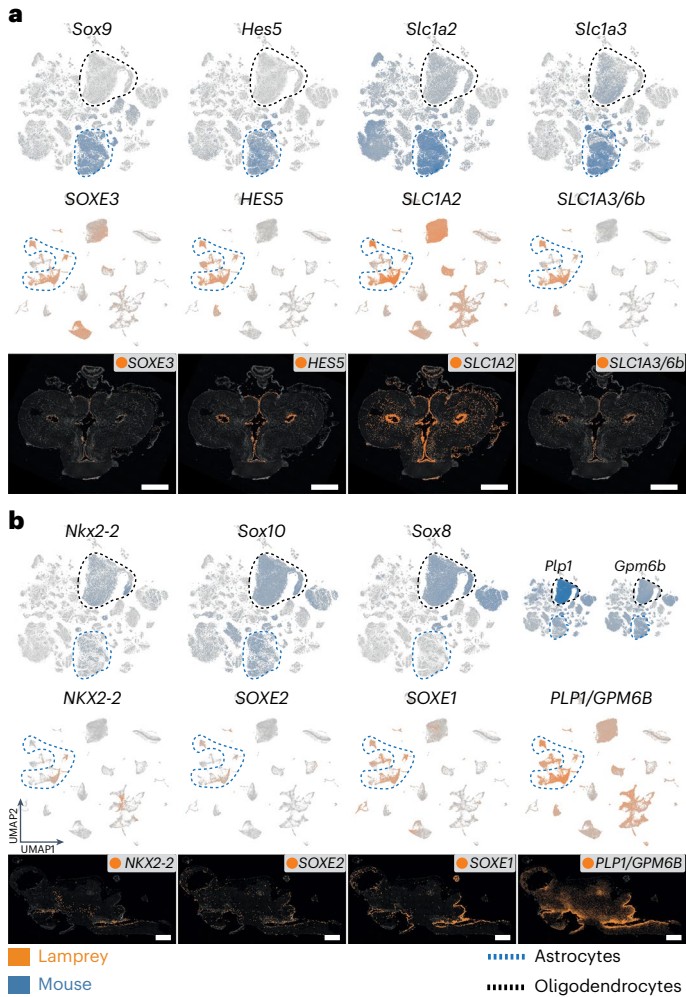

**Fig. 3 | Expression of astrocyte-specific and oligodendrocyte-specific genes. a,b,** UMAPs showing the expression of astrocyte-specific (**a**) and oligodendrocyte-specific (**b**) orthologous genes in the mouse (top) and lamprey (middle) atlases. Bottom: ISS maps of the adult lamprey brain for the same genes, showing coronal sections of the telencephalon (**a**) and sagittal sections of the whole brain (**b**; same orientation as in Fig. 1a). See Supplementary Fig. 2 for the ISS section schemes. Scale bars, 500 μm.

expression of crucial TFs of oligodendrocyte identity and the presence of myelin-related genes within lamprey astrocytes, our findings lend strong support to the hypothesis that oligodendrocytes originated from astrocyte-like glia in gnathostome ancestors[27].

## Neuronal diversity across brain regions

Finally, we scrutinized neuronal cell types across the different brain regions. Hindbrain and spinal cord neurons are defined by the expression of several *HOX* genes (*HOXA/B3*, *HOXA/B4* and *HOXA/B5*; Extended Data Fig. 5a,b). Two types of hindbrain glycinergic cells (ReInh5 and ReInh6), probably corresponding to inhibitory reticulospinal neurons[35], are highly correlated to reticular neurons of the medulla in mouse[18] (Supplementary Table 5) and express related markers (*SLC6A5*, *SLC32A1b* and *EBF2/3*; Extended Data Fig. 5a,b,f,g). Cholinergic neurons expressing the TF gene *TBX6/20* show very localized expression within the hindbrain, probably corresponding to afferent nuclei of cranial nerves[36] (Extended Data Fig. 5c–e). None of the detected midbrain/hindbrain clusters specifically express markers related to Purkinje (for example, *ALDOC*, *PCP2*, *SLC1A6* and *CAR8*) or granule (for example, *NEUROD1*, *CBLN1* and *GABRA6*) neurons of the cerebellum, nor are

these markers expressed in the dorsal isthmic region (Supplementary Data 2). We also did not detect the expression of marker genes in this region that are associated with neurons of inferred ancestral cerebellar nuclei[37], which were shown to have diversified in the gnathostome lineage through duplications[37]. These observations confirm the absence of proper cerebellar nuclei in the lamprey brain[37,38]. Within the rostral spinal cord, we identified two types of GABAergic CSF-contacting cells[39] (ReInh1 and ReInh2); these are ciliated neurons that are homologous to the gnathostome CSF-contacting neurons of the spinal cord central canal and express genes coding for channels that respond to changes in CSF pH (*PKD2L1* and *PKD2L2*) and for proteins that remove toxic oxidative compounds from the CSF (*AMBP*) (Extended Data Fig. 5a,b).

Thalamic, pre-tectal and tectal neurons are divided into excitatory and inhibitory classes (Extended Data Fig. 5a) and express TFs that are typical of homologous anatomical regions in mouse (that is, thalamus, pre-tectum and superior colliculus)[18]. In fact, like in the murine brain, glutamatergic neurons are characterized by the expression of *SHOX2*, *EBF1* and *EBF2/3*, whereas GABAergic neurons express *GATA2/3a*, *GATA2/3b*, *TAL1* and *OTX2* (Figs. 1e and 4a and Extended Data Fig. 5a,f–m).

Epithalamic neurons (that is, neurons stemming from the dorsal-most region of the diencephalon) are divided into habenular types and pineal/parapineal photoreceptors, like in the gnathostome brain[40]. All habenular neurons express the same TFs (*NR4A2*, *ETV1* and *IRX2/5*; Extended Data Fig. 10a,e), with the medial and lateral nuclei showing very distinct expression patterns for several genes (for example, *MYO9A*, *PRKCQ*, *GNG2* and *TMEM64*; Fig. 4b and Extended Data Fig. 10a,d). The medial habenula is occupied by glutamatergic, nitrergic and cholinergic neurons[41,42] (Fig. 1c and Extended Data Fig. 10a,d), with a cell type expressing neuropeptide Y (*NPY*; Extended Data Fig. 10a,d). The lateral habenulae are molecularly related to each other; they co-express several markers (*GNG2*, *TMEM64* and *SLC1A3/6a*; Fig. 4b and Extended Data Fig. 10a,d) and can be distinguished by the differential expression of two neuropeptide genes: proenkephalin (*PENK*; right) and cholecystokinin-like (*CCK*-like; left) (Extended Data Fig. 10f).

The pineal and parapineal of the lamprey are directly photosensitive organs with neuroendocrine outputs[43]. We detected signatures of both aspects of these organs at the molecular level by the expression of the genes *CRX* (necessary for the differentiation of photoreceptors), *GUCA1B* (involved in visual phototransduction), *LHX3/4* and *ISL1/2* (required for the development of retinal photoreceptors[44,45], as well as of neuroendocrine cells in the mammalian anterior pituitary[46]) in all detected cell types (Extended Data Fig. 10a,o,r). We could assign four clusters to these organs: predominantly in the pineal, we detected cone opsin-expressing (type: Photo1) and rod opsin-expressing (Photo2) cells defined by their expression of marker genes commonly associated with cones and rods, including arrestin, phosphodiesterase and *GRK*s. The pineal and parapineal organs differ in their expression of *RCVRN* and genes involved in the biosynthesis of melatonin (Extended Data Fig. 10a,n,p,q and the online atlas). More prominent in the parapineal organ, types Photo3 and Photo4 express the non-visual opsin gene parietopsin and the neuropeptide gene *TAC1* (Extended Data Fig. 10a). Unlike the pineal stalk, characterized by the expression of pineal markers, the parapineal ganglion and tract cells are marked by genes also detected in the habenulae (for example, *PPP1R14A/B/*C and *GNG2*; Extended Data Fig. 10k,n,o), consistent with reports that the secondary (downstream) neurons of the parapineal are segregated away from the photoreceptors of the parapineal vesicle[47].

Nearly all monoaminergic neurons, identified by the expression of monoamine transport (*SLC18A1a* and *SLC18A1b*) and metabolic (*TH* and *TPH1/2*) genes, form a unique taxon within the cell type tree (Fig. 1c and Extended Data Fig. 10a), which includes serotoninergic and dopaminergic neurons of the hindbrain, midbrain and hypothalamus. Dopaminergic neurons of the posterior tubercle nucleus of the

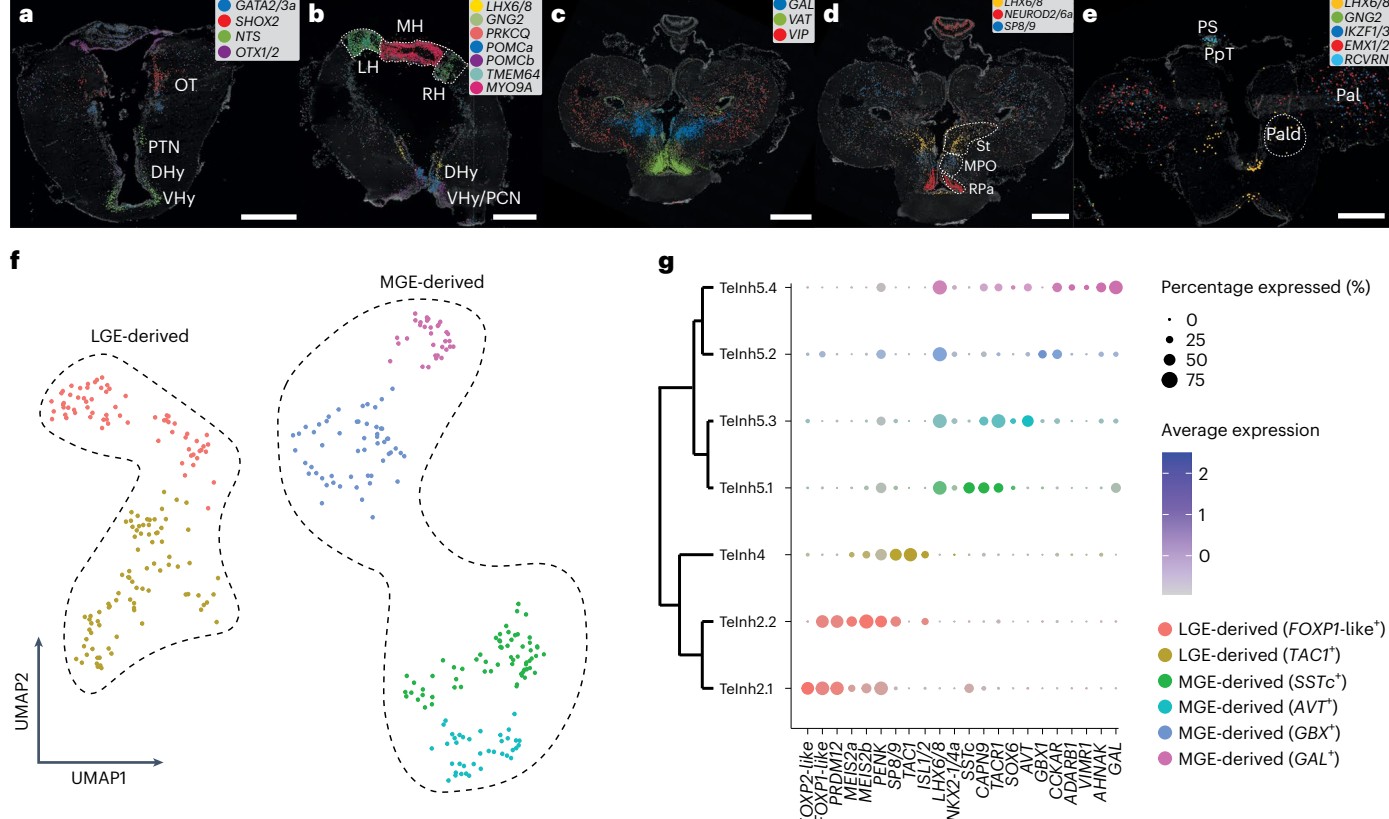

**Fig. 4 | Neuronal diversity. a–e**, ISS maps of selected neuronal marker genes across caudal and rostral coronal sections through the diencephalon (**a**,**b**) and telencephalon (**c**–**e**). **f**, UMAP showing LGE-derived and MGE-derived GABAergic neurons of the telencephalon. **g**, Dendrogram showing the relationships between the clusters in **f** and the expression of selected marker genes. Dhy, dorsal hypothalamus; LH, left habenula; Pal, pallium; Pald, pallidum; PpT, parapineal tract; PS, pineal stalk; MH, medial habenula; OT, optic tectum; PCN, postoptic commissure nucleus; PTN, posterior tubercle nucleus; RH, right habenula; RPa, rostral paraventricular area; St, striatum; VHy, ventral hypothalamus. See Supplementary Fig. 2 for ISS section schemes; scale bars, 500 µm.

hypothalamus (type: MeDopa1) co-express dopamine-related and glutamate-related genes[48] (Extended Data Fig. 10a,c) and are considered homologues of the dopaminergic neurons of the substantia nigra pars compacta of amniotes[49], an important component of basal ganglia. These cells are located next to *NTS*-producing neurons[50] (a modulator of dopaminergic activity[51]; Fig. 4a) and express the TF *PROX1* (Fig. 1c), which is crucial for the development of dopaminergic posterior tubercle nucleus cells in zebrafish[52].

Like in the mouse brain atlas[18], most hypothalamic peptidergic neurons co-cluster with monoaminergic cells (Fig. 1c and Extended Data Fig. 10a). Neurons of the ventral hypothalamus and postoptic commissure nucleus express the neuropeptide genes *CCK*[53] and pro-opiomelanocortins (*POMCa* and *POMCb*) (Fig. 4b and Extended Data Fig. 10e,h), as well as the circadian-rhythm-related genes *SIX3/6a* and *PER1/2* (also expressed in the pineal complex; see the online atlas). Other neuropeptides expressed in the hypothalamus are galanin (*GAL*)[54], somatostatins (*SSTa* and *SSTc*)[55], *NPY*[56], neurotensin (*NTS*), vasotocin (*VAT*), *PENK*, prepronociceptin (*PNOC*), gonadotropin-releasing hormones (*GNRH1* and *GNRH2*), prolactin-releasing hormone (*PRLH*) and *FAM237A/B* (Extended Data Fig. 10a,d–h). Additional peptidergic neurons cluster with inhibitory neurons of the pallium/sub-pallium (Extended Data Fig. 10a); these include *GAL*+ neurons of the septum and preoptic area (type: DePep10) and glutamatergic neurons expressing *VAT*, *GNRH1* and *NEUROD2/6a* (type: DePep9) located in the rostral paraventricular area of the preoptic area (Fig. 4c,d and Extended Data Fig. 10a,n).

Inhibitory neurons of the telencephalon are classified into olfactory bulb (OB) and pallium/sub-pallium cell types and are all enriched for typical forebrain GABAergic markers (*GAD1/2*, *DLX1a*, *DLX1b* and *DLX2/3/5*; Extended Data Figs. 5l,m and 10a). OB neurons can be recognized by (1) the conserved expression of several TFs that are characteristic of the anterior forebrain and placodes in chordates[3] (for example, *SP8/9*, *PAX6*, *FOXG1* and *ETV1*; Extended Data Fig. 10a,s), (2) the unique expression of *PRDM12* (expressed in pain-sensing nerve cells and V1 interneurons in gnathostomes[57,58]; Extended Data Fig. 10a,t), and (3) the presence of dopaminergic cells (type: TeDopa1; Extended Data Fig. 10a).

*SP8/9*+ neurons are also present in the sub-pallium (type: TeInh4), within a region traditionally considered to correspond to the medial preoptic nucleus (MPO)[59] (Fig. 4d), where they co-express *ISL1/2* and *TAC1*, both markers of striatal projection neurons in gnathostomes (Fig. 4g and Extended Data Fig. 10a). The presence of *SP8/9*+–*ISL1/2*+ and *SP8/9*+–*ETV1*+ neurons in the sub-pallium and OB, respectively, is already known for mammals, where they originate from the lateral ganglionic eminence (LGE)[60], suggesting that these two cell populations share the same developmental origin and migratory patterns across vertebrates.

Another important sub-pallial progenitor zone in jawed vertebrates is the medial ganglionic eminence (MGE). We identified neurons (type: TeInh5) expressing *LHX6/8* and *NKX2-1/4a* (both markers of MGE-derived cells in mammals) (Fig. 4g) within two sub-pallial regions: (1) dorsal to the MPO (Fig. 4d), in a region traditionally called 'striatum' (ref. 61), and (2) the putative pallidum[62], a nucleus located ventrolateral to the thalamic eminences (Fig. 4e). Recursive clustering revealed the presence of subtypes that express markers that are typical of MGE-derived neurons of the sub-pallial amygdala (SPA) and pallidum in jawed vertebrates[63] (for example, *TACR1*, *GBX1* and *SOX6*; Fig. 4g).

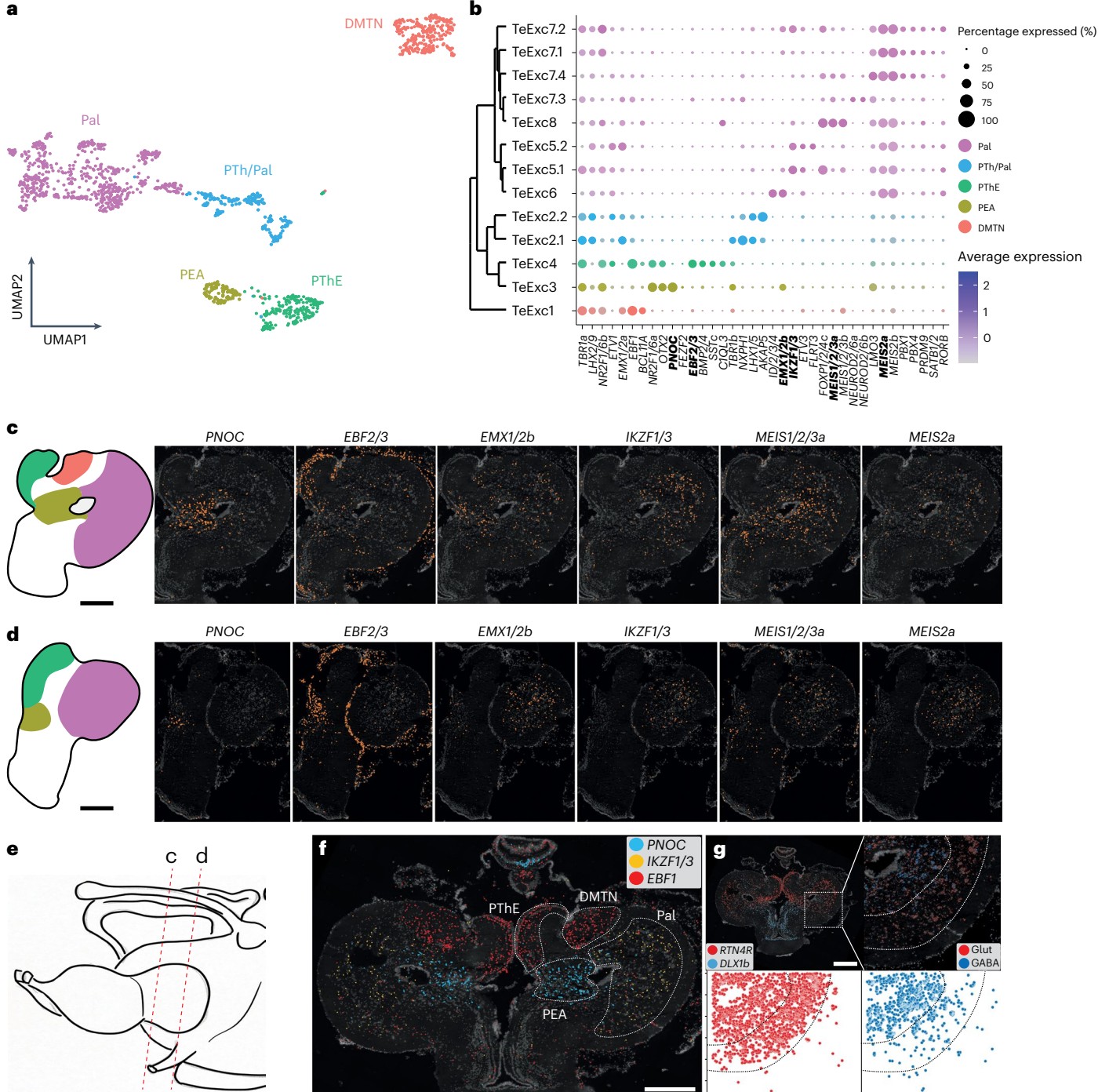

**Fig. 5 | Excitatory neurons of the telencephalon. a**, UMAP of excitatory neurons of the telencephalon highlighting the different regions. **b**, Dendrogram showing the relationships between the clusters in **a** and the expression of selected marker genes. **c**,**d**, Spatial localization of the clusters of **a** and **b** and the expression (ISS) of selected marker genes (highlighted in bold in **b**) in caudal (**c**) and rostral (**d**) coronal sections through the telencephalon. **e**, Section scheme of **c**, **d**, **f** and **g**. **f**, Spatial expression (ISS) of selected marker genes within the dorsal telencephalon. **g**, Top left: spatial expression of glutamatergic (*RTN4R*) and GABAergic (*DLX1b*) marker genes in the telencephalon. Top right: magnification from the dashed square showing the layered organization of the pallium; each neuronal class is highlighted by plotting the expression of multiple specific marker genes: GABA (*GAD1/2*, *DLX1a*, *DLX1b* and *DLX2/3/5*) and Glut (*SLC17A/6/7/8* and *TBR1*). Bottom: spatial scatter plots highlighting the positions of GABAergic (right) and glutamatergic (left) neurons within the pallium. PTh, pre-thalamus. Scale bars, 500 μm.

The presence of *DLX*[+] GABAergic neurons expressing LGE-related and MGE-related markers in the pallium (Extended Data Fig. 10j,o) implies that migration from progenitor zones of the sub-pallium also occurs in lamprey. Many of these neurons express the neuropeptide genes *PENK* and *SSTc*, which mark GABAergic interneuron types in the pallium of several gnathostome species[64–66] (Extended Data Fig. 10n,o,q,r). The vasoactive intestinal peptide (a marker of a sub-population of cortical GABAergic interneurons in amniotes) is also

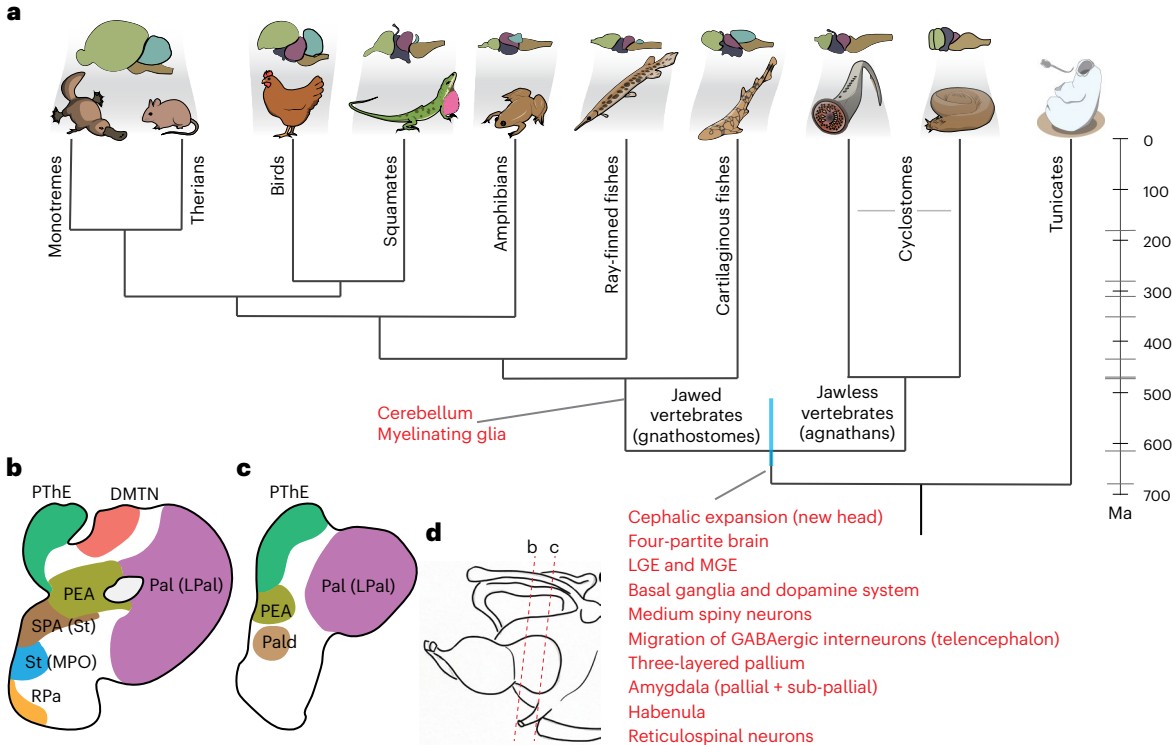

**Fig. 6 | Brain innovations and telencephalic region designation. a**, Vertebrate phylogenetic tree as in Fig. 1a showing key brain innovations (in red) as indicated, shown or confirmed by our study. **b,c**, Our interpretation of lamprey telencephalic regions; the names in parentheses indicate the original designation of the region before this study. The diagrams represent caudal (**b**) and rostral (**c**) coronal sections through the telencephalon. **d**, Scheme for sections in **b** and **c**. LPal, lateral pallium.

present in the lamprey pallium, but, contrary to gnathostomes, it is expressed exclusively in glutamatergic neurons (Fig. 4c and Extended Data Fig. 10a,p).

The expression programs of excitatory neurons of the lamprey telencephalon are overall highly correlated to those of the corresponding cell types in mouse (Fig. 2f). This similarity is confirmed by the expression of marker genes typical of mammalian cortical glutamatergic neurons within the lamprey pallium[67] (previously denoted 'lateral pallium' (ref. 59); see also the discussion below) and, partially, OB (for example, *TBR1*, *EMX1/2*a, *EMX1/2*b, *RTN4R*, *LHX2/9*, *BCL11B* and *IKZF1/3*; Fig. 5b–d and Extended Data Figs. 5h,i and 10a,g–i,k,l,q,v). We identified eight distinct cell types populating four different regions of the lamprey dorsal telencephalon and anterior diencephalon: (1) dorsomedial telencephalic nucleus (DMTN; type: TeExc1), (2) anterior pre-thalamic eminence (a region previously believed to correspond to the 'medial pallium' (ref. 2); see also the discussion below) (PThE; type: TeExc4), (3) pallial extended amygdala (PEA; type: TeExc3) and (4) pallium (types: TeExc2 and TeExc5–8) (Fig. 5a,b,f and Supplementary Table 3; see the online atlas). DMTN is a relay nucleus that is innervated by tufted-like cells of the OB[68] and is located at the interface between the pallium and OB, of which it constitutes the caudal-most portion. Like the OB, the DMTN displays a layered structure with outer glutamatergic neurons, which share the same expression profile with cells of the OB glomerular layer (for example, *EBF1*) and inner GABAergic (*PRDM12*+) neurons (Extended Data Fig. 10p,r–t,v,w). PThE and PEA neurons express the TFs *OTX2* and *NR2F1/6a* and are defined by the expression of *EBF1*, *SSTc* (TeExc4) and *C1QL3*, *PNOC* (TeExc3) (Fig. 5b and Extended Data Fig. 10a,i,o,s). We found that pallial neurons form a three-layered cortex with an inner GABAergic/glutamatergic layer, a middle glutamatergic layer and an external molecular, fibre-rich layer, in accord with previous work[69] (Fig. 5g). They all express multiple genes associated with cortical

projection neurons in amniotes (for example, *FOXP1/2/4*, *MEIS2*, *LAMP5*, *RORB* and *TCAP*; Extended Data Fig. 10a). However, contrary to what is known for amniotes and, since recently, also for amphibians (that is, for tetrapods in general)[70], we did not observe any regional specification of gene expression patterns among these neurons (for example, dorsal, lateral or ventral) that could be related to known, functionally distinct areas of the pallium (for example, somatosensory, visual, motor or olfactory), as previously observed on the basis of connectivity data[71,72].

## Discussion

In this study, we used extensive scRNA-seq and targeted spatial transcriptomics data to create a neural cell type atlas for a cyclostome representative: the sea lamprey (https://lampreybrain.kaessmannlab.org/). Our cell type tree analyses revealed that lampreys and gnathostomes share a common fundamental cellular and molecular organization of the brain that emerged in the vertebrate stem lineage more than ~515–645 Ma. This finding is in line with previous studies, such as the shared broad brain regionalization (Figs. 1a and 6a) and previously described patterning mechanisms across vertebrates[1]. Our comparisons of lamprey and mouse cell types revealed homologous relationships for many cell type families; that is, we identified groups of cell types partly sharing the same gene expression programs. These cell type families probably constituted the core of the ancestral vertebrate cell type repertoire.

Our analyses of non-neuronal cells revealed the presence of two distinct cell types within the lamprey ependymoglia that are probably homologous to ependymal cells and astrocytes of gnathostomes (Fig. 1c), suggesting that two of the three main macroglial cell types (astrocytes, ependyma and oligodendrocytes) were already established in the common vertebrate ancestor. Notably, however, our work confirms the absence of oligodendrocytes and sheds new light

on their origination. We found that lamprey astrocytes express several oligodendrocyte-specific genes, including master regulators and effector genes (Fig. 3b). Our observations suggest that key components of the molecular machinery of oligodendrocytes were present in astrocyte-like cells of the vertebrate ancestor, and indicate that oligodendrocytes originated from these evolutionary precursors in the gnathostome lineage (Fig. 6a). Our work thus extends previous studies, which showed that lamprey axons seem to be physically associated with astrocytes[27] and that key aspects of the regulatory program required for oligodendrocyte differentiation in gnathostomes are present during lamprey gliogenesis[34]. While our data also confirm the absence of actual myelin in the lamprey PNS[3], our study lends strong support to the previous hypothesis[4] that lamprey PNS glia are homologous to both mammalian Schwann and satellite cells.

Furthermore, our analyses did not provide evidence for the presence of granule or Purkinje cells in the rostral hindbrain, strongly supporting the notion that the mature lamprey brain lacks a proper cerebellum. We note, however, that a recent study detected the expression of granule and Purkinje cell TFs in the dorsal rhombomere 1 of lamprey embryos[73]. A targeted prospective analysis of the dorsal isthmic region in the adult and developmental lamprey brain might thus reveal the presence of potential rare homologues of cerebellar cell types.

The discovery of both LGE-derived and MGE-derived inhibitory neurons in the lamprey telencephalon confirms that the two main GABAergic progenitor zones of the sub-pallium were already present in the common vertebrate ancestor[8,70] (Figs. 4f and 6a). Our findings challenge the traditional neuroanatomy view regarding the localization of the main sub-pallial regions of lampreys. Previous studies[2,59] used to locate the striatum dorsal to the MPO and ventrolateral to the pallium (Fig. 4d). In this study, however, we identified a group of LGE-derived neurons (type: TeInh4), located in the MPO, that express the genes *ISL1/2*, *TAC1* and *PENK* (Fig. 4g), whose orthologues are typical markers of projection neurons of the dorsal striatum in jawed vertebrates (medium spiny neurons). This evidence indicates that the MPO of lampreys is in fact homologous to the dorsal striatum of jawed vertebrates and that it should be renamed accordingly (Fig. 6b). The region traditionally considered to correspond to the striatum, in contrast, is populated by MGE-derived cells (type: TeInh5) that express the markers *LHX6/8*, *GAL*, *GBX1* and *SOX6* (Fig. 4c,d,g), whose orthologues are expressed in the same combination in the SPA of jawed vertebrates. We therefore propose that this region corresponds to the SPA and not to the striatum, as previously believed[2,59] (Fig. 6b). MGE-derived cells can also be found caudal to the SPA, where they form the pallidum (Figs. 4e and 6c). Outside the lamprey sub-pallium, LGE-derived and MGE-derived cells also contribute to GABAergic interneurons of the OB and pallium, indicating that their migratory patterns are conserved across vertebrates (Fig. 6a).

Our analysis of the dorsal telencephalon confirms the hypothesis that the region previously denoted 'medial pallium' in lamprey is actually a rostral enlargement of the PThE[74] (that is, it is part of the diencephalon). This notion is supported by the expression of genes that are typically expressed in pre-thalamic excitatory neurons in both lampreys and jawed vertebrates (*EBF1* and *EBF2/3*) in the corresponding lamprey cell type (TeExc4), located in the previously denoted medial pallium, and by the absence of expression of marker genes that are typically associated with excitatory neurons of the pallium (*FOXG1* and *EMX1/2*) (Fig. 5b–d,f). Our data also indicate the presence of a PEA in lamprey, in support of a previous hypothesis[74], and show that it is located dorsal to the SPA. This region is populated by cells (type: TeExc3) that express markers of the extended amygdala in mouse[18,75] (*LMO3* and *PNOC*) (Figs. 5b–d,f and 6b).

Within the region previously denoted as 'lateral pallium' (ref. 2,59), we identified groups of cell types that are probably homologous to glutamatergic mammalian cortical neurons, supporting the hypothesis that the core cell types composing cortical/nuclear circuits

across jawed vertebrates emerged in common vertebrate ancestors[72,76]. These neurons express genes that are associated with different projection modalities (for example, input, intratelencephalic or output) (Fig. 5a,b), but not in the same combinations as observed in jawed vertebrates[77]. Altogether, our observations indicate that only the evaginated (that is, lateral) portion of the lamprey telencephalon should be considered a bona fide pallium (Fig. 6b,c), which—in terms of cell type expression signatures—is homologous to all subdivisions (dorsal, ventral, lateral and medial) of the pallium of tetrapods. This suggests that the regional specification of gene expression patterns among pallial neurons evolved during gnathostome evolution in the lineage leading to tetrapods[70,78]. Future work may illuminate the timing and mechanisms underlying this regionalization.

Altogether, our study provides a global view of the cellular composition and molecular architecture of the ancestral vertebrate brain and provides the groundwork for investigating its extensive cellular and structural diversification during vertebrate evolution.

## Methods

### Sea lamprey samples

Sea lamprey (*Petromyzon marinus*) samples were dissected from specimens obtained from three different sources (Supplementary Table 1). The sampled animals were euthanized by submersion in 0.1% MS-222 (Sigma, A5040-25G), unless specified otherwise, followed by decapitation according to local guidelines. Tissue samples from larvae (that is, ammocoetes, between 90 and 130 mm in body length), juveniles (Youson stages 6–7) and adults used for bulk tissue RNA-seq and genome annotation (see below) were collected from freshwater streams in Maine, USA, and held in large, aerated tanks with sand and freshwater until being sacrificed. All procedures were approved by the University of Colorado, Boulder, Institutional Animal Care and Use Committee as described in protocol 2392. Larvae (between 70 and 120 mm in body length) used for scRNA-seq, smRNA-FISH and Cartana experiments were collected form the River Ulla in Galicia, Spain, and kept at the Interfaculty Biomedical Research Facility of Heidelberg University in freshwater aerated tanks with river sediment and appropriate temperature conditions (~15 °C) until used for tissue collection. All animal procedures were performed in accordance with European Union and German ethical guidelines on animal care and experimentation and were approved by the local animal welfare authorities (Regierungspräsidium Karlsruhe). Upstream migrating mature adults used for the scRNA-seq experiments were obtained from a commercial supplier (Novas Y Mar, Galicia, Spain) and were processed immediately upon their arrival at the laboratory. All procedures were approved by the Bioethics Committee of the University of Santiago de Compostela and the Xunta de Galicia Government and conformed to European Union and Spanish regulations for the care and handling of animals in research. Adult specimens used for the Cartana experiments were obtained from the US Fish and Wildlife Service and Department of the Interior and were euthanized by immersion in 0.25% MS-222, followed by decapitation. All procedures were approved by the California Institute of Technology Institutional Animal Care and Use Committee protocol 1436.

### RNA extraction and sequencing of bulk tissue samples

In total, 63 sea lamprey tissue samples from six organs (brain, heart, liver, kidney, ovary and testis) were dissected from larval, juvenile and adult specimens. Total RNA was extracted using different extraction protocols (Supplementary Table 1); RNA quality was inspected using the Fragment Analyzer (Advanced Analytical Technologies), and its concentration was determined using a NanoDrop (Thermo Fisher Scientific). Strand-specific RNA-seq libraries were generated using the Illumina TruSeq Stranded mRNA Library protocol. Each library was sequenced on Illumina HiSeq 2500 platforms (100 nucleotides, single-end) at the Lausanne Genomic Technologies Facility (https://www.unil.ch/gtf).

## Sea lamprey genome annotation

Bulk tissue RNA-seq reads were mapped to the sea lamprey germline genome[10] using GSNAP[79] (v.2018-03-01) with the option to find known and new splice junctions in individual reads activated (novelsplicing, 1). The resulting BAM files for each stage and tissue were merged before being used for transcriptome assembly with StringTie[80] (v.1.3.4d). Each resulting GTF file was filtered for putative assembly artefacts using GffRead[81] (v.0.9.9) by discarding single-exon transcripts and multi-exon mRNAs that have any intron with a non-canonical splice site consensus (that is, not GT-AG, GC-AG or AT-AC). Individual annotated transcriptomes were then merged together with the already available set of annotated protein-coding genes from the germline genome study[10] to obtain a non-redundant set of transcripts. Genome annotation was further refined using TransDecoder (v.5.3.0; https://github.com/TransDecoder/TransDecoder) to identify candidate coding regions within the transcript sequences; this process involves identifying the longest putative open reading frame within each transcript and then searching the corresponding peptides against SwissProt (https://uniprot.org) using BlastP[82] (v.2.5.0+) and Pfam (https://pfam.xfam.org) using HMMER[83] (v.3.2). Annotation quality was assessed by comparing the number of reads mapping to exonic, intronic and intergenic regions of the genome (Extended Data Fig. 1). Annotation completeness was also estimated using BUSCO[84] (v.3) by comparing the set of translated longest coding sequences from each transcript against a set of metazoan-conserved single-copy orthologues from OrthoDB[85] (Supplementary Table 6).

## Orthology assignment and gene nomenclature

Homology information for the set of annotated genes was retrieved by applying the OrthoFinder[86] (v.2.3.11) pipeline against a group of selected chordates: vase tunicate (*Ciona intestinalis*)[87], inshore hagfish (*Eptatretus burgeri*; permission to use unpublished genome data was given exclusively for the purposes of the present study; personal communication), Australian ghostshark (*Callorhinchus milii*)[88], spotted gar (*Lepisosteus oculatus*)[89], zebrafish (*Danio rerio*)[90], West Indian Ocean coelacanth (*Latimeria chalumnae*)[91], western clawed frog (*Xenopus tropicalis*)[92], red junglefowl (*Gallus gallus*)[93], house mouse (*Mus musculus*) and human (*Homo sapiens*). By reconstructing a complete set of rooted gene trees among the analysed species, this tool allows us to establish all orthology relationships among all genes and to infer duplication events and cross-reference them to the corresponding nodes on the gene and species trees. Proteomes were downloaded from Ensembl[94] (remaining species; v.97) databases and used for a BlastP Best Reciprocal Hit analysis; to avoid redundancies in the blast results, only the peptides coming from the longest isoform within each gene were used. Rooted gene trees from the inferred orthogroups—that is, groups of genes descended from a single gene in the last common ancestor—were obtained using multiple sequence alignments (MAFFT[95] v.7.455) with IQ-TREE[96] (v.1.6.12; 1,000 bootstrap replicates) and STRIDE[97]. Orthology relationships can be explored in our online atlas. Throughout this work, we use mouse orthologue names to indicate lamprey gene names. This choice is justified by the fact that most lamprey genes lack a clear and consistent nomenclature and the fact that mouse is used as the main reference in our study. In cases where multiple mouse genes correspond to one lamprey gene (one-to-many relationships), we append all orthologue names, separated by slashes. We made an exception for *SOX* genes, where we used the well-established cyclostome annotation (*SOXA*, *SOXB*,…) and lamprey reference work.

## Cell dissociation and scRNA-seq data generation

Larval and adult heads were air-dissected, and the brains were placed in 1× HBSS (Life Technologies, 14185052) for cleaning and removal of the meninges. Once cleaned, the brains were further treated as a whole sample or, for the second set of experiments, divided into regions (telencephalon, diencephalon, mesencephalon and rhombencephalon). Brain tissue was dissociated using the Papain Dissociation System (Worthington, LK003150), according to the manufacturer's protocol, with the following modifications: the tissue was incubated in papain solution (volume adjusted for tissue size, 100–300 µl) at 28 °C for 15 min under constant agitation. Then, the tissue was gently triturated by pipetting up and down and collected by centrifugation for 1 min at 300 *g*. This step was followed by a second incubation in fresh papain solution and a final trituration, performed as described above. The dissociated cells were spun down at 300 *g* for 5 min and resuspended in the inhibitor solution (prepared following the Papain Dissociation System specifications). The suspension was filtered using a 40 µM falcon strainer (Sigma-Aldrich, CLS431750-50EA), and, immediately afterwards, a discontinuous density gradient was performed. The cells were then resuspended in Leibovitz's L-15 Medium (Life Technologies, 21083027), reaching a final volume between 50 and 100 µl, depending on the original tissue size. The cells were examined for viability and counted using a trypan blue staining and a Neubauer counting chamber (Assistent).

After ensuring a cell viability greater than 90% and a concentration equal or higher than 300 cells per µl, cell suspensions (-15,000 cells per reaction) were loaded onto the Chromium system (10x Genomics). Complementary DNA amplification and scRNA-seq libraries were constructed using Single-Cell 3′ Gel Bead and Library v.2 (for larvae) and v.3 kits (for adults and larvae), following the instructions of the manufacturer. For three larval whole brains, we additionally produced libraries using the v.3 kit (SN580, SN582 and SN588) for adequate technical comparisons between the larval and adult datasets (main text and Extended Data Fig. 4b–d). Complementary DNA libraries were amplified using 12 or 13 PCR cycles and quantified on a Qubit Fluorometer (Thermo Fisher Scientific). Average fragment size was determined on a Fragment Analyzer (Agilent). The libraries were sequenced using the NextSeq 500/550 High Output Kit v.2.5 on the Illumina NextSeq 550 system (28 cycles for Read 1, 56 cycles for Read 2, 8 cycles for i7 index and 0 cycles for i5 index).

## Single-cell RNA-seq data processing

The scRNA-seq reads were mapped to the reference genome[10] with our extended annotation (see above), and unique molecular identifier (UMI) count matrices were produced using CellRanger v.3.0.2 (10x Genomics). Cell-containing droplets were obtained from the CellRanger calling algorithm and validated by checking (1) the cumulative distribution of UMIs, (2) the distribution of UMIs coming from mitochondrial genes and (3) the distribution of the proportion of UMIs coming from intronic regions. Putative multiplets (that is, droplets containing more than one cell) were identified using DoubletFinder[98] and Scrublet[99]; droplets labelled as multiplets by either of the two methods were removed from the count matrices.

The obtained count matrices were analysed using Seurat v.3.1.5 (ref. 100) and pre-processed by keeping only genes expressed in at least five cells and by removing cells containing fewer than 200 UMIs and more than 5% (ammocoete) or 10% (adult) mitochondrial UMIs. The raw UMI counts were then normalized using the SCTransform method[101], and the top 3,000 highly variable genes (HVGs) across all cells were used for subsequent analyses. Principal component analysis (PCA) was applied to the normalized HVG matrices, and the resulting 75 most significant PCs were used for building a shared nearest-neighbour graph that was then clustered using the Louvain method with different resolution values (0.5–10). Differential expression analysis was run to find potential marker genes from all clusters across all resolution values (Wilcoxon rank sum test: logFC ≥ 0.25; min. pct = 0.1; Bonferroni-adjusted *P* < 0.01). The PCA-transformed matrices were finally embedded in two-dimensional space using UMAP and *t*-distributed stochastic neighbour embedding (*t*-SNE) dimensionality reduction techniques.

The clustered cells were further manually inspected to identify and then remove spurious clusters (that is, clusters composed by damaged/stressed cells or multiplets/empty droplets that escaped the previous filtering steps). Cell types/states were annotated on top of the clusters obtained using the highest resolution value (10); a putative phenotype/function was assigned to each cluster by allocating marker genes to any of the following Gene Ontology[102] categories: transcription (co-)factor, neurotransmitter metabolism, neurotransmitter transport, neurotransmitter receptor, neuropeptide[103], neuropeptide receptor[103], immune response, erythrocyte differentiation, blood vessel development, neurogenesis or gliogenesis. Annotated clusters that were contiguous on the UMAP and *t*-SNE embeddings were manually inspected and joined together if they were showing similar expression patterns among their respective marker genes. Additional functional information was added by comparing the annotated clusters to published vertebrate neural single-cell datasets[18,104].

Datasets coming from different samples were integrated using integrative non-negative matrix factorization as implemented in LIGER v.0.5.0 (ref. [105]). The datasets were integrated at two levels: (1) integration of replicates coming from the same brain region (that is, telencephalon, diencephalon, mesencephalon, rhombencephalon and whole brain) and stage (that is, ammocoete and adult), and (2) integration, within each stage, of all replicates together in the same dataset encompassing all sampled regions. Each integrated dataset was then imported to Seurat to perform shared nearest-neighbour graph construction, clustering, differential expression analysis, 2D embedding and cluster annotation as described above.

We noticed that the number of UMIs and expressed genes per cell was consistently lower for the larval dataset (produced using Chromium kit v.2) than for the adult one (produced using Chromium kit v.3) (Supplementary Tables 1 and 2). To establish whether this difference reflected an actual biological property of the two stages, we produced three larval datasets using the v.3 kit and compared their number of expressed genes per cell to the larval v.2 and adult v.3 datasets (Extended Data Fig. 4b) (see also 'Cell dissociation and scRNA-seq data generation'). We also compared the distributions of cell-type-specific gene expression signals across datasets based on gene specificity indices calculated using the method developed by Tosches and colleagues[64]. Briefly, to obtain the specificity index with this method, the mean of normalized scRNA-seq read counts of each gene ($g_c$) is calculated for each cell type ($C$) and then divided by its mean across all cells:

$$s_{g,c} = \frac{g_c}{\frac{1}{N}\sum_{i \in c} g_i}$$

### Lamprey–mouse comparisons

To find cross-vertebrate similarities and differences in neural cell types, the adult integrated brain atlas was compared against a published juvenile mouse nervous system atlas[18]. The two datasets were first compared via a correlation-based approach. That is, the raw UMI count matrices were extracted from both species datasets, and orthology information for the corresponding gene IDs was added; orthology relationships between mouse and lamprey were obtained from the OrthoFinder analysis (see above; Supplementary Table 7). The UMI counts coming from paralogues in the respective species were summed ('meta-gene' method[20]), and the species-specific gene IDs were replaced by numeric indices (1.*n*, where *n* is the number of orthology groups between the mouse and lamprey) shared by the two species. The new meta-gene count matrices were then normalized using SCTransform, filtered for HVGs and averaged across all annotated clusters. The expression levels were finally transformed to specificity indices (see above), which were then used for Pearson correlation

analyses. Dendrograms relating cell-type families between lamprey and mouse were constructed using the pvclust[106] R package with complete hierarchical clustering and 1,000 replicates.

In addition, the two datasets were compared using the SAMap (v.0.2.3) algorithm[19], a method that enables mapping single-cell transcriptomic atlases between phylogenetically distant species. A gene–gene bipartite graph with cross-species edges connecting homologous gene pairs was constructed by performing reciprocal BlastP searches between the two proteomes of the two species. The graph was used in a second step to project the two datasets into a joint, lower-dimensional manifold representation, where the expression correlation between homologous genes was iteratively used to update the homology graph connecting the two atlases. After the analysis was run, a mapping score (ranging from 0 to 1) was computed among all possible cross-species cluster pairs. The full list of all lamprey gene names used in this study with their respective gene IDs is reported in Supplementary Table 8.

### ISS

Whole brains (adults) and heads (larvae) were embedded in OCT mounting medium and then flash-frozen by laying them on isopentane, previously cooled on liquid nitrogen. Adult tissues were rinsed with ice-cold PBS before being frozen. The tissues were cryosectioned in 10 μm coronal and sagittal sections and stored at −80 °C until further use. Sections were processed for ISS using the High Sensitivity Library Preparation Kit from CARTANA AB (10x Genomics). The method and data processing are described by Ke and colleagues[11]. Processing of sections was done following CARTANA's protocol with minor modifications. In brief, sections on SuperFrost Plus glass slides (Thermo Fisher Scientific) were air-dried for 5 min. Afterwards, the sections were fixed by 3.7% (v/v) paraformaldehyde in UltraPure distilled water (DNase/RNase-Free, Thermo Fisher Scientific, 10977035) for 7 min and washed in PBS (Thermo Fisher Scientific, 70011036; diluted in UltraPure distilled water), followed by 0.1 N HCl treatment for 5 min and a wash with PBS. The sections were then dehydrated with ethanol and air-dried before being covered with SecureSeal hybridization chambers (Grace Bio-Labs, 10910000). All subsequent steps, including probe hybridization and ligation, amplification, fluorescent labelling and quality control imaging, followed the manufacturer's specifications. Finally, the mounted sections were shipped to CARTANA's facility (Solna, Sweden) for ISS.

### Single-molecule RNA-FISH

Larval whole heads were snap-frozen and cryosectioned (horizontal sections) as described above. This time, however, the sections were collected on coverslips (22 mm × 22 mm) previously pretreated with a silanization solution (0.3% (v/v) bind-silane (GE Healthcare Life Sciences, 17-1330-01), 0.1% (v/v) acetic acid and 99.6% (v/v) ethanol).

To reduce tissue autofluorescence, the sections were embedded in polyacrylamide (PA) gel, RNAs were anchored to the gel by LabelX treatment, and cellular proteins and lipids were cleared as previously described[107,108], with modifications. LabelX solution was prepared by reacting Label-IT (Mirus Bio) with Acryloyl X, SE (Thermo Fischer Scientific), as described by Chen and colleagues[108]. Specifically, sections were air-dried for 15–20 min and fixed in 3.7% paraformaldehyde in PBS for 10–15 min, followed by a 2 min incubation in 4% SDS in PBS and washes with PBS. The fixed sections were then incubated in 70% ethanol at 4 °C for at least 16 h. Next, sections on coverslips were washed twice with PBS and once with 1× MOPS pH 7.7 (Sigma-Aldrich, M9381) and incubated with LabelX (diluted to a concentration of 0.006 mg ml⁻¹ in 1× MOPS) at room temperature for 4 h, followed by two PBS washes. To anchor LabelX-modified RNAs, the sections were embedded in thin 4% PA gels. First, the coverslips were washed for 2 min with a PA solution, consisting of 4% (v/v) of 19:1 acrylamide/bis-acrylamide (Sigma-Aldrich, A9926-5), 60 mM Tris·HCl pH 8 and 0.3 M NaCl. The

coverslips were then washed for 2 min with the PA solution supplemented with ammonium persulfate (Sigma-Aldrich, 7727-54-0) and TEMED (Sigma-Aldrich, T7024) at final concentrations of 0.03% (w/v) and 0.15% (v/v), respectively. To cast the gel, 75 μl of the PA solution (supplemented with the polymerizing agents) was added to glass slides previously treated with Repel Silane (GE Healthcare Life Sciences, 17-1332-01) and washed with ethanol. Each coverslip was then layered on top of a slide, with one drop of PA solution, ensuring that a thin PA layer formed between the slide and the coverslip. The gel was allowed to cast at room temperature for 1.5 h. The coverslips and slides were gently separated, leaving coverslips with sections embedded into the PA gel. The coverslips were then washed with digestion buffer consisting of 0.8 M guanidine-HCl, 50 mM Tris·HCl pH 8, 1 mM EDTA and 0.5% (v/v) Triton X-100. The coverslips were incubated with digestion buffer supplemented with 8 U ml$^{-1}$ proteinase K (Sigma-Aldrich, P2308) at 37 °C for 2–3 h.

After background reduction, the sections were hybridized with HuluFISH probes, designed and developed by PixelBiotech. The hybridization protocol followed the manufacturer's recommendations. Briefly, the coverslips were washed twice with HuluWash buffer (PixelBiotech GmbH) and incubated in 50 μl of probe solution, consisting of each probe diluted in hybridization buffer at a concentration of 1:100. The coverslips were incubated at 37 °C for 12 h, inside a light-protected humidified chamber. Afterwards, the coverslips were washed four times with HuluWash buffer. Each wash lasted 10 min and was done at room temperature. The last wash was supplemented with Hoechst 33342 (Thermo Fisher Scientific, H3569). The coverslips were then mounted in two drops of Prolong Diamond mounting medium (Thermo Fisher Scientific, P36961). The mounted sections were allowed to cure at room temperature for 24 hours.

All sections were imaged on a Leica TCS-SP5, a confocal laser scanning microscope controlled by the Leica Application Suite. All images shown are the projections of mosaics built by stitching individual z-stacks. Each z-stack consisted of individual images (50 images for *SSPOa*, *VAT* and *GNRH1a*; 15 images for *ZFP704*) taken by setting a range of 10–15 μm and a step size below 0.8 μm. The images were captured with a ×63 immersion oil objective and sequentially excited by a 405 nm diode laser (for the Hoechst 33342 staining), followed by the laser required for each probe (561 nm DPSS laser for *SSPOa*, *ZFP704* and *VAT*; and 633 nm HeNe laser for *GNRH1a*). Projections of the z-stacks were performed in Fiji v.2 (ref. 109) by using the average intensity projection. Further processing (only when required) involved contrast enhancing (saturated pixels between 0.1% and 0.3%) and background subtraction for noise reduction (rolling ball with a radius of 50 pixels).

### Reporting summary

Further information on research design is available in the Nature Portfolio Reporting Summary linked to this article.

## Data availability

The raw and processed bulk and scRNA-seq data have been deposited to ArrayExpress with the accession numbers E-MTAB-11085 (bulk) and E-MTAB-11087 (single cell) (https://www.ebi.ac.uk/arrayexpress/). The genome annotation files and in situ images have been deposited to Zenodo[110] (https://doi.org/10.5281/zenodo.5903844). Information about gene expression, cell type annotation and gene orthology relationships across species can be visualized using the online atlas (https://lampreybrain.kaessmannlab.org/).

## Code availability

All code underlying the published atlas is available on GitHub (https://github.com/f-lamanna/LampreyBrainAtlas/) and Zenodo[111] (https://doi.org/10.5281/zenodo.8113793) together with detailed instructions about its usage.

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

## Acknowledgements

We thank all members of the Kaessmann group for fruitful discussions; Robert Frömel for his help with library sequencing; E. Panzariello, M. Sanchez-Delgado and N. Trost for the brain and animal illustrations; M. Mall for providing temporary lab space; and M. Cardoso-Moreira for discussions and comments on the manuscript. The computations were performed on the Kaessmann lab server (managed by N. Trost) and the bwForCluster from the Heidelberg University Computational Center (supported by the state of Baden-Württemberg through bwHPC and the German Research Foundation, INST 35/1134-1 FUGG). We thank W. Wang of the Northwestern Polytechnical University of Xi'an, China; Y. Zhang of the Institute of Zoology, Chinese Academy of Sciences; and J. Pascual-Anaya from the University of Málaga, Spain, for granting access to the early draft of the inshore hagfish genome (https://doi.org/10.1101/2023.04.08.536076). This work was supported by grants from the European Research Council (no. 615253, OntoTransEvol), the European Commission (Marie Skłodowska-Curie Actions ITN: EvoCELL) and the Tschira foundation, which funded the Illumina NextSeq machine used for sequencing. D.M.M. and D.J. were supported by grants from the National Science Foundation (IOS 1656843), the National Institutes of Health/National Institute of Dental and Craniofacial Research (RDE025940) and University of Colorado, Boulder RIO Innovative Seed Grant FY21 (all to D.M.M.). D.J. was also supported by a grant from the Scientific Grant Agency of the Slovak Republic (VEGA 1/0450/21). This project has received funding from the European Research Council under the European Union's Horizon 2020 research and innovation programme (VerteBrain to H.K., grant agreement no. 101019268; and NeuralCellTypeEvo to D.A., grant agreement no. 788921).

## Author contributions

F.L., F.H.-S and H.K. conceived and organized the study on the basis of H.K.'s original design. F.L., F.H.-S and H.K. wrote the manuscript with input from all authors. F.L. performed all analyses and developed the brain atlas app. F.H.-S. established and optimized the tissue dissociation protocol and performed all scRNA-seq and in situ experiments with support from A.P.O., J. Schmidt and C.S. and guidance from M.S. F.L. and F.H.-S. annotated and interpreted the data. T.B. prepared the bulk libraries with guidance from K.M. M.S. established the smRNA-FISH protocol. F.H.-S., A.P.O., D.J., D.S.-C., G.N.S.-D., M.L.M., J. Stundl and S.A.G. collected the samples. A.B.-I., D.M.M., M.E.B. and M.C.R. provided the samples. J.J.S. provided early access to genome assemblies and annotations. A.P.O., M.S., F.M., D.S.-C., A.B.-I., D.M.M. and D.A. provided useful feedback and discussions. H.K. supervised the study and provided funding.

## Competing interests

The authors declare no competing interests.

## Additional information

**Extended data** is available for this paper at https://doi.org/10.1038/s41559-023-02170-1.

**Correspondence and requests for materials** should be addressed to Francesco Lamanna, Francisca Hervas-Sotomayor or Henrik Kaessmann.

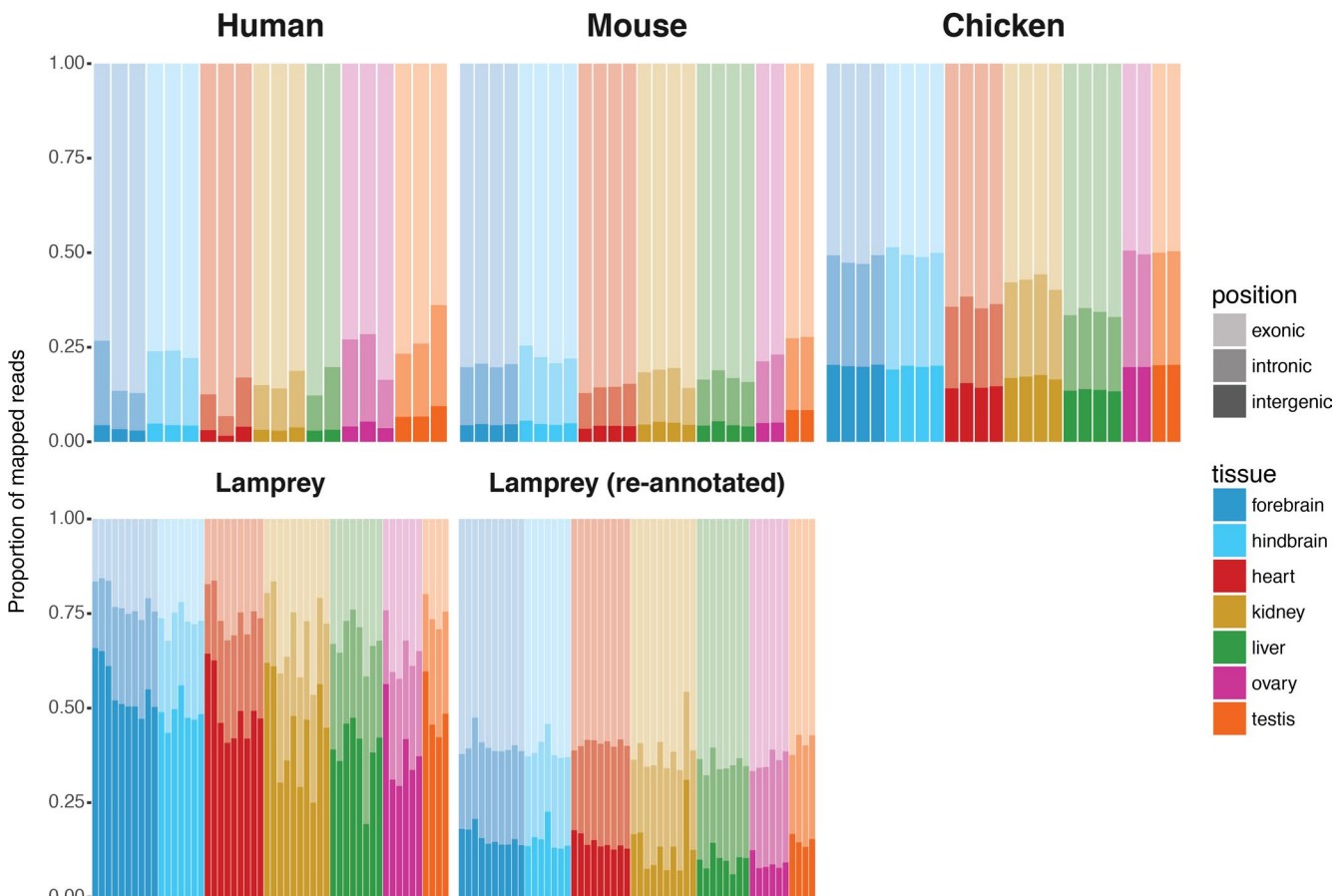

**Extended Data Fig. 1 | Assessment of lamprey genome annotation quality.** Barplots comparing the proportion of reads mapping to exonic, intronic, and intergenic regions between the lamprey genome annotation produced in this study (re-annotated) and the published annotations of lamprey[10], chicken, mouse, and human.

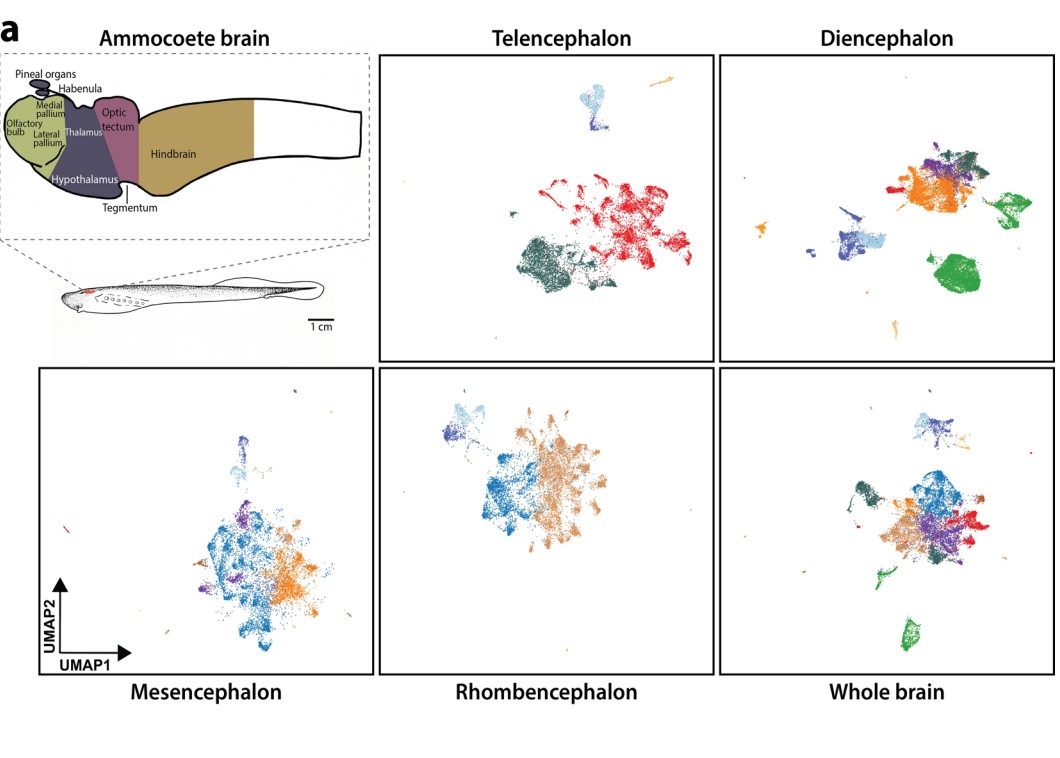

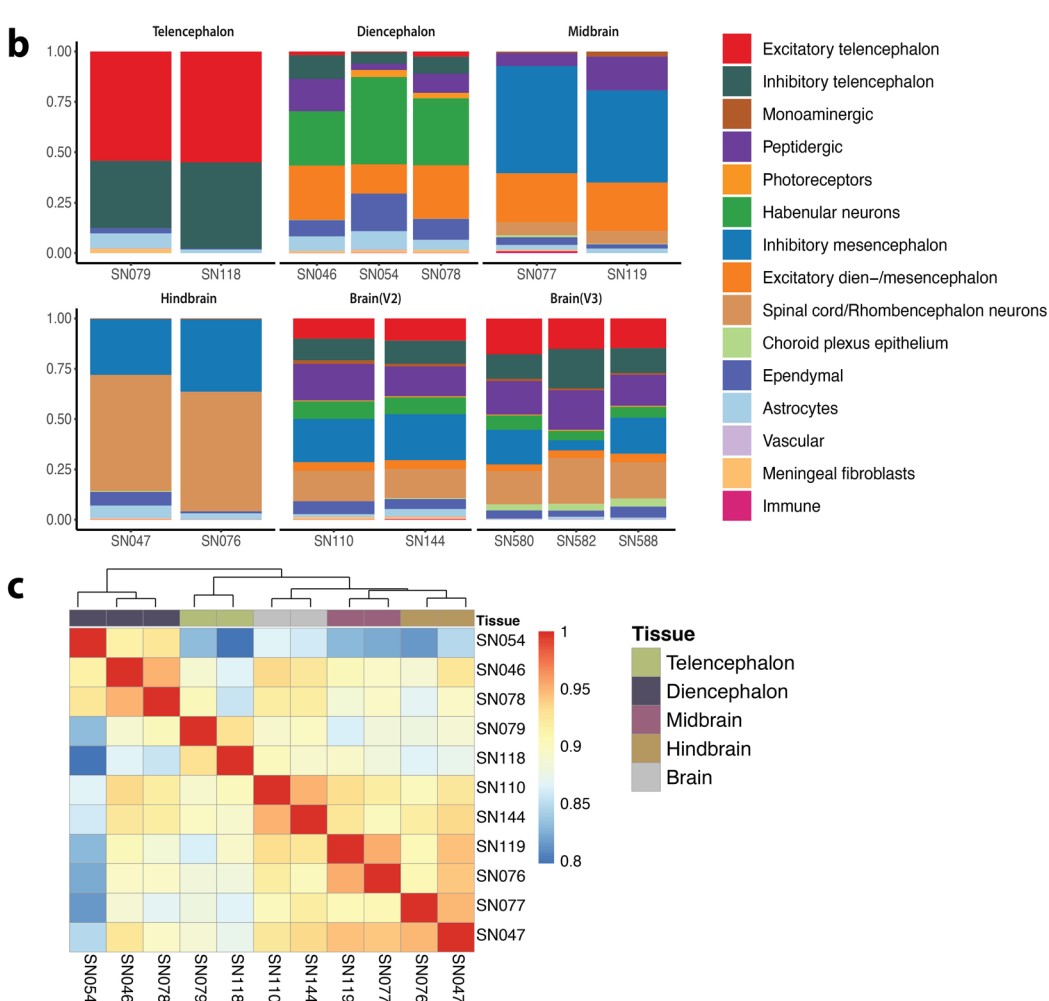

**Extended Data Fig. 2 | See next page for caption.**

**Extended Data Fig. 2 | Larval (ammocoete) brain dataset. a**, Schematic of the sea lamprey larval brain showing the different regions dissected for this study and UMAP projections for each brain region. Each UMAP projection represents an integration of the biological replicates (as indicated in panel b) for each brain region, respectively. Additional information is available in the interactive atlas (https://lampreybrain.kaessmannlab.org/ammocoete.html). **b**, Barplots showing the proportions of each cell type group (as reported in a and Fig. 1b, c) for each sample. For whole brain samples, proportions are showed separately for datasets obtained using the v2 and v3 kits. **c**, Heatmap showing clustered pseudobulk brain regions based on Spearman's ρ. Median ρ = 0.90.

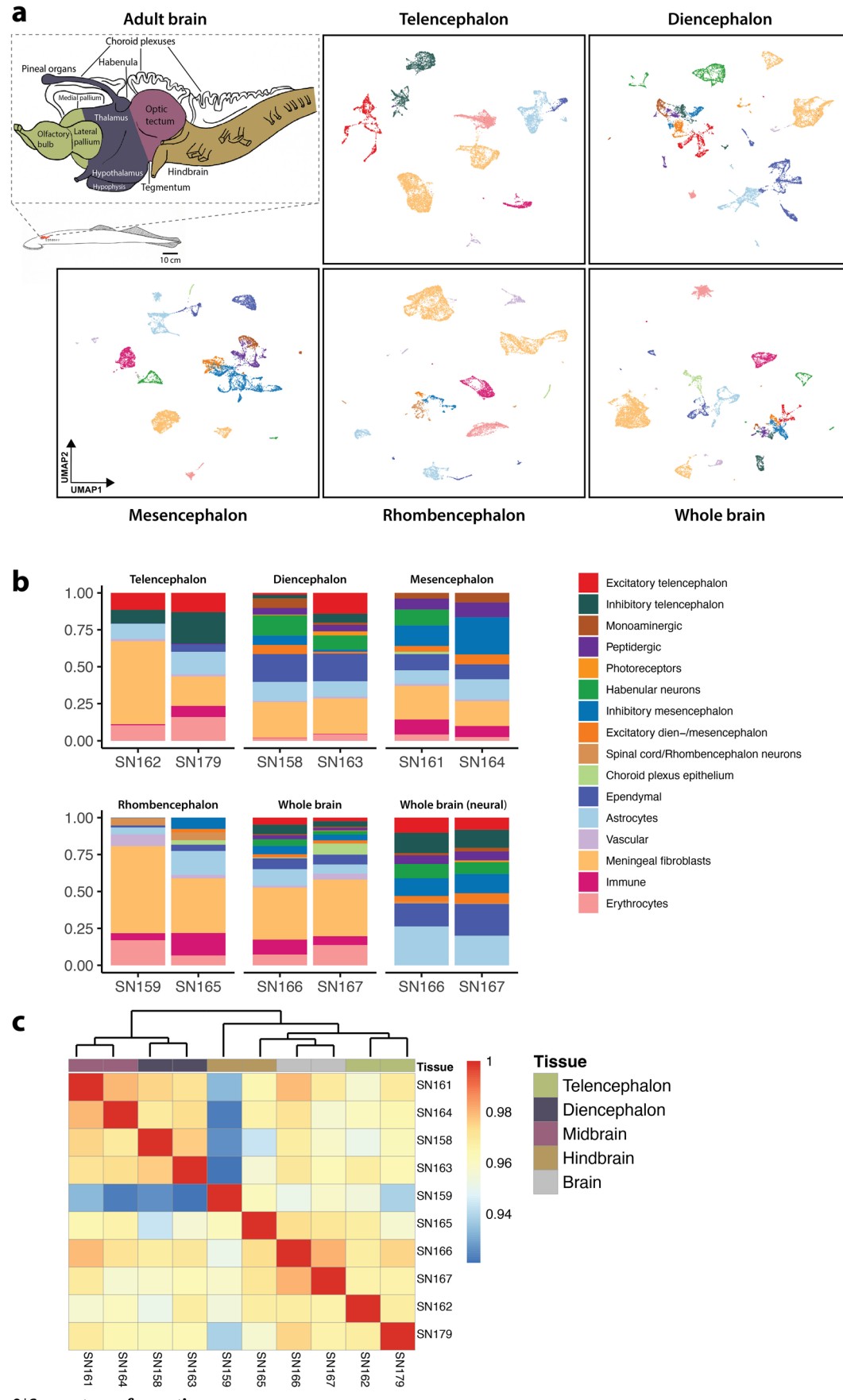

**Extended Data Fig. 3 | See next page for caption.**

**Extended Data Fig. 3 | Adult brain dataset. a**, Schematic of the sea lamprey adult brain showing the different regions dissected in this study and UMAP projections for each brain region. Each UMAP projection represents an integration of the biological replicates (as indicated in panel b) for each brain region, respectively. Additional information available in the interactive atlas (https://lampreybrain.kaessmannlab.org/adult.html). **b**, Barplots showing the proportions of each cell type group (as reported in a and Fig. 1b, c) for each sample. For whole brain samples, the proportions of neural cell types only are additionally showed. **c**, Heatmap showing clustered pseudobulk brain regions based on Spearman's ρ. Median ρ = 0.97.

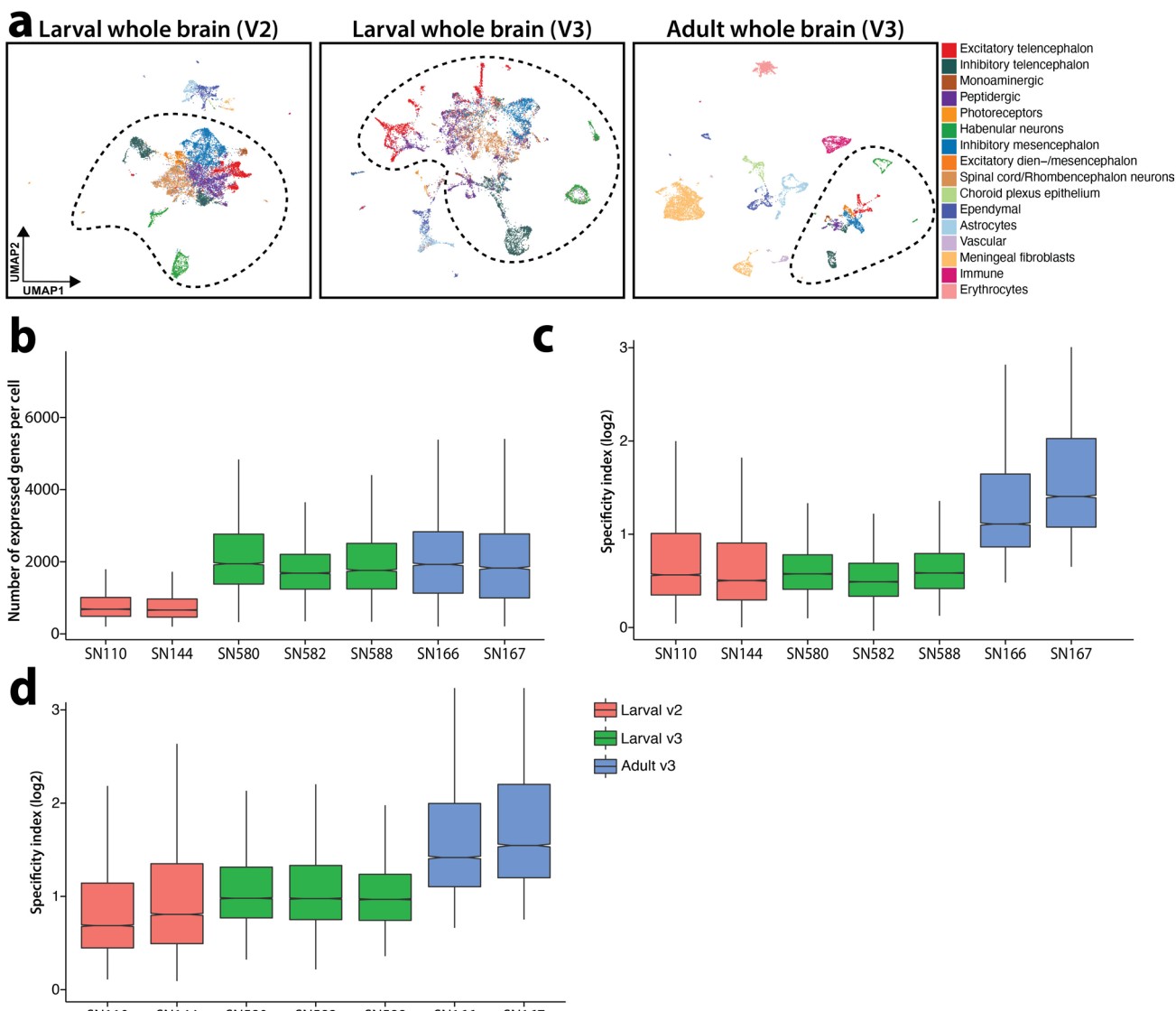

**Extended Data Fig. 4 | Differences between larval and adult datasets and between v2 and v3 kits. a**, UMAP of 13,301, 22,950, and 10,557 cells from the larval (v2; left), larval (v3; middle), and adult (right) whole brain datasets, respectively. Dashed lines highlight neuronal cells. **b**, Distribution of the number of expressed genes per cell for each whole brain sample. **c**, Distributions of specificity index scores for whole brains (neurons only). **d**, Same as b with each replicate downsampled to the same number of cells ($N = 1,000$). Statistical significance between groups in b, c, and d calculated with two-sided Mann-Whitney $U$ test ($P = 0.000013$). Boxplot annotation: bounds of box, Q1 (25th percentile), Q3 (75th percentile), Q3 - Q1 = IQR (interquartile range); centre, median; minimum, (Q1 − 1.5 * IQR); maximum (Q3 − 1.5 * IQR).

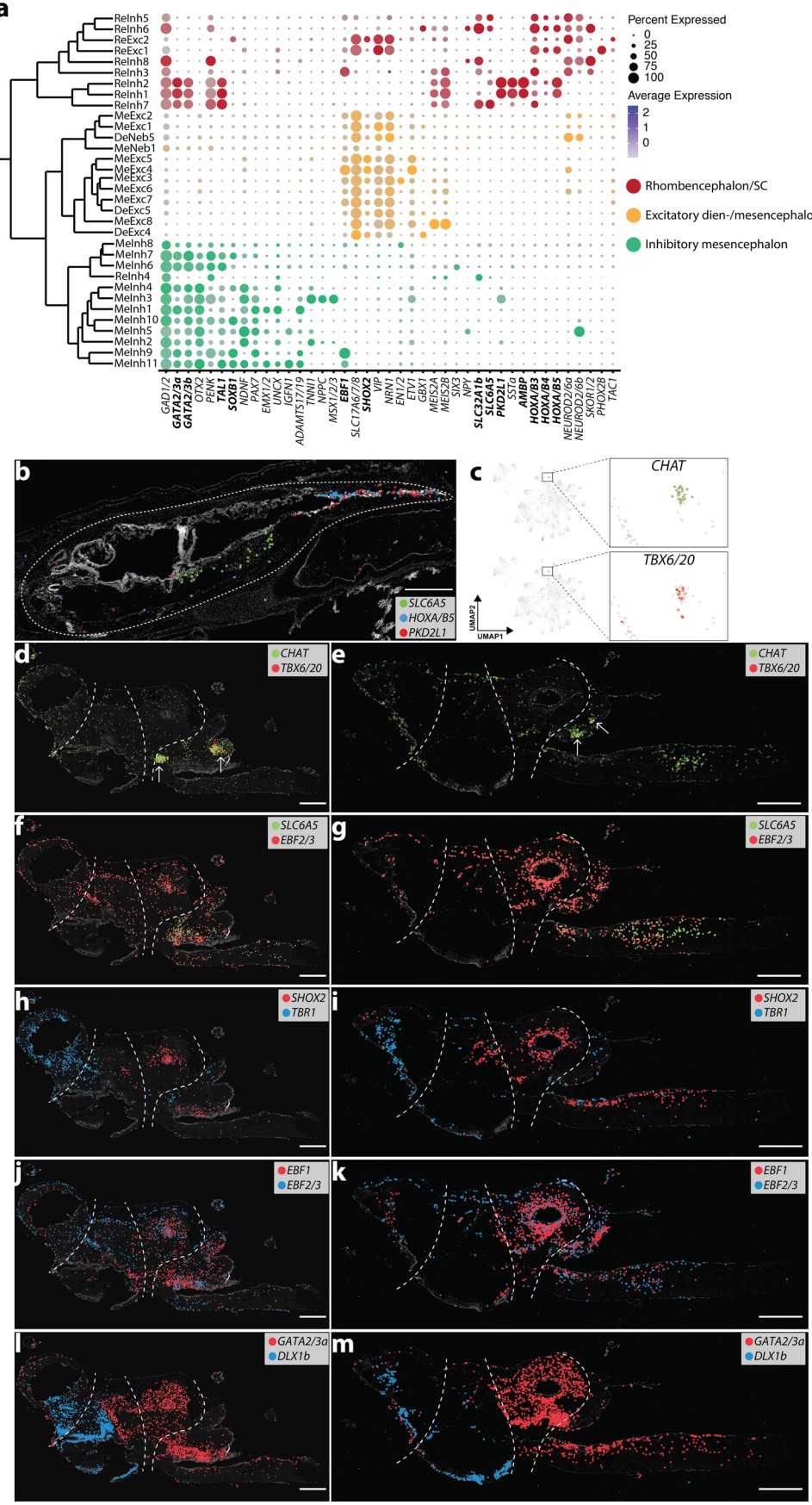

**Extended Data Fig. 5 | See next page for caption.**

**Extended Data Fig. 5 | Posterior forebrain, midbrain, and hindbrain neurons.**
**a**, Subtree from the dendrogram of Fig. 1c displaying the expression of selected marker genes for each cell type (gene names mentioned in main text are highlighted in bold). SC, spinal cord. **b**, Sagittal section (anterior end to the left) of a larval head (white dashed line outlines the brain) showing the expression of *SLC6A5*, *HOXA/B5*, and *PKD2L1*. Dashed lines separate the main brain regions.

**c**, UMAP projection of a larval hindbrain dataset showing the expression of *CHAT* and *TBX6/20*. **d-m** Sagittal sections (anterior end to the left) of the adult brain showing the expression of selected marker genes. Arrows in d and e indicate the putative location of cranial nerve nuclei. See Supplementary Fig. 2 for the ISS section schemes; scale bars, 500 μm.

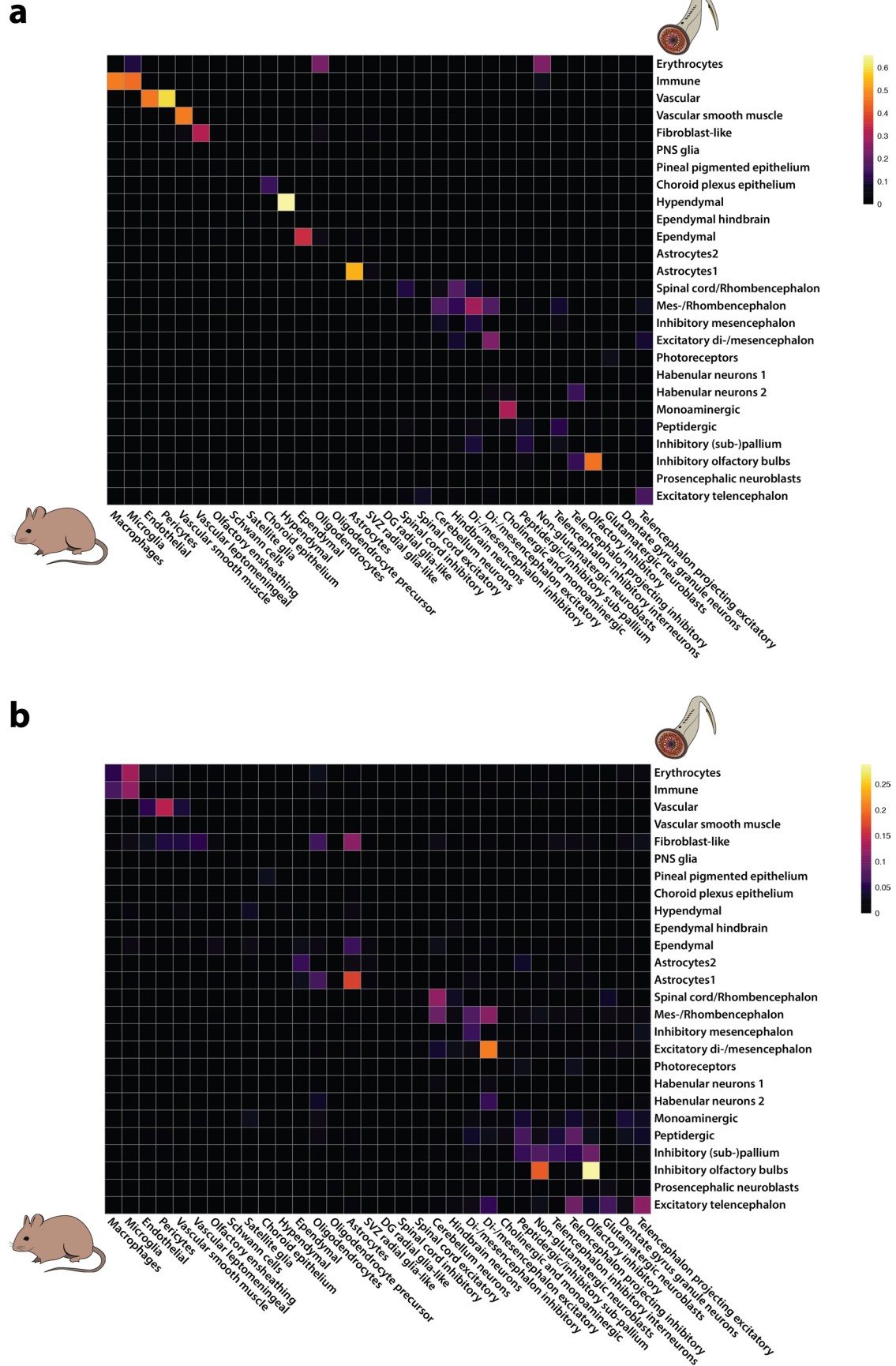

**Extended Data Fig. 6 | SAMap scores for all cell type groups. a, b,** Heatmaps of SAMap mapping scores for all groups of non-neuronal and neuronal cell types between mouse and lamprey, including all genes (a) and TF genes only (b).

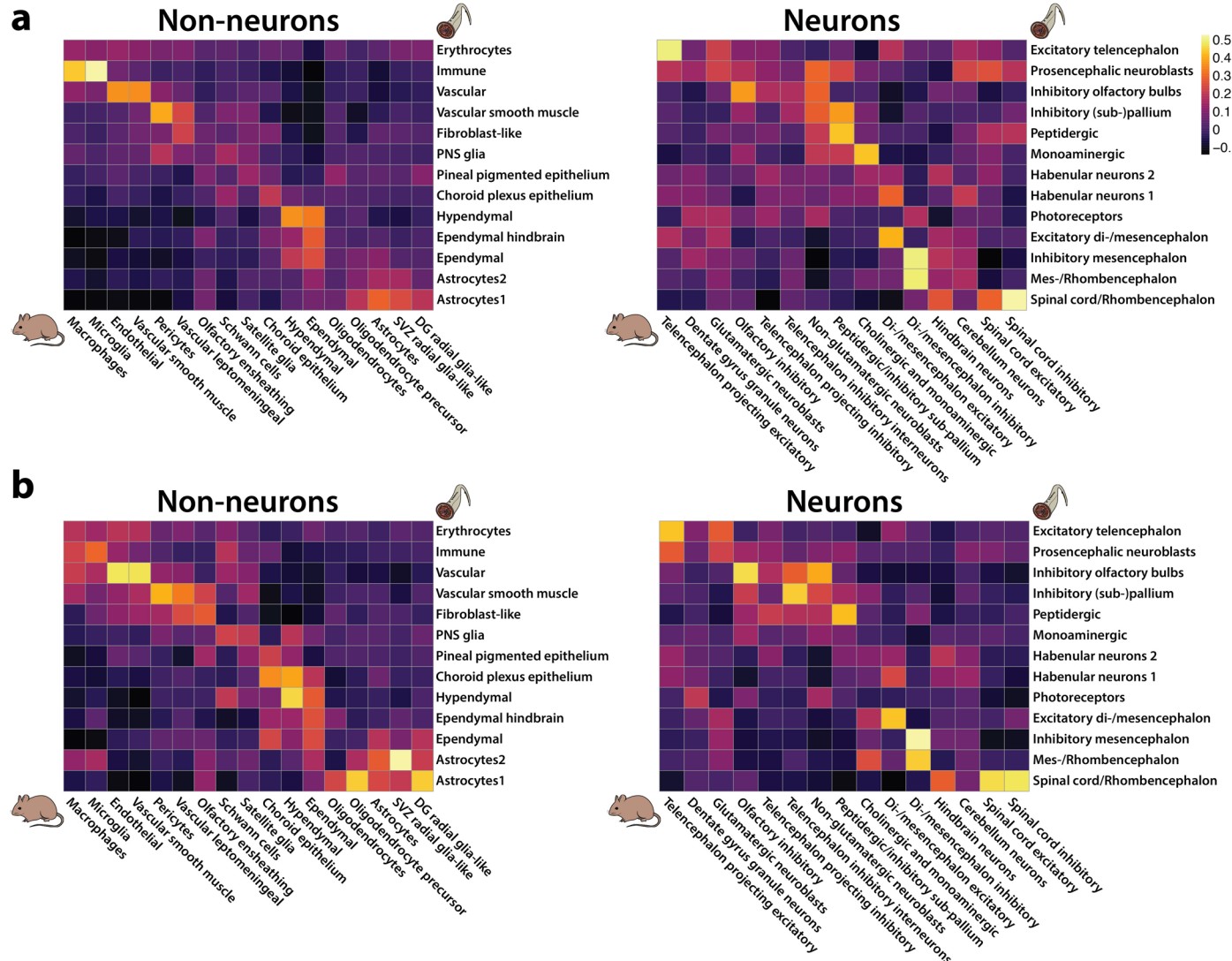

**Extended Data Fig. 7 | Correlations between cell type groups. a, b,** Heatmaps showing Pearson's correlation coefficients of specificity indexes of lamprey and mouse cell type groups for all orthologous genes (a) and for TFs only (b).

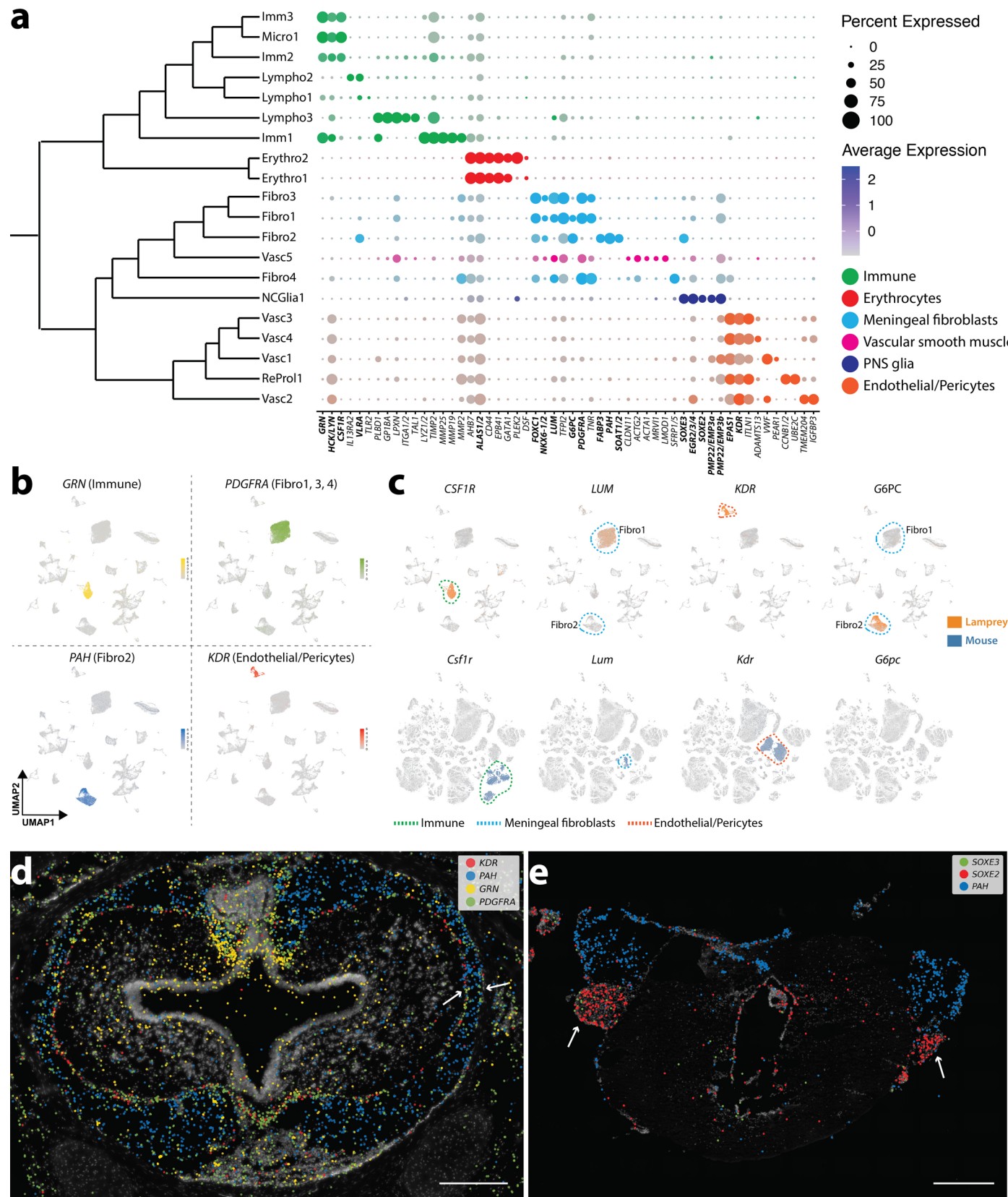

**Extended Data Fig. 8 | Blood, vascular, and PNS cells. a,** Subtree from the dendrogram of Fig. 1c displaying the expression of selected marker genes for each cell type (gene names mentioned in main text are highlighted in bold). **b,** Expression of marker genes for immune (*GRN*), meningeal (*PDGFRA*, *PAH*), and vascular (*KDR*) cells. **c,** Expression of lamprey markers and their mouse orthologs in the respective brain atlases (UMAPs). **d,** Coronal section of the larval telencephalon showing the spatial expression of the genes shown in b (same color code). Arrows mark leptomeningeal layers. **e,** Coronal section of the adult isthmic region (mesencephalon/rhombencephalon) showing the expression of *SOXE1* and *SOXE2* within cranial nerve roots (white arrows). See Supplementary Fig. 2 for ISS section schemes; scale bars, 500 μm.

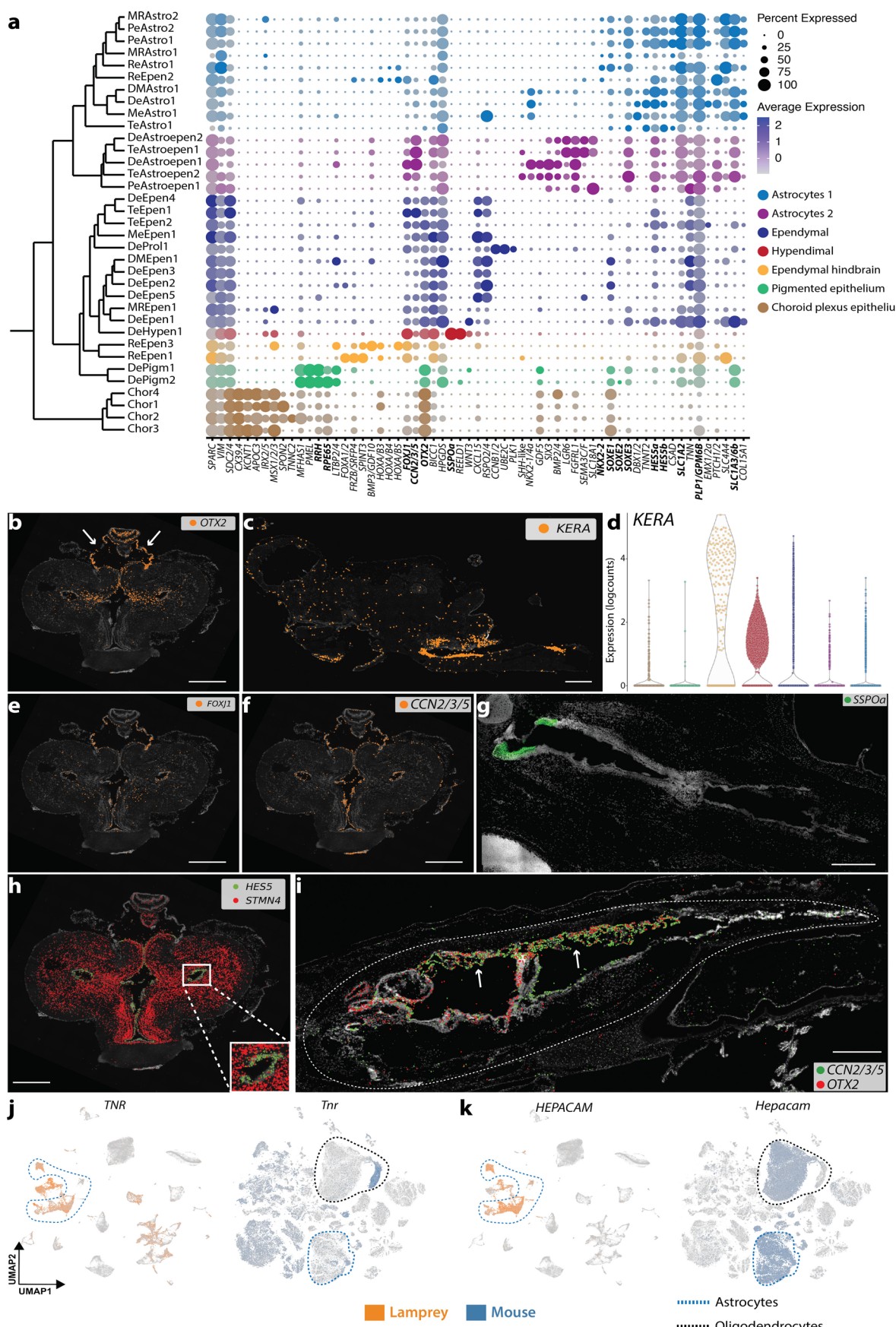

**Extended Data Fig. 9 | See next page for caption.**

**Extended Data Fig. 9 | Ependymoglial cells. a**, Subtree from the dendrogram of Fig. 1c displaying the expression of selected marker genes for each cell type (gene names mentioned in main text are highlighted in bold). **b**, Expression of *OTX2* within the adult telencephalon. White arrows point to choroid plexus. **c**, Expression of *KERA* within the adult brain (sagittal section; anterior end to the left), showing its concentration in the hindbrain. **d**, Violin plot displaying *KERA* expression among ependymoglial cell types; color code as in a. **e, f**, Expression of *FOXJ1* (e) and *CCN1-5* (f) within the adult telencephalon. **g**, Horizontal section of the larval brain (anterior end to the upper left corner) showing the expression of *SSPO* (smFISH) around the rostral end of the third ventricle (sub-commissural organ). **h**, Expression of *STMN4* (neurons) and *HES5* (astrocytes) within the adult telencephalon showing the periventricular localization of lamprey astrocytes. **i**, Sagittal section (anterior end to the left) of a larval head (brain enclosed within white dashed line) showing the expression of *CCN1-5* and *OTX2*. White arrows point to choroid plexuses. **j, k**, Expression of *TNR* (j), *HEPACAM* (k), and their mouse orthologs on the respective brain atlases. See Supplementary Fig. 2 for ISS section schemes; scale bars, 500 μm.

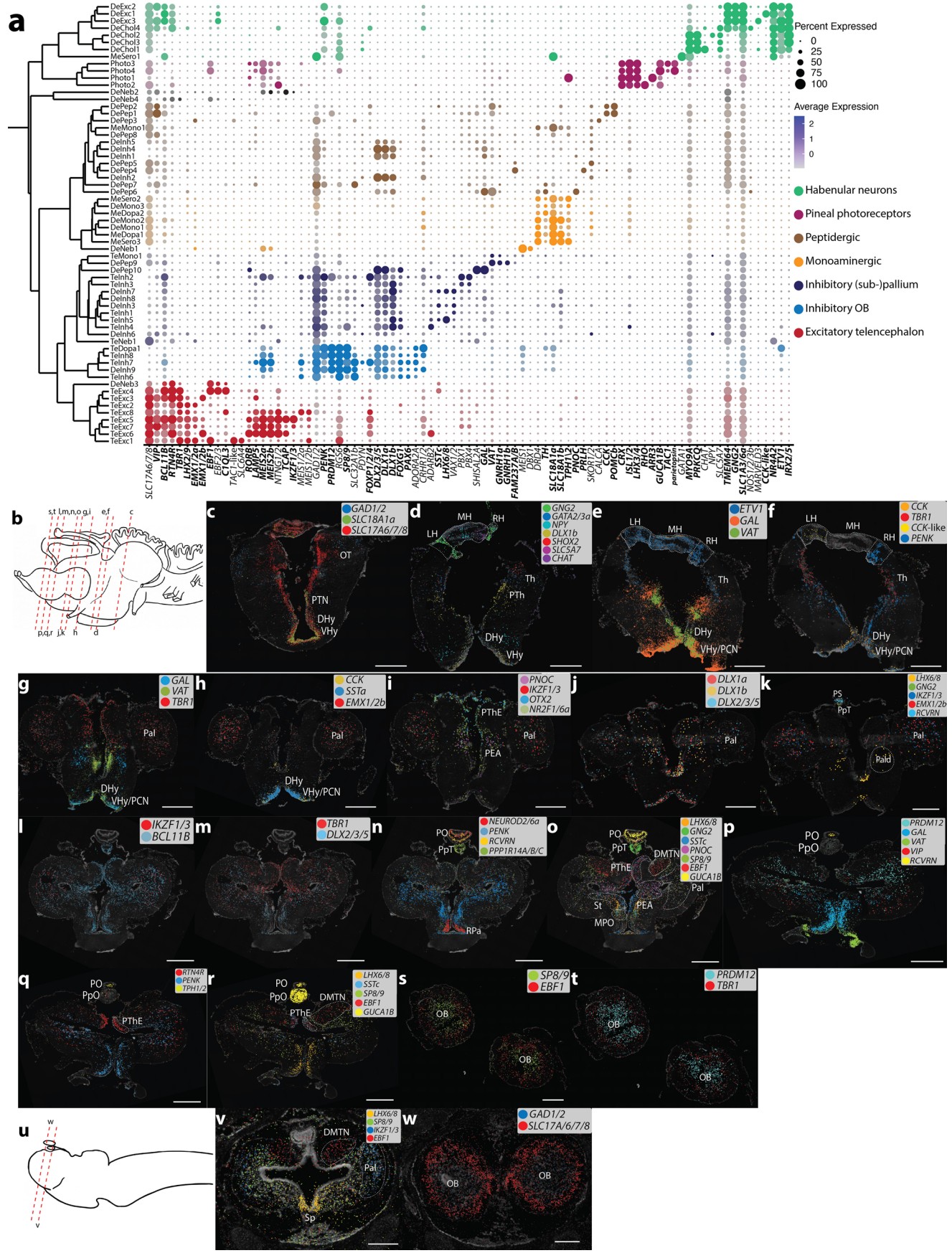

**Extended Data Fig. 10 | See next page for caption.**

**Extended Data Fig. 10 | Anterior forebrain and epithalamic neurons.**
**a**, Subtree from the dendrogram of Fig. 1c displaying the expression of marker genes for each cell type (gene names mentioned in main text are highlighted in bold). **b**, Section scheme of the adult brain. **c-t**, ISS maps of selected neuronal marker genes across various adult brain coronal sections (shown in b).
**u**, Section scheme of the larval brain. **v-w**, ISS maps of selected neuronal marker genes across various larval brain coronal sections (shown in u). DHy, dorsal hypothalamus; DMTN, dorsomedial telencephalic nucleus; LH, left habenula;

MH, medial habenula; MPO, medial preoptic nucleus; OB, olfactory bulb; OT, optic tectum; Pal, pallium; Pald, pallidum; PEA, pallial extended amygdala; PCN, postoptic commissure nucleus; PO, pineal organ; PpO, parapineal organ; PpT, parapineal tract; PS, pineal stalk; PTh, pre-thalamus; PThE, pre-thalamic eminence; PTN, posterior tubercle nucleus; RH, right habenula; RPa, rostral paraventricular area; Sp, septum; St, striatum; Th, thalamus; VHy, ventral hypothalamus. Scale bars, 500 μm.

# Reporting Summary

## Statistics

For all statistical analyses, confirm that the following items are present in the figure legend, table legend, main text, or Methods section.

| n/a | Confirmed | |
|---|---|---|
| ☐ | ☒ | The exact sample size (*n*) for each experimental group/condition, given as a discrete number and unit of measurement |
| ☐ | ☒ | A statement on whether measurements were taken from distinct samples or whether the same sample was measured repeatedly |
| ☐ | ☒ | The statistical test(s) used AND whether they are one- or two-sided <br> *Only common tests should be described solely by name; describe more complex techniques in the Methods section.* |
| ☒ | ☐ | A description of all covariates tested |
| ☐ | ☒ | A description of any assumptions or corrections, such as tests of normality and adjustment for multiple comparisons |
| ☐ | ☒ | A full description of the statistical parameters including central tendency (e.g. means) or other basic estimates (e.g. regression coefficient) AND variation (e.g. standard deviation) or associated estimates of uncertainty (e.g. confidence intervals) |
| ☒ | ☐ | For null hypothesis testing, the test statistic (e.g. *F*, *t*, *r*) with confidence intervals, effect sizes, degrees of freedom and *P* value noted <br> *Give P values as exact values whenever suitable.* |
| ☒ | ☐ | For Bayesian analysis, information on the choice of priors and Markov chain Monte Carlo settings |
| ☒ | ☐ | For hierarchical and complex designs, identification of the appropriate level for tests and full reporting of outcomes |
| ☐ | ☒ | Estimates of effect sizes (e.g. Cohen's *d*, Pearson's *r*), indicating how they were calculated |

*Our web collection on statistics for biologists contains articles on many of the points above.*

## Software and code

Policy information about availability of computer code

| Data collection | No software was used. |
|---|---|
| Data analysis | Open source software including CellRanger v3, GSNAP (version: 2018-03-01), StringTie (v1.3.4d), GffRead (v0.9.9), TransDecoder (v5.3.0), BlastP (v2.5.0+), HMMER (v3.2), BUSCO (v3), OrthoFinder (v2.3.11), MAFFT (v7.455), IQ-TREE (v1.6.12), Seurat (v3.1.5), LIGER (v0.5.0), SAMap (v0.2.3), pvclust (v2.2.0), Fiji (v2.0) |

For manuscripts utilizing custom algorithms or software that are central to the research but not yet described in published literature, software must be made available to editors and reviewers. We strongly encourage code deposition in a community repository (e.g. GitHub). See the Nature Portfolio guidelines for submitting code & software for further information.

## Data

Policy information about availability of data

All manuscripts must include a data availability statement. This statement should provide the following information, where applicable:
- Accession codes, unique identifiers, or web links for publicly available datasets
- A description of any restrictions on data availability
- For clinical datasets or third party data, please ensure that the statement adheres to our policy

Raw and processed bulk and single-cell RNA-seq data have been deposited to ArrayExpress with the accession numbers E-MTAB-11085 (bulk) and E-MTAB-11087 (single cell) (https://www.ebi.ac.uk/arrayexpress/).

# Field-specific reporting

Please select the one below that is the best fit for your research. If you are not sure, read the appropriate sections before making your selection.

☒ Life sciences ☐ Behavioural & social sciences ☐ Ecological, evolutionary & environmental sciences

For a reference copy of the document with all sections, see nature.com/documents/nr-reporting-summary-flat.pdf

# Life sciences study design

All studies must disclose on these points even when the disclosure is negative.

| | |
|---|---|
| Sample size | No statistical methods were used to determine sample size. Sample size was based on the number of individuals available (see Supplementary Tables 1 and 2). |
| Data exclusions | Low quality cells were excluded as described in Methods. |
| Replication | We generated 2 biological replicates for all collected brain samples, with the only exception of the larval diencephalon for which we generated 3 biological replicates (see Supplementary Table 2) |
| Randomization | Randomization was not used in this study. |
| Blinding | Blinding was not relevant to our study. Both data collection and analyses required an understanding of the nature of the sample being collected/analyzed. |

# Reporting for specific materials, systems and methods

We require information from authors about some types of materials, experimental systems and methods used in many studies. Here, indicate whether each material, system or method listed is relevant to your study. If you are not sure if a list item applies to your research, read the appropriate section before selecting a response.

## Materials & experimental systems

| n/a | Involved in the study |
|---|---|
| ☒ ☐ | Antibodies |
| ☒ ☐ | Eukaryotic cell lines |
| ☒ ☐ | Palaeontology and archaeology |
| ☐ ☒ | Animals and other organisms |
| ☒ ☐ | Human research participants |
| ☒ ☐ | Clinical data |
| ☒ ☐ | Dual use research of concern |

## Methods

| n/a | Involved in the study |
|---|---|
| ☒ ☐ | ChIP-seq |
| ☒ ☐ | Flow cytometry |
| ☒ ☐ | MRI-based neuroimaging |

# Animals and other organisms

Policy information about studies involving animals; ARRIVE guidelines recommended for reporting animal research

| | |
|---|---|
| Laboratory animals | Not used |
| Wild animals | Sea lamprey (Petromyzon marinus) |
| Field-collected samples | Sea lamprey (Petromyzon marinus) |
| Ethics oversight | All procedures involving animal care and experimentation were approved by: University of Colorado, Boulder Institutional Animal Care and Use Committee as described in protocol 2392; Regierungspräsidium Karlsruhe; Bioethics Committee of the University of Santiago de Compostela and the Xunta de Galicia Government; California Institute of Technology Institutional Animal Care and Use Committee (IACUC) protocol 1436. |

Note that full information on the approval of the study protocol must also be provided in the manuscript.

