## [Peer Review File · Nature Ecology & Evolution]

Peer Review Information

Journal: Nature Ecology & Evolution

Manuscript Title: A lamprey neural cell type atlas illuminates the origins of the vertebrate brain

Corresponding author name(s): Francesco Lamanna, Francisca Hervas-Sotomayor, Henrik Kaessmann

Editorial Notes:

Reviewer Comments & Decisions:

Decision Letter, initial version:

28th April 2022

Dear Henrik,

Your manuscript entitled "Reconstructing the ancestral vertebrate brain using a lamprey neural cell type atlas" has now been seen by three reviewers, whose comments are attached. The reviewers have raised a number of concerns which will need to be addressed before we can offer publication in Nature Ecology & Evolution. We will therefore need to see your responses to the criticisms raised and to some editorial concerns, along with a revised manuscript, before we can reach a final decision regarding publication.

In addition to the reviewers' comments, we felt that any conclusions about the vertebrate ancestor brain need to be toned down given the very limited number of species available for comparison. For example, we don't think it should be mentioned in the title.

It has come to our attention that you use unpublished data of the hagfish genome, *Eptatretus burgeri*. These data have been made publicly available in Ensembl under the Fort Lauderdale Agreement (https://www.ensembl.org/Eptatretus_burgeri/Info/Annotation) and it is our understanding that researchers should request permission to use the data. When resubmitting your manuscript, please include correspondence with the authors of the hagfish genome to this effect.

We therefore invite you to revise your manuscript taking into account all reviewer and editor comments. Please highlight all changes in the manuscript text file in Microsoft Word format.

2* If you have not done so already please begin to revise your manuscript so that it conforms to our Article format instructions at <http://www.nature.com/natecolevol/info/final-submission>. Refer also to any guidelines provided in this letter.

[REDACTED]

Nature Ecology & Evolution is committed to improving transparency in authorship. As part of our efforts in this direction, we are now requesting that all authors identified as 'corresponding author' on published papers create and link their Open Researcher and Contributor Identifier (ORCID) with their account on the Manuscript Tracking System (MTS), prior to acceptance. ORCID helps the scientific community achieve unambiguous attribution of all scholarly contributions. You can create and link your ORCID from the home page of the MTS by clicking on 'Modify my Springer Nature account'. For more information please visit www.springernature.com/orcid.

[REDACTED]

Reviewer expertise:

Reviewer #1: evo-devo of the vertebrate brain, comparative genomics

Reviewer #2: evolution of the vertebrate brain, single-cell genomics

Reviewer #3: brain development, evolution of the vertebrate brain

2Reviewers' comments:

Reviewer #1 (Remarks to the Author):

Here, Lamanna and colleagues provide the first cell type atlas of the brain of a cyclostome (jawless vertebrate), the sea lamprey *Petromyzon marinus*, an important model to understand the origin of vertebrate traits through comparative analysis with jawed vertebrates.

While this cell type atlas is an impressive resource, with a tremendously high cellular and molecular resolution, obtained through deep scRNA-seq and spatial transcriptomics, the ancestral reconstruction they obtain of the vertebrate brain does not significantly change or improve our previous views, obtained through decades of comparative analysis of lamprey and hagfish brains with jawed vertebrates. Their main find of a large set of homologous cell type families between the lamprey and the mouse, implying their presence in the last common ancestor of vertebrate is not surprising and largely expected, given the great conservation of general regionalization and gene expression patterns between lamprey, hagfish and gnathostome brains reported through countless past studies. As such, most of the conclusions they reach through the manuscript are mostly confirmatory. In fact, a large amount of the cell types described have been previously reported and just confirmed here (e.g., inhibitory reticulospinal neurons: ref. 34 in the article; CSF-c neurons: ref. 38 in the article; habenular nuclei in the epithalamus: ref. 40; etcetera). See also Villar-Cerviño et al. (2011; <https://doi.org/10.1002/cne.22597>) and Pombal et al. (2011; <https://doi.org/10.3389/fnana.2011.00020>), just to mention a couple of extra examples not cited but familiar to the authors. Other confirmatory instances are the lack of myelin from the lamprey PNS and the hypothetical origin of oligodendrocytes from astrocyte-like glia in the gnathostome lineage (refs. 26, 27 in the ms). Interestingly, the authors propose, based on their analysis of larval and adult brains, that the brain of the last common vertebrate ancestor likely lack cerebellar cell types. However, as they even mention in the manuscript, a recent report in the lamprey and hagfish has identified these in hagfish and lamprey embryos (Sugahara et al., 2021; <https://doi.org/10.3389/fcell.2021.700860>). This highlights the importance of comparative embryological studies which are not considered here in order to fully understand the evolution of vertebrates. I am sure this will be a tremendously important resource for comparative neuroscientists, but as it stands now, this manuscript is largely descriptive and mostly confirmatory.

I also have some technical concerns about the differences found between larval and adult brains. One of the earliest findings in the article is that "the adult brain is characterized by larger numbers of expressed genes per cell and greater cell type specificities of gene expression compared to the ammocoete brain". However, the larval and adult brain data are obtained using different versions of the kit, v2 and v3, respectively. The authors try to control for technical differences by sequencing a whole larval brain with a v3 kit (library SN352). While the authors conclude that data obtained from this library confirm the putative biological differences observed, how is this exactly concluded? The bar plot in Extended Data Fig. 4b shows that they find what seems a significantly larger number of expressed genes per cell in SN352 (v3) than in the v2 data. Also, Extended Data Fig. 4c shows that SN352 data is in an intermediate position between larval v2 data and adult v3 data. Without statistical

3testing is difficult to say that they can rule out technical variation between v2 and v3 kits. This considered, their sentence “This result is robust to controls for technical differences between the datasets” (Lines 93-94) seems an exaggerated claim when lacking statistical support. I strongly recommend, at least, appropriate statistical testing, and, optimally, sequencing more larval samples with v3 kits to make the different samples comparable to properly confirm that biological causes are behind the differences observed. A bar plot showing the proportions of each cell types obtained from SN352 should also be included in Extended Data Fig. 2 together to those from SN110 and SN144 to test for equivalent recoveries of cell types using v2 and v3 kits.

The proportions of each cell types look very different between different libraries of the same brain region in the adult (Extended Data Fig. 3). I’m no expert in scRNA-seq, so is this a technical issue? I might be wrong, but I would expect that different libraries of the same brain region would count as sort of biological replicates, and thus certain correlation would be expected. If this is not observed, are the data still reliable? Is there a statistical way to measure the correlation and/or the variation in the data from different libraries? This is not the case in the larval brain, what makes me think there might be a problem with the adult samples/seq processing/data, but I might be wrong.

Given all of the above, I am not sure if the hypothesis about the undifferentiated nature of the larval brain holds true or not.

Continuing on Extended Data Figs. 2 and 3, are the UMAP projections a mix of all libraires of each brain region, or do they come from particular libraries of each of the regions? Please indicate.

Again, I’m no expert on scRNA-seq, so it surprises me that the adult whole brain UMAP projection in Extended Data Fig. 3a/4a is quite different from that in main Fig. 1b.

The authors conclude in their first section that “The hierarchical relationship of cell types in the lamprey brain is very similar to that observed for a reference mammalian brain atlas (i.e., that of the mouse), which suggests that all vertebrates share common general cellular and molecular organization of neural tissues that was established during the evolution of the vertebrate stem lineage.” However, as I mention before, this is expected, given that we already know, from all the previous literature published, that the anatomical, molecular and even cellular organization of both cyclostome and gnathostome brains are greatly conserved.

- Some minor issues:

1) Explain what ‘scRNA-seq’ stands for the first time is mentioned

2) L57: add Albuixech-Crespo et al. (2017; <https://doi.org/10.1371/journal.pbio.2001573>) after the sentence about the brain molecular architecture of invertebrate chordates, together with references 3,4.

3) L176 and Extended Data Fig. 5 (L1025): change ‘hematopoietic cells’ for ‘blood cells’. ‘Poiesis, derived from ancient Greek, means the ‘creation of’. Hematopoietic is thus used in the case of stem and progenitor cells, which create all blood cell types. Unless you are exclusively describing those stem cells, then ‘blood cells’ is a more appropriate term.

44) There are no HOXA/B3, HOXA/B4 and HOXA/B5 genes in the lamprey (Mehta et al., 2013: <https://doi.org/10.1073/pnas.1315760110>; Smith et al., 2018: <https://doi.org/10.1038/s41588-017-0036-1>; Parker et al., 2019: <https://doi.org/10.1016/j.ydbio.2019.05.001>). Some of the authors have a large experience working with the lamprey, including Hox genes. Please, name the genes properly. Another instance is that sometimes the authors mention OTX1/2 (Line 280) but elsewhere OTX2. Upon checking the corresponding transcript (MSTRG.8275), it appears to be the old lamprey gene OtxA, which is Otx2 proper based on synteny conservation (Yamamoto et al., 2020: <https://doi.org/10.1016/j.celrep.2019.12.072>). I strongly suggest the authors to add an extra column in Supplementary Tables S4 and/or S8 with a reference to the first report (if any) of those genes that have been previously identified and described in the lamprey.

5) Following on gene nomenclature, lamprey literature does not tend to write, with exceptions, lamprey gene symbols in capital letters (a writing style that I think is usually restricted to human genes). Although the literature contains examples of all writing styles for the lamprey, using a fish nomenclature style is strongly recommended (see Zebrafish Nomenclature Conventions): thus, lamprey gene symbols should be written completely in lowercase and in italics. If you agree, change this throughout the manuscript, figures and supplementary files and data.

6) The authors should cite the articles that first provided the gene sets they used for orthology assignment. Ensembl is just the hosting genome browser of these data (*Ciona intestinalis*, *Eptatretus burgeri*, *Callorhinchus milii*, *Lepisosteus oculatus*, *Danio rerio*, *Latimeria chalumnae*, *Xenopus tropicalis*, *Gallus gallus*), but not the ones generating the data (except for mouse and human, for which they do their own annotation I think). Accordingly, since you use Ensembl, the authors should also cite it appropriately (see <https://www.ensembl.org/info/about/publications.html>)

7) Reference section: Species names in the titles of literature cited needs to be in italics. E.g., *Petromyzon marinus* in references 37, 42, 49, 50. Same goes for gene names. For instance, in reference 15, *Gata2* should go in italics, as per the original title (<https://doi.org/10.1242/dev.029900>). I haven't checked all the literature thoroughly, so please, double check. The authors should be especially careful about correctly spelling the names of authors cited.

Reviewer #2 (Remarks to the Author):

In this manuscript, Lamanna, Hervas-Sotomayor and colleagues describe a cell type atlas of the lamprey brain. In their study, they identify 151 and 120 types of neurons in lamprey adults and larvae, respectively. The spatial localization of many of these types is identified by in situ sequencing.

This is a paper that comparative neurobiologists have been waiting for a very long time. As early-branching, jawless vertebrates, lampreys occupy a key phylogenetic position to infer the early evolution of the vertebrate brain. The lamprey brain has puzzled neuroanatomists for decades. The existence of a cerebellum in lampreys is still unclear. The identity of subregions in the lamprey telencephalon is also unclear. For example, the telencephalic area that the authors refer to as "medial

5pallium" has been considered by others a part of the diencephalon (the prethalamic eminence).

An important and thought-provoking finding is the expression of oligodendrocyte markers in ependymogial cells. This leads to the interesting idea that oligodendrocytes and ependymogial cells might have evolved from multifunctional glial cells.

The discovery of broad molecular similarities between what the authors call "neuron type families" is also a nice result but it is less surprising, given that the homology of broad brain regions such as the olfactory bulb or the hypothalamus is well settled. The reader is left wondering about what makes the brain of a lamprey different from the brain of a mouse at the cell type level.

My biggest critique to this manuscript is that it shies away from providing the answers that readers are looking for. Part of the problem is that the dataset is so large, that it is impossible to describe all of it in detail. Furthermore - if I may - the authors seem too concerned about following the neuroanatomy tradition instead of listening to their own beautiful data. In the last decade, molecular studies on brain evolution have shattered decades of neuroanatomy dogma, providing surprising re-interpretations of entire brain regions (an example that comes to mind is the discovery of a telencephalon in amphioxus by co-author Detlev Arendt and colleagues). Here I could find at least three examples (there are probably more...) where the molecular data described in this manuscript are in contrast with dominant interpretations of the lamprey brain:

1. Striatum. The term striatum is used inconsistently in the mammalian literature; most people equate the term striatum to the dorsal striatum (caudoputamen in mouse), but a less common usage includes other subpallial areas, such as the "striatum-like amygdalar nuclei" (see Allen Brain Reference Atlas). The authors should qualify this term precisely before using it.

The authors follow the work of previous scholars and identify as "striatum" a region of the lamprey subpallium nested between the preoptic area and the pallium. This region corresponds to cluster TeInh5, and expresses genes like *lhx6/8*, *galanin*, *gbx1*, and *nkx2-1*. From this, I came to wonder whether the lamprey "striatum" corresponds partially or entirely to the subpallial amygdala. In amniotes, *lhx6*, *lhx8*, *galanin*, and *nkx2-1* are not markers of differentiated medium spiny neurons (MSNs), and *gbx1* is expressed in MSNs at very low levels, if at all (see mousebrain.org). Typical MSN markers, such as *Sp8/9* and *Isl1*, are not expressed in TeInh5. *Lhx6*, *lhx8*, *galanin*, and *nkx2-1*, instead, are expressed prominently in the subpallial portion of the extended amygdala (bed nucleus of the stria terminalis, central amygdala, medial amygdala, see for example Garcia-Lopez et al *Journal of Comparative Neuroscience* 2008 PMID: 17990271, Choi et al *Neuron* 2005 PMID: 15944132, and many others). Topologically, the idea that the dorsal striatum is immediately adjacent to the preoptic area is inconsistent with data from tetrapods. The tetrapod dorsal striatum is adjacent medially to the pallidum, dorsally to the ventral pallium, and caudally to the amygdala. Therefore, there is no boundary between the preoptic region and the dorsal striatum; instead, there is a boundary between the preoptic area and subpallial amygdala. Back to figure 4d, one could speculate that at least the medial *lhx6/8*-expressing portion of the "striatum" is subpallial amygdala.

This interpretation may seem in conflict with the results of Grillner and colleagues on the connectivity of the lamprey basal ganglia, proposing the homology of lamprey and mammalian dorsal striatum. However, Swanson (figure 5 PMID: 12724158) pointed out that the circuits connecting parts of the cortex, amygdala, and brainstem motor systems are organized exactly like the dorsal striatum circuits compared by Grillner. One could speculate that these systems are evolutionarily related as serial

6homologs (reviewed in Loonen and Ivanova 2016 PMID: 27920666). The real question that remains still unaddressed is whether lampreys do have homologs of dorsal striatum medium spiny neurons.

2. Medial pallium. Pombal, Puelles and colleagues (PMID: 19729892) proposed (following previous literature) that the so-called medial pallium is a rostral enlargement of the lamprey prethalamic eminence. Strikingly, the authors show expression of *ebf1* in this area (cluster TeExc4, Figure 4g) and also in the DMTN. *Ebf* genes are not detected in the medial pallium/hippocampus of tetrapods. Instead, *Ebf* genes, together with *Lhx1/5* and *Tbr1*, are markers of the prethalamic eminence of tetrapods. These observations call for a more rigorous assessment of the diencephalic nature of the medial pallium.

3. Lateral pallium. The nature of the lamprey lateral pallium is strongly debated. Grillner and colleagues have proposed its homology with the neocortex, on the basis of projections. Striedter and Northcutt (PMID: 34175847) have recently argued that the dorsal pallium might be an amniote innovation! This would imply that the lamprey lateral pallium is homologous to the ventrolateral pallium of amniotes.

This controversy could be solved by molecular data. At the very least, the hypothesis of Grillner and colleagues would be supported if similar lamprey and mammalian projection types were to express the same transcription factors. Extended figure 8 does not provide an exhaustive list, but it shows that *etv1/er81*, a marker of corticofugal projections in amniotes (Dugas-Ford et al PMID: 23027930), is not expressed in lateral pallium glutamatergic neurons (how about *fezf2*? *Sox5*? *Satb1/2*?). *Rorb* is uninformative, because it is expressed in both the dorsal and the ventral pallia of amniotes.

Some or all of the points above could be followed up with in depth comparison with data from mammals and other vertebrates, using state-of-the-art computational techniques.

Additional comments:

4. Is there any evidence from this dataset on the presence or absence of a caudal ganglionic eminence in lamprey?

5. The use of the term "lamprey astrocytes" is misleading. The cells described in the paper are ependymogial cells, similar to the ependymogial cells of other non-mammalian vertebrates. They are quite different from mammalian astrocytes, although evolutionarily related in a complex way (see also Laywell et al PMID: 11095732).

6. Is there a GABAergic prethalamus?

7. I was surprised to see *foxg1* only in telencephalic GABAergic neurons. Can technical artifacts (e.g. gene annotation, isoforms) be excluded?

8. The website <https://lampreybrain.kaessmannlab.org/adult.html> is great, but having to switch between gene ids (MSTRG.10634) and gene names (*GAD1*) makes data exploration cumbersome.

Minor suggestion:

79. In Extended Figure 8, it would be helpful to separate larval and adult panels in different rows.

Reviewer #3 (Remarks to the Author):

This is an exceptionally well-written report of a series of scRNAseq done in larval and adult sea lamprey brain and of the separate 4 brain areas: telencephalon, diencephalon, mesencephalon and rhombencephalon. The quality of analysis of these scRNAseq datasets is outstanding. The conclusions are of great interest for brain evolution and the comparisons with published mouse scRNAseq datasets bring new insights in the field. They find conserved cell type 'families' building for the first time a framework for the ancestral brain cell type repertoire. They also identify modules of evolution and confirm the absence of oligodendrocytes and thereby absence of myelination. The authors found part of the oligodendrocyte transcriptome signature in astrocytes, supporting the current view of them stemming from astrocytes. The clustering also confirms the absence of a proper cerebellum in the hindbrain and the presence of MGE and LGE in the subpallium of the telencephalon. In the telencephalon, the analysis deepens in generating findings on the type of projection modalities of the pallial excitatory neurons.

Overall, this is an outstanding study of single cell transcriptomics accompanied by the delivery of a very useful web tool for the community. Specific points of improvement are listed below.

Specific comments:

[Lines 90-97] Authors should perform statistical analyses between samples, and comment whether different levels of expression was observed between the two chemistries at Larval stage.

[Lines 670-674] Clearer description of how the Specificity Indexes was quantified will benefit readers. For instance, did the authors use raw counts or normalized counts for this quantification, and was this value normalised to the number of cell types from each species?

The beautiful and complex set of findings made and highlighted in the Discussion section would deserve a Figure summarising the key findings.

*****END*****

Author Rebuttal to Initial comments

8Decision Letter, first revision:

21st June 2023

Dear Henrik,

Thank you for submitting your revised manuscript "A lamprey neural cell type atlas illuminates the origins of the vertebrate brain" (NATECOLEVOL-220215899A). It has now been seen again by the original reviewers and their comments are below. The reviewers find that the paper has improved in revision, and therefore we'll be happy in principle to publish it in Nature Ecology & Evolution, pending minor revisions to satisfy the reviewers' final requests and to comply with our editorial and formatting guidelines.

Please email us a copy of the file in an editable format (Microsoft Word or LaTeX)-- we can not proceed with PDFs at this stage.

[REDACTED]

Reviewer #1 (Remarks to the Author):

The authors have done here an impressive revision of their work, and I feel humbled that some of my comments have been found helpful. I also appreciate their explanation about the technique, which I'm not familiar with. Especially, I appreciate their extra work put on the larvae brain, with the sequencing of three more libraries (SN580, SN582, SN588), which now seems to support their results more robustly (though no differences in the overall cell type composition were found). I don't think this work could be better done than it is now, the authors have done everything they could in their revision and have satisfactorily answered most of the reviewer's concerns. However, I still think (and this is probably subjective) that this work doesn't provide significant novel findings: it remains mostly confirmatory and so acknowledged by the authors in many instances. On the other hand, the outstanding resource here provided will have a significant impact in the work of those studying the evolution of the vertebrate brain.

Reviewer #2 (Remarks to the Author):

9Lamanna, Hervas-Sotomayor and colleagues extensively revised the initial manuscript, carefully following all reviewers' suggestions. This revised version is excellent, and highlights the new insights this scRNAseq datasets provides on the early evolution of the vertebrate brain.

I am grateful to see that the authors found my comments and speculations helpful and that they tested those hypotheses with additional analyses. I am slightly puzzled by the relative topological positions of the newly-identified striatal medium-spiny neurons and the subpallial amygdala, but I do not feel competent to give feedback at this level of detail, and I trust that the authors considered all alternative scenarios carefully.

The new results on the pallium are exciting and will surely instigate lots of discussion and follow-up work. It might be worth strengthening them by mentioning explicitly in the text that "medial pallium" and DMTN do not express typical markers of pallial glutamatergic neurons (emx genes, foxg1). I have also been wondering whether tetrapod medial pallium markers, such as prox1 and zbtb20, are expressed in the "medial pallium" and DMTN (their absence would further support the idea that the "medial pallium" is, in reality, part of the diencephalon).

Congratulations on this beautiful work!
Maria Antonietta Tosches

Reviewer #3 (Remarks to the Author):

The authors have improved substantially the manuscript and provided all clarifications I requested. They also responded to the other reviewers' questions with great depth. The revised version of the manuscript is excellent.

Note: Regarding the FOXG1 question raised by one reviewer, the result is not really surprising. Even in mouse and human this gene is expressed differentially in the foetal and adult telencephalon, with high level of expression in MGE, LGE and much lower level in cortex, with some of the excitatory neuronal populations not expressing significant level.

Our ref: NATECOLEVOL-220215899A

30th June 2023

10Dear Dr. Kaessmann,

Thank you for your patience as we've prepared the guidelines for final submission of your Nature Ecology & Evolution manuscript, "A lamprey neural cell type atlas illuminates the origins of the vertebrate brain" (NATECOLEVOL-220215899A). Please carefully follow the step-by-step instructions provided in the attached file, and add a response in each row of the table to indicate the changes that you have made. Please also check and comment on any additional marked-up edits we have proposed within the text. Ensuring that each point is addressed will help to ensure that your revised manuscript can be swiftly handed over to our production team.

****We would like to start working on your revised paper, with all of the requested files and forms, as soon as possible (preferably within two weeks). Please get in contact with us immediately if you anticipate it taking more than two weeks to submit these revised files.****

In recognition of the time and expertise our reviewers provide to Nature Ecology & Evolution's editorial process, we would like to formally acknowledge their contribution to the external peer review of your manuscript entitled "A lamprey neural cell type atlas illuminates the origins of the vertebrate brain". For those reviewers who give their assent, we will be publishing their names alongside the published article.

Nature Ecology & Evolution offers a Transparent Peer Review option for new original research manuscripts submitted after December 1st, 2019. As part of this initiative, we encourage our authors to support increased transparency into the peer review process by agreeing to have the reviewer comments, author rebuttal letters, and editorial decision letters published as a Supplementary item. When you submit your final files please clearly state in your cover letter whether or not you would like to participate in this initiative. Please note that failure to state your preference will result in delays in accepting your manuscript for publication.

Cover suggestions

As you prepare your final files we encourage you to consider whether you have any images or illustrations that may be appropriate for use on the cover of Nature Ecology & Evolution.

11We accept TIFF, JPEG, PNG or PSD file formats (a layered PSD file would be ideal), and the image should be at least 300ppi resolution (preferably 600-1200 ppi), in CMYK colour mode.

Nature Ecology & Evolution has now transitioned to a unified Rights Collection system which will allow our Author Services team to quickly and easily collect the rights and permissions required to publish your work. Approximately 10 days after your paper is formally accepted, you will receive an email in providing you with a link to complete the grant of rights. If your paper is eligible for Open Access, our Author Services team will also be in touch regarding any additional information that may be required to arrange payment for your article.

Please note that *Nature Ecology & Evolution* is a Transformative Journal (TJ). Authors may publish their research with us through the traditional subscription access route or make their paper immediately open access through payment of an article-processing charge (APC). Authors will not be required to make a final decision about access to their article until it has been accepted. [Find out more about Transformative Journals](https://www.springernature.com/gp/open-research/transformative-journals)

Authors may need to take specific actions to achieve [compliance with funder and institutional open access mandates](https://www.springernature.com/gp/open-research/funding/policy-compliance-faqs). If your research is supported by a funder that requires immediate open access (e.g. according to [Plan S principles](https://www.springernature.com/gp/open-research/plan-s-compliance)) then you should select the gold OA route, and we will direct you to the compliant route where possible. For authors selecting the subscription publication route, the journal's standard licensing terms will need to be accepted, including <https://www.nature.com/nature-portfolio/editorial-policies/self-archiving-and-license-to-publish>. Those licensing terms will supersede any other terms that the author or any third party may assert apply to any version of the manuscript.

[REDACTED]

[REDACTED]

Reviewer #1:

Remarks to the Author:

The authors have done here an impressive revision of their work, and I feel humbled that some of my comments have been found helpful. I also appreciate their explanation about the technique, which I'm not familiar with. Especially, I appreciate their extra work put on the larvae brain, with the sequencing of three more libraries (SN580, SN582, SN588), which now seems to support their results more robustly (though no differences in the overall cell type composition were found). I don't think this work could be better done than it is now, the authors have done everything they could in their revision and have satisfactorily answered most of the reviewer's concerns. However, I still think (and this is probably subjective) that this work doesn't provide significant novel findings: it remains mostly confirmatory and so acknowledged by the authors in many instances. On the other hand, the outstanding resource here provided will have a significant impact in the work of those studying the evolution of the vertebrate brain.

Reviewer #2:

Remarks to the Author:

Lamanna, Hervas-Sotomayor and colleagues extensively revised the initial manuscript, carefully following all reviewers' suggestions. This revised version is excellent, and highlights the new insights this scRNAseq datasets provides on the early evolution of the vertebrate brain.

I am grateful to see that the authors found my comments and speculations helpful and that they tested those hypotheses with additional analyses. I am slightly puzzled by the relative topological positions of the newly-identified striatal medium-spiny neurons and the subpallial amygdala, but I do not feel competent to give feedback at this level of detail, and I trust that the authors considered all alternative scenarios carefully.

The new results on the pallium are exciting and will surely instigate lots of discussion and follow-up work. It might be worth strengthening them by mentioning explicitly in the text that "medial pallium" and DMTN do not express typical markers of pallial glutamatergic neurons (emx genes, foxg1). I have also been wondering whether tetrapod medial pallium markers, such as prox1 and zbtb20, are expressed in the "medial pallium" and DMTN (their absence would further support the idea that the "medial pallium" is, in reality, part of the diencephalon).

13Congratulations on this beautiful work!
Maria Antonietta Tosches

Reviewer #3:

Remarks to the Author:

The authors have improved substantially the manuscript and provided all clarifications I requested. They also responded to the other reviewers' questions with great depth. The revised version of the manuscript is excellent.

Note: Regarding the FOXP1 question raised by one reviewer, the result is not really surprising. Even in mouse and human this gene is expressed differentially in the foetal and adult telencephalon, with high level of expression in MGE, LGE and much lower level in cortex, with some of the excitatory neuronal populations not expressing significant level.

Author Rebuttal, first revision:1 We would like to thank the editor and the referees for the positive appreciation of our manuscript and the very useful
 and constructive comments, which helped us to clarify various aspects of our work. Moreover, they also motivated us
 to revisit our data, which allowed us to obtain additional novel results and insights – including reinterpretations of our
 original findings – in particular regarding structures and cell types of the lamprey telencephalon and their evolutionary
 relationships with those from jawed vertebrates. We also generated additional scRNA-seq data for lamprey larvae, to
 clarify biological/technical similarities/differences between the adult and larval datasets (the delay in being able to
 source new samples underlies the delay in completing and submitting our revised manuscript).

Please find below our detailed responses to the editor’s and referees’ comments. We used the following font
 emphases/colors to facilitate readability:

**Comments from referees**

Our response to the referees

**Changes to the manuscript (tables, figures, page, and line numbers always refer to the revised manuscript)**

17 **Referee #1 (Remarks to the Author):**

**While this cell type atlas is an impressive resource, with a tremendously high cellular and molecular resolution,**
 **obtained through deep scRNA-seq and spatial transcriptomics, the ancestral reconstruction they obtain of the**
 **vertebrate brain does not significantly change or improve our previous views, obtained through decades of**
 **comparative analysis of lamprey and hagfish brains with jawed vertebrates. Their main find of a large set of**
 **homologous cell type families between the lamprey and the mouse, implying their presence in the last common**
 **ancestor of vertebrate is not surprising and largely expected, given the great conservation of general regionalization**
 **and gene expression patterns between lamprey, hagfish and gnathostome brains reported through countless past**
 **studies. As such, most of the conclusions they reach through the manuscript are mostly confirmatory. In fact, a large**
 **amount of the cell types described have been previously reported and just confirmed here (e.g.,**
 **inhibitory reticulospinal neurons: ref. 34 in the article; CSF-c neurons: ref. 38 in the article; habenular nuclei in the**
 **epithalamus: ref. 40; etcetera). See also Villar-Cerviño et al. (2011; <https://doi.org/10.1002/cne.22597>) and Pombal**
 **et al. (2011; <https://doi.org/10.3389/fnana.2011.00020>), just to mention a couple of extra examples not cited but**
 **familiar to the authors.**

We thank the referee for the positive appreciation of our resource and its high cellular and molecular resolution. We
 are also grateful to the referee for various points raised below, which helped us to clarify and improve several aspects
 of our work and its presentation in the manuscript, as detailed below each of the referee’s comments.

While our observation of many homologous cell type families between lamprey and mouse and hence their inferred
 ancestral status may have been largely expected based on previously observed regionalization conservation and
 expression studies of individual genes, it was and could not be made before because of the lack of suitable data. The
 value of our study, which is based on unique large-scale single-cell and spatial transcriptomic data (covering all cell types
 and marker genes), therefore is that it provides an unprecedented and detailed overview of the cellular and molecular
 organization across the entire lamprey brain. Our data thus afford a detailed comparison of a jawless vertebrate with
 corresponding patterns in jawed vertebrates, in particular the comprehensive mouse data (Zeisel et al. *Cell* 2018), which
 allow us for the first time to shed rather detailed light on the cellular and molecular characteristics of the brain of the
 last common vertebrate ancestor.

In addition to various novel observations, we thus indeed confirm in a global and unbiased way – across cell types and
 brain regions – various interesting previous findings, hypotheses, and speculations, which were based on limited
 numbers of marker genes, using our large-scale and high-resolution datasets and available gnathostome data. We had
 done our best to cite the relevant papers, but agree that we should also have cited the Villar-Cerviño et al. and Pombal
 et al. papers – these are now cited in appropriate places in the manuscript (P11, L349; P12, L384; P14, L459; P15, L469
 and 484).

Moreover, we also refine and extend previous findings and hypotheses, and our data allow us to resolve previous
 competing notions and inferences. We would like to note in this context that our new analyses, motivated by comments
 from referee #2, provide several additional novel and exciting insights regarding subregions of the lamprey
 telencephalon and their homologous relationships with those of jawed vertebrates, which also allows us to favor some

previous hypotheses over others. We thus obtain new findings regarding the striatum (i.e., we identify a lamprey cell
 type that is homologous to medium spiny neurons, find that the lamprey MPO corresponds to the dorsal striatum of
 jawed vertebrates, and that the anterior part of the traditionally denominated lamprey “striatum” corresponds to the
 gnathostomes sub-pallial amygdala), lamprey “medial” pallium (i.e., we confirm the hypothesis by Pombal et al. that
 this region is actually a rostral enlargement of the prethalamic eminence; our data indicate the presence of a pallial
 extended amygdala in lamprey), and lateral pallium (e.g., we find that only the evaginated (lateral) portion of the
 lamprey telencephalon should be considered a *bona fide* pallium that is homologous to all subdivisions of the pallium
 of amniotes/tetrapods). For details, please refer to our responses to reviewer #2 below (L382-601 of this response
 document).

 We have added these new findings to the manuscript and overall sought to improve the presentation of our results and
 insights, also by adding new figures (e.g., the new overview figure Fig. 6, which was stimulated by a comment from
 referee #3).

 **Other confirmatory instances are the lack of myelin from the lamprey PNS and the hypothetical origin of**
 **oligodendrocytes from astrocyte-like glia in the gnathostome lineage (refs. 26, 27 in the ms).**

 Besides the value of robustly confirming previous observations and hypotheses that were based on electron microscopy
 and/or a few marker genes (refs 27, 28 and 34 in the revised manuscript) using our extensive data, our work provides
 substantial new data and insights. With respect to the lamprey PNS, we identify glial cells at the cranial nerve roots that
 likely correspond to the ensheathing glia described by Weil et al. (ref. 27) (Extended Data Fig. 5a, e). We thus confirm
 the presence of these cells, identify additional marker genes whose orthologs are expressed in satellite glia and Schwann
 cells in the mouse, and also illustrate for the first time the absence of expression of several typical peripheral myelin
 genes known from mammals. Moreover, importantly, our evolutionary tree analyses based on the comprehensive
 whole transcriptome information show that lamprey PNS glia co-cluster with satellite glia and Schwann cells (Fig. 2c),
 strikingly reflecting the homologous relationships of these cell types. Thus, our PNS-related work both strongly confirms
 and extends previous work and hypotheses by Weil et al. We have now optimized the wording in the paragraph, to
 better reflect our findings and conclusions:

 (P6, L200-211) “PNS glia are represented by a small cluster (n = 53) expressing the orthologs of the mouse TF genes
 *Sox10* and *Sox9* (denominated *SOXE2* and *SOXE3* in lamprey, respectively²⁶); they co-localize with cranial nerve roots
 (Extended Data Fig. 5a, e). This group of cells, which most likely corresponds to the previously described peripheral
 ensheathing glia²⁷, expresses some markers whose mouse orthologs are characteristic of satellite glia (*SOXE2*) and
 Schwann cells and their precursors (*EGR2/3/4*, *PMP22/EMP3*) (Extended Data Fig. 5a). However, they lack the
 expression of key peripheral myelin constituent genes like *MPZ* and *PMP2*, confirming the absence of actual myelin from
 the lamprey PNS²⁸. Together with the co-clustering of this cell type with mouse satellite glia and Schwann cells (Fig. 2c),
 our observations strongly support and extend the hypothesis that lamprey PNS ensheathing glia are homologous to
 mammalian Schwann cells/precursors. The co-localization of this cell type with meningeal fibroblasts and vascular
 smooth muscle cells in the cell type tree (Extended Data Fig. 5a) likely reflects their common developmental origin from
 the neural crest¹⁸.”

 We have now also added a summary of this part in the Discussion section:

 (P14, L443-446) “While our data also confirm the absence of actual myelin from the lamprey PNS, our study lends strong
 support to the previous hypothesis³ that lamprey PNS glia are homologous to both mammalian Schwann and satellite
 cells.”

 With respect to the CNS, our work for the first time demonstrates that there are two large families of lamprey
 ependymogial cells (section: “Ependymogial cells and the origin of myelination”; Fig.2, Extended Data Fig. 6): one that
 is likely homologous to mammalian astrocytes and the other that is homologous to mammalian ependymal cells, a
 conclusion that is based on unbiased and comprehensive transcriptome-based comparisons of cell type gene expression
 profiles between the mouse and lamprey atlases (Fig. 2). Moreover, we then show for the first time, that adult lamprey
 astrocytes are enriched for key marker genes that are expressed by oligodendrocytes in jawed vertebrates. Altogether,
 our study thus provides unprecedented support for the hypothesis that oligodendrocytes arose during evolution from
 astrocyte-like cells in the common gnathostome ancestor. We have optimized the text in this section:

(P7, L231-236) "Notably, lamprey astrocytes are highly comparable to those from mouse in terms of their overall
 transcriptome signature (Fig. 2c, e). They share key marker genes that are fundamental for the development and
 function of astrocytes, such as *SOXE3* (*Sox9*), *HES5*, and *SLC1A2* (Fig. 3a; Extended Data Fig. 6a). However, like in other
 anamniotes (e.g., fishes, amphibians), lamprey astrocytes are mainly localized around the ventricles (Fig.3; Extended
 Data Fig. 6h), forming the so-called ependymo-radial glia³³.

 Like in the PNS, lamprey CNS axons are not myelinated²⁸, consistent with the absence of key master regulators of
 oligodendrocyte identity (*OLIG1*, *OPALIN*) and myelin specific genes (*MOBP*, *TSPAN2*) from its genome. Other myelin-
 related genes are present in the genome, but they are not expressed in glial cells (e.g., *PDGFRA* and *NKX6-1/2* are
 expressed in meningeal fibroblasts; Extended Data Fig. 5a). Notably, despite the lack of myelination, lamprey astrocytes
 express several oligodendrocyte-specific genes, such as the TFs *NKX2-2* and *SOXE2* (*Sox10*)³⁴ (Fig. 3b; Extended Data Fig.
 6a), the proteolipid gene *PLP1/GPM6B* (orthologous to the myelin components *Plp1* and *Gpm6b*) (Fig. 3b; Extended
 Data Fig. 6a), and the extracellular matrix glycoproteins *TNR* and *HEPACAM* (Extended Data Fig. 6a, j, k). Given the
 expression of crucial TFs of oligodendrocyte identity and the presence of myelin-related genes within lamprey
 astrocytes, our findings lend strong support to the hypothesis that oligodendrocytes originated from astrocyte-like glia
 in gnathostome ancestors²⁷."

 We now extended and refined the corresponding section in the Discussion, to better highlight these findings:

(P14, L432-446) "Our analyses of non-neuronal cells revealed the presence of two distinct cell types within the lamprey
 ependymoglia that are likely homologous to ependymal cells and astrocytes of gnathostomes (Fig.1c), suggesting that
 two of the three main macroglial cell types (astrocytes, ependyma, oligodendrocytes) were already established in the
 common vertebrate ancestor. Notably, however, our work confirms the absence of oligodendrocytes and sheds new
 light on their origination. We found that lamprey astrocytes express several oligodendrocyte-specific genes, including
 master regulators and effector genes (Fig. 3b). Our observations suggest that key components of the molecular
 machinery of oligodendrocytes were present in astrocyte-like cells of the vertebrate ancestor, and indicate that
 oligodendrocytes originated from these evolutionary precursors on the gnathostome lineage (Fig. 6a). Our work thus
 extends previous studies, which showed that lamprey axons seem to be physically associated with astrocytes²⁷ and that
 key aspects of the regulatory program required for oligodendrocyte differentiation in gnathostomes are present during
 lamprey gliogenesis³⁴."

 I also have some technical concerns about the differences found between larval and adult brains. One of the earliest
 findings in the article is that "the adult brain is characterized by larger numbers of expressed genes per cell and
 greater cell type specificities of gene expression compared to the ammocoete brain". However, the larval and adult
 brain data are obtained using different versions of the kit, v2 and v3, respectively. The authors try to control for
 technical differences by sequencing a whole larval brain with a v3 kit (library SN352). While the authors conclude that
 data obtained from this library confirm the putative biological differences observed, how is this exactly concluded?
 The bar plot in Extended Data Fig. 4b shows that they find what seems a significantly larger number of expressed
 genes per cell in SN352 (v3) than in the v2 data. Also, Extended Data Fig. 4c shows that SN352 data is in an
 intermediate position between larval v2 data and adult v3 data.
 Without statistical testing is difficult to say that they can rule out technical variation between v2 and v3 kits. This
 considered, their sentence "This result is robust to controls for technical differences between the datasets" (Lines 93-
 94) seems an exaggerated claim when lacking statistical support. I strongly recommend, at least, appropriate
 statistical testing, and, optimally, sequencing more larval samples with v3 kits to make the different samples
 comparable to properly confirm that biological causes are behind the differences observed.

We thank the referee for this valid and useful comment, which prompted us to compare the adult and ammocoete
 brains in more detail using additional data that we generated during the revision period.

Specifically, we generated scRNA-seq data for whole brain samples from three additional ammocoetes (SN580, SN582,
 SN588; Methods: P19, L647-649; Supplementary Table 1 and 2; total of 22,950 cells). In parallel, we also found out
 (based on assessments of data also from other lab projects from that time) that during the transition phase from v2 to
 v3 kits, we had received a few suboptimal v3 kit batches, one of which was used to generate the data for the previous
 SN352 sample, which we hence discarded from the analyses.

Analyses of the new v3 data show that while the number of expressed genes is similar between adult and larval cells
 (Extended Data Fig. 4b), gene expression patterns remain overall significantly more cell type-specific in adults compared

to ammocoetes ($P < 0.001$; Mann-Whitney U test; **Extended Data Fig. 4c**). This observation was confirmed in analyses in
 which we downsampled all datasets to both the same number of cells and scRNA-seq reads, which rules out that the
 result is due to technical differences (i.e., amounts of data) between samples (**Extended Data Fig. 4d**).

However, we agree with the reviewer (see also further below) that our data (including the high abundance of
 periventricular ZFP704 marked neurons in larvae) do not directly imply that a larger proportion of neurons in larvae are
 undifferentiated. We therefore removed this notion from the discussion section and condensed/toned down the
 paragraph describing these observations in the beginning of the paper, which now reads:

**(P3, L90-93) “Overall, neural cell type compositions are similar between the two stages (Extended Data Figs. 2-4a).
 However, we note that there is a generally higher cell type specificity of gene expression patterns in adults compared
 to ammocoetes (Extended Data Fig. 4c) – a result that is robust to controls for technical differences between the
 datasets (Extended Data Fig. 4b-d).”**

In any event, the neural cell type atlases are generally very similar for adult lampreys and larvae and the conclusions
 made in our paper apply to both life stages. We just focus the manuscript more on the adult stage, given the more
 distinct cell type/gene expression data, the overall higher resolution of the adult scRNA-seq (all v3) and ISS data (e.g.,
 the larger number of marker genes), and the fact that adults may represent the life stage that is more relevant for
 evolutionary inferences of ancestral traits (Miyashita et al. *Nature* 2021; PMID: 33692547). We now introduce the latter
 notion in the beginning of our paper’s result section:

**(P3, L95-97) “A cell type tree derived from the datasets for the adult lamprey, which is thought to be better suited for
 the inference of ancestral vertebrate traits than ammocoetes¹, reflects cell type relationships based on gene expression
 distances (Fig. 1b, c).”**

**A bar plot showing the proportions of each cell types obtained from SN352 should also be included in Extended Data
 Fig. 2 together to those from SN110 and SN144 to test for equivalent recoveries of cell types using v2 and v3 kits.**

We have now added three new corresponding bar plots in **Extended Data Fig. 2b** for the new larval datasets (SN580,
 SN582, SN588) that were generated using v3 kits. The proportions of cell types recovered using the two kit versions are
 overall similar. In any event, we note that none of the conclusions in our work rest on cell type proportions.

**The proportions of each cell types look very different between different libraries of the same brain region in the adult
 (Extended Data Fig. 3). I’m no expert in scRNA-seq, so is this a technical issue? I might be wrong, but I would expect
 that different libraries of the same brain region would count as sort of biological replicates, and thus certain
 correlation would be expected. If this is not observed, are the data still reliable? Is there a statistical way to measure
 the correlation and/or the variation in the data from different libraries? This is not the case in the larval brain, what
 makes me think there might be a problem with the adult samples/seq processing/data, but I might be wrong.
 Given all of the above, I am not sure if the hypothesis about the undifferentiated nature of the larval brain holds true
 or not.**

The reason for the more uneven cell type proportions between replicate samples from adults compared to ammocoetes
 is that it is substantially more challenging and time consuming to dissect adult brains, given the much more rigid
 structure of the skull of adult lampreys and greater difficulty in removing the more pronounced meninges and choroid
 plexuses. Because scRNA-seq is time-sensitive (i.e., cell death/RNA degradation needs to be avoided/limited) brains
 have to be dissected in a timely manner. Thus, a larger (variable) contribution of meninges/choroid plexus cells and
 partly less precise dissection of individual brain regions compared to dissection of larvae brains is inevitable. We note
 that when (rapidly) dissecting individual brain regions, the diencephalon is particularly challenging; that is, fully avoiding
 telencephalic cells in samples of this brain region is difficult.

Notably, new gene expression correlation analyses between all samples for the different adult brain regions – as
 suggested by the referee – show high (the highest) correlation coefficients (median Spearman’s ρ : 0.97 for replicate
 samples for the same brain regions, respectively (**Extended Data Fig. 2c**). The correlation coefficients for replicates are
 similar in a corresponding analysis for ammocoetes (median Spearman’s ρ : 0.90) (**Extended Data Fig. 3c**). These
 observations support that our data is of high quality and that datasets for biological replicates are very similar also in
 the adult, in spite of the aforementioned dissection challenges.

We furthermore note that when removing the meninges/choroid plexus cells in the analyses from the whole adult brain
 samples (new **Extended Data Fig. 3b**), the proportions of neural cell types – the focus of our study – are similar between
 samples (approximately as similar as between ammocoete samples), which supports that our scRNA-seq procedure
 captures the different cell types in a reproducible way.

 It is also important to note that when combining the data from the different brain regions, telencephalic cells captured
 in the diencephalon sample are automatically and correctly placed in the corresponding telencephalon cluster based on
 their transcriptome signatures in the overall cell type atlases and corresponding UMAPs.

 In any event, we would like to again emphasize that none of the conclusions in our work rest on cell type proportions.

 **Continuing on Extended Data Figs. 2 and 3, are the UMAP projections a mix of all libraries of each brain region, or do
 they come from particular libraries of each of the regions? Please indicate.**

 The UMAP projections on **Extended Data Figs. 2 and 3** indeed represent integrations of the data for the biological
 replicates for each brain region. We have now added this information to the corresponding figure legends (**P32, L1103-
 1104; P34, L1112-1113**). We apologize for not having made this sufficiently clear in the original manuscript.

 **Again, I'm no expert on scRNA-seq, so it surprises me that the adult whole brain UMAP projection in Extended Data
 Fig. 3a/4a is quite different from that in main Fig. 1b.**

 The two UMAP projections in the two figures are different because one (**Fig. 1b**) reports an integration of the data from
 all replicates (all brain regions and whole brain) into a single UMAP projection, whereas the others (**Extended Data Fig.
 3a/4a**) represent integrations of biological replicates for each brain region, respectively (i.e., one UMAP per brain region,
 as indicated by the labels). We have optimized the respective legends to make this clearer.

 **The authors conclude in their first section that "The hierarchical relationship of cell types in the lamprey brain is very
 similar to that observed for a reference mammalian brain atlas (i.e., that of the mouse), which suggests that all
 vertebrates share common general cellular and molecular organization of neural tissues that was established during
 the evolution of the vertebrate stem lineage." However, as I mention before, this is expected, given that we already
 know, from all the previous literature published, that the anatomical, molecular and even cellular organization of
 both cyclostome and gnathostome brains are greatly conserved.**

 This general observation may have indeed been expected based on previous work, but was never actually made before.
 The value of our study, which is based on unique large-scale single-cell (spatial) transcriptomic data (covering all cell
 types and marker genes), thus is that it provides an unprecedented detailed overview of the cellular and molecular
 organization across the entire lamprey brain, affording for the first time a detailed comparison of a jawless vertebrate
 with corresponding patterns in jawed vertebrates, in particular the comprehensive mouse data (Zeisel et al. *Cell* 2018).
 Indeed, similarly, the value of the mouse work by Zeisel et al. was that it provided the first such overview of the cellular
 relationships for a jawed vertebrate brain, even though the observed pattern itself might not have been surprising at
 the time.

 **- Some minor issues:**

**1) Explain what 'scRNA-seq' stands for the first time is mentioned**

Thanks for pointing this out. We now define scRNA-seq upon first mention:

(**P2, L75-76**) **"We generated single-cell RNA-sequencing (scRNA-seq) data (21 libraries in total) for whole adult and
 ammocoete brains, and separately for their four major anatomical regions..."**.

**2) L57: add Albuixech-Crespo et al. (2017; <https://doi.org/10.1371/journal.pbio.2001573>) after the sentence about
 the brain molecular architecture of invertebrate chordates, together with references 3,4.**

We have added the suggested reference (**P2, L57**).

**3) L176 and Extended Data Fig. 5 (L1025): change 'hematopoietic cells' for 'blood cells'. 'Poiesis, derived from ancient**
**Greek, means the 'creation of'. Hematopoietic is thus used in the case of stem and progenitor cells, which create all**
**blood cell types. Unless you are exclusively describing those stem cells, then 'blood cells' is a more appropriate term.**

We agree and have implemented the suggested change of terminology in all instance in the manuscript (P3, L100; P6,
L175; P36, L1128 – legend of Extended Data Fig. 5).

**4) There are no HOXA/B3, HOXA/B4 and HOXA/B5 genes in the lamprey (Mehta et al.,**
**2013: <https://doi.org/10.1073/pnas.1315760110>; Smith et al., 2018: [https://doi.org/10.1038/s41588-017-0036-](https://doi.org/10.1038/s41588-017-0036-1)**
**1; Parker et al., 2019: <https://doi.org/10.1016/j.ydbio.2019.05.001>).**

**Some of the authors have a large experience working with the lamprey, including Hox genes. Please, name the genes**
**properly.**

Given that most genes are not specifically defined/named yet in lamprey and that brain marker genes are best
established for the mouse, which is generally used as a major mammalian reference in our study also due to the
comprehensive available cell atlas, we chose to systematically use the name of corresponding mouse ortholog(s) as a
reference for each of the reported orthologous lamprey genes in the study (with the exception of *SOX* genes, where we
used the well-established cyclostome annotation (*SOXA*, *SOXB*, ...) and lamprey reference work; P6, L200-201).
Importantly, this also avoids confusion when comparing orthologous genes and their expression between species; that
is, it would otherwise be difficult to understand homology relationships in our genome-wide study. In the case of *HOX*
genes, some of the lamprey gene models correspond to two *Hox* genes in mouse – the name of the two orthologs is
separated by a slash (“/”). We now clearly explain the naming convention in the Methods section:

(P19, L617-623) "Throughout this work, we use mouse ortholog names to indicate lamprey gene names. This choice is
justified by the fact that most lamprey genes lack a clear and consistent nomenclature, and that mouse is used as the
main reference in our study. In cases where multiple mouse genes correspond to one lamprey gene (one-to-many
relationships) we append all ortholog names, separated by slashes. We made an exception for *SOX*-genes, where we
used the well-established cyclostome annotation (*SOXA*, *SOXB*, ...) and lamprey reference work."

We refer to this explanation in the main text upon the first mention of gene names:

(P3, L116-119) "...but by *GATA2/3*, *OTX2* and *TAL* genes in the posterior forebrain, midbrain, hindbrain and spinal
cord^{16,17} (Fig. 1c; Extended Data Fig. 7a; gene names are based on respective names of the mouse ortholog(s) – see
Methods for details regarding the gene nomenclature used in this study)."

**Another instance is that sometimes the authors mention OTX1/2 (Line 280) but elsewhere OTX2. Upon checking the**
**corresponding transcript (MSTRG.8275), it appears to be the old lamprey gene OtxA, which is Otx2 proper based on**
**synteny conservation (Yamamoto et al., 2020: <https://doi.org/10.1016/j.celrep.2019.12.072>). I strongly suggest the**
**authors to add an extra column in Supplementary Tables S4 and/or S8 with a reference to the first report (if any) of**
**those genes that have been previously identified and described in the lamprey.**

We thank the referee for pointing out this mistake. We have now replaced the instance where we inadvertently referred
to "*OTX1/2*" with "*OTX2*" (P9, L282).

While it is possible for a few selected well-defined/studied cases to assign previous gene names to genes in our study
(see comment/response above), it would be extremely difficult, unreasonably time-consuming and partly impossible to
systematically do so for all assigned names from the various previous studies. Contrary to our unbiased approach, which
is based on a systematic genome-wide inferences/comparisons and orthology assignments, previous work was limited
in various way when assigning gene names. For example, previous studies used the closest known ortholog known at
the time as a basis. To assess correspondences with our data, we would need to extract sequences from all of these
studies – if at all available (see also hereafter) – and for example BLAST them against our gene database, which may
then also uncover misassignments and even disagreements between studies due to the previously limited data and
various many-to-many orthologous gene relationships etc. Another problem would be that many previous studies were
based on immunohistochemistry (i.e., antibody-based approaches) to detect genes in lamprey – hence no sequences
are available for potential comparisons with our whole genome annotations.

5) Following on gene nomenclature, lamprey literature does not tend to write, with exceptions, lamprey gene symbols in capital letters (a writing style that I think is usually restricted to human genes). Although the literature contains examples of all writing styles for the lamprey, using a fish nomenclature style is strongly recommended (see Zebrafish Nomenclature Conventions): thus, lamprey gene symbols should be written completely in lowercase and in italics. If you agree, change this throughout the manuscript, figures and supplementary files and data.

Given the differences in the lamprey literature and major databases and that gene names also in fish species are written using different conventions, including all capitalized names (e.g., for the coelacanth genome in Ensembl), we prefer to not change the nomenclature in the manuscript, which is consistent with at least part of the literature/databases (e.g., the NCBI capitalizes all gene name letters in the lamprey genome, as done in our study).

6) The authors should cite the articles that first provided the gene sets they used for orthology assignment. Ensembl is just the hosting genome browser of these data (*Ciona intestinalis*⁸², inshore hagfish (*Eptatretus burgeri*, *Callorhynchus milii*, *Lepisosteus oculatus*, *Danio rerio*, *Latimeria chalumnae*, *Xenopus tropicalis*, *Gallus gallus*), but not the ones generating the data (except for mouse and human, for which they do their own annotation I think). Accordingly, since you use Ensembl, the authors should also cite it appropriately.

We thank the referee for the useful suggestion that we have implemented accordingly:

(P18, L601-608) "Orthology assignment and gene nomenclature Homology information for the set of annotated genes was retrieved by applying the OrthoFinder⁸¹ (v2.3.11) pipeline against a group of selected chordates: vase tunicate (*Ciona intestinalis*)⁸², inshore hagfish (*Eptatretus burgeri*; permission to use unpublished genome data was given exclusively for the purposes of the present study; personal communication), Australian ghostshark (*Callorhynchus milii*)⁸³, spotted gar (*Lepisosteus oculatus*)⁸⁴, zebrafish (*Danio rerio*)⁸⁵, West Indian Ocean coelacanth (*Latimeria chalumnae*)⁸⁶, Western clawed frog (*Xenopus tropicalis*)⁸⁷, red junglefowl (*Gallus gallus*)⁸⁸, house mouse (*Mus musculus*), and human (*Homo sapiens*)."

7) Reference section: Species names in the titles of literature cited needs to be in italics. E.g., *Petromyzon marinus* in references 37, 42, 49, 50. Same goes for gene names. For instance, in reference 15, *Gata2* should go in italics, as per the original title (<https://doi.org/10.1242/dev.029900>). I haven't checked all the literature thoroughly, so please, double check. The authors should be especially careful about correctly spelling the names of authors cited.

Thanks for spotting this. We have corrected these instances in the titles of the cited literature.

Referee #2 (Remarks to the Author):

In this manuscript, Lamanna, Hervas-Sotomayor and colleagues describe a cell type atlas of the lamprey brain. In their study, they identify 151 and 120 types of neurons in lamprey adults and larvae, respectively. The spatial localization of many of these types is identified by in situ sequencing.

This is a paper that comparative neurobiologists have been waiting for a very long time. As early-branching, jawless vertebrates, lampreys occupy a key phylogenetic position to infer the early evolution of the vertebrate brain. The lamprey brain has puzzled neuroanatomists for decades. The existence of a cerebellum in lampreys is still unclear. The identity of subregions in the lamprey telencephalon is also unclear. For example, the telencephalic area that the authors refer to as "medial pallium" has been considered by others a part of the diencephalon (the prethalamic eminence).

An important and thought-provoking finding is the expression of oligodendrocyte markers in ependymoglia cells. This leads to the interesting idea that oligodendrocytes and ependymoglia cells might have evolved from multifunctional glial cells. The discovery of broad molecular similarities between what the authors call "neuron type families" is also a nice result but it is less surprising, given that the homology of broad brain regions such as the olfactory bulb or the hypothalamus is well settled. The reader is left wondering about what makes the brain of a lamprey different from the brain of a mouse at the cell type level.

**My biggest critique to this manuscript is that it shies away from providing the answers that readers are looking for.**
 **Part of the problem is that the dataset is so large, that it is impossible to describe all of it in detail. Furthermore - if I**
 **may - the authors seem too concerned about following the neuroanatomy tradition instead of listening to their own**
 **beautiful data. In the last decade, molecular studies on brain evolution have shattered decades of neuroanatomy**
 **dogma, providing surprising re-interpretations of entire brain regions (an example that comes to mind is the discovery**
 **of a telencephalon in amphioxus by co-author Detlev Arendt and colleagues). Here I could find at least three examples**
 **(there are probably more...) where the molecular data described in this manuscript are in contrast with dominant**
 **interpretations of the lamprey brain:**

 We thank the referee for the enthusiasm and overall appreciation of our manuscript. We are especially grateful for the
 very useful suggestions and thoughts that motivated and inspired us to obtain novel results from our data (indeed that
 dataset is large, and it is challenging to adequately describe all interesting facets). That is, the new analyses prompted
 by the referee's notions provide several additional interesting insights into the structures and cell types of the lamprey
 telencephalon and their relationships with those from jawed vertebrates, as detailed below.

 **1. Striatum. The term striatum is used inconsistently in the mammalian literature; most people equate the term**
 **striatum to the dorsal striatum (caudoputamen in mouse), but a less common usage includes other subpallial areas,**
 **such as the "striatum-like amygdalar nuclei" (see Allen Brain Reference Atlas). The authors should qualify this term**
 **precisely before using it.**

**The authors follow the work of previous scholars and identify as "striatum" a region of the lamprey subpallium nested**
 **between the preoptic area and the pallium. This region corresponds to cluster Telnh5, and expresses genes like**
 **lhx6/8, galanin, gbx1, and nkx2-1. From this, I came to wonder whether the lamprey "striatum" corresponds partially**
 **or entirely to the subpallial amygdala. In amniotes, lhx6, lhx8, galanin, and nkx2-1 are not markers of differentiated**
 **medium spiny neurons (MSNs), and gbx1 is expressed in MSNs at very low levels, if at all (see mousebrain.org). Typical**
 **MSN markers, such as Sp8/9 and Isl1, are not expressed in Telnh5. Lhx6, lhx8, galanin, and nkx2-1, instead, are**
 **expressed prominently in the subpallial portion of the extended amygdala (bed nucleus of the stria terminalis, central**
 **amygdala, medial amygdala, see for example Garcia-Lopez et al Journal of Comparative Neuroscience 2008 PMID:**
 **17990271, Choi et al Neuron 2005 PMID: 15944132, and many others).**
 **Topologically, the idea that the dorsal striatum is immediately adjacent to the preoptic area is inconsistent with data**
 **from tetrapods. The tetrapod dorsal striatum is adjacent medially to the pallidum, dorsally to the ventral pallidum,**
 **and caudally to the amygdala. Therefore, there is no boundary between the preoptic region and the dorsal striatum;**
 **instead, there is a boundary between the preoptic area and subpallial amygdala. Back to figure 4d, one could**
 **speculate that at least the medial lhx6/8-expressing portion of the "striatum" is subpallial amygdala.**
 **This interpretation may seem in conflict with the results of Grillner and colleagues on the connectivity of the lamprey**
 **basal ganglia, proposing the homology of lamprey and mammalian dorsal striatum. However, Swanson (figure 5**
 **PMID: 12724158) pointed out that the circuits connecting parts of the cortex, amygdala, and brainstem motor**
 **systems are organized exactly like the dorsal striatum circuits compared by Grillner. One could speculate that these**
 **systems are evolutionarily related as serial homologs (reviewed in Loonen and Ivanova 2016 PMID: 27920666). The**
 **real question that remains still unaddressed is whether lampreys do have homologs of dorsal striatum medium spiny**
 **neurons.**

 We are grateful to the reviewer for these interesting and useful notions, which prompted us to reanalyze our data and
 obtain interesting novel insights, in line with the referee's thoughts and speculations.

 Specifically, when reanalyzing our gene expression data, we noticed that the cell type Telnh4, which corresponds to
 LGE-derived neurons of the medial preoptic nucleus (MPO), is characterized by the high and specific expression of the
 *TAC1* gene (substance P). Together with the expression of orthologs of other important marker genes (*ISL1/2*, *SP8/9*) in
 the Telnh4 cell type, this observation suggests that this cell type is homologous to medium spiny neurons of the striatum
 of jawed vertebrates (i.e., our data do suggest that lampreys have homologs of dorsal striatum spiny neurons), which in
 turn suggests that the lamprey MPO corresponds to the dorsal striatum of jawed vertebrates (Fig. 4 and Fig.6). We now
 summarize these observations in the manuscript:

 (P10-11, L338-354) "Inhibitory neurons of the telencephalon are classified into olfactory bulb (OB) and pallium/sub-
 pallium cell types and are all enriched for typical forebrain GABAergic markers (*GAD1/2*, *DLX1a*, *DLX1b*, *DLX2/3/5*;
 Extended Data Fig. 7l, m; Extended Data Fig. 8a). OB neurons can be recognized by: i) the conserved expression of several
 TFs that are characteristic of the anterior forebrain and placodes in chordates³ (e.g., *SP8/9*, *PAX6*, *FOXP1*, *ETV1*;
 Extended Data Fig. 8a, s), ii) the unique expression of *PRDM12* (expressed in pain-sensing nerve cells and V1

interneurons in gnathostomes^{54,55}; Extended Data Fig. 8a, t), and iii) the presence of dopaminergic cells (type: TeDopa1; Extended Data Fig. 8a).

*SP8/9** neurons are present also in the sub-pallium (type: Telnh4), within a region traditionally considered to correspond to the medial preoptic nucleus⁵⁶ (MPO; Fig. 4d), where they co-express *ISL1/2* and *TAC1*, both markers of striatal projection neurons in gnathostomes (Extended Data Fig. 8a, Fig. 4g). The presence of *SP8/9*-ISL1/2** and *SP8/9*-ETV1** neurons in the sub-pallium and OB, respectively, is already known for mammals, where they originate from the lateral ganglionic eminence (LGE)⁵⁷, suggesting that these two cell populations share the same developmental origin and migratory patterns across vertebrates.”

In light of this new result and reconsiderations of our data stimulated by the reviewer, we now provide a new interpretation to the region that was traditionally called “striatum” in lamprey and is populated by MGE-derived neurons (Telnh5). Consistent with the referee’s notion, we now propose that the anterior region of this area, located dorsal to what we could now define as the lamprey striatum (previously termed MPO; Fig. 4d, g) (see response above), corresponds to the sub-pallial amygdala of jawed vertebrates based on the expression of many shared marker gene (Fig. 4d, g and 6b).

(P11, L356-362) “Another important sub-pallial progenitor zone in jawed vertebrates is the medial ganglionic eminence (MGE). We identified neurons (type: Telnh5) expressing *LHX6/8* and *NKX2-1/4a* (both markers of MGE-derived cells in mammals) (Fig. 4g) within two sub-pallial regions: i) dorsal to the MPO (Fig. 4d), in a region traditionally called “striatum”⁵⁸, and ii) the putative pallidum⁵⁹, a nucleus located ventrolateral to the thalamic eminences (Fig. 4e). Recursive clustering revealed the presence of subtypes that express markers that are typical of MGE-derived neurons of the sub-pallial amygdala and pallidum in jawed vertebrates⁶⁰ (e.g., *TACR1*, *GBX1*, *SOX6*; Fig. 4g).”

We now also summarize and put into context these findings in the Discussion section:

(P14-15, L455-472) “The discovery of both LGE- and MGE-derived inhibitory neurons in the lamprey telencephalon confirms that the two main GABAergic progenitor zones of the sub-pallium were already present in the common vertebrate ancestor^{6,70} (Figs. 4f and 6a). Our findings challenge the traditional neuroanatomy view regarding the localization of the main sub-pallial regions of lampreys. Previous studies^{2,56} used to locate the striatum dorsal to the MPO and ventrolateral to the pallium (Fig. 4d). In this study, however, we identified a group of LGE-derived neurons (type: Telnh4), located in the MPO, that express the genes *ISL1/2*, *TAC1* and *PENK* (Fig. 4g), whose orthologs are typical markers of projection neurons of the dorsal striatum in jawed-vertebrates (medium spiny neurons; MSNs). This evidence indicates that the MPO of lampreys is in fact homologous to the dorsal striatum of jawed-vertebrates and that it should be renamed accordingly (Fig. 6b). The region traditionally considered to correspond to the striatum, on the other hand, is populated by MGE-derived cells (type: Telnh5) that express the markers *LHX6/8*, *GAL*, *GBX1* and *SOX6* (Fig. 4c, d, g), whose orthologs are expressed in the same combination in the sub-pallial amygdala (SPA) of jawed vertebrates. We therefore propose that this region corresponds to the SPA and not to the striatum, as previously believed^{2,56} (Fig. 6b). MGE-derived cells can also be found caudal to the SPA where they form the pallidum (Figs. 4e and 6c). Outside the lamprey sub-pallium, LGE- and MGE-derived cells also contribute to GABAergic interneurons of the OB and pallium, indicating that their migratory patterns are conserved across vertebrates (Fig. 6a).”

2. Medial pallium. Pombal, Puelles and colleagues (PMID: 19729892) proposed (following previous literature) that the so-called medial pallium is a rostral enlargement of the lamprey prethalamic eminence. Strikingly, the authors show expression of *ebf1* in this area (cluster TeExc4, Figure 4g) and also in the DMTN. *Ebf* genes are not detected in the medial pallium/hippocampus of tetrapods. Instead, *Ebf* genes, together with *Lhx1/5* and *Tbr1*, are markers of the prethalamic eminence of tetrapods. These observations call for a more rigorous assessment of the diencephalic nature of the medial pallium.

Thanks for this very useful comment, which stimulated a reassessment of the “medial pallium” based on our data. Our data regarding the dorsal telencephalon indeed confirms the hypothesis by Pombal et al. that the region previously denoted “medial pallium” in lamprey is actually a rostral enlargement of the prethalamic eminence (i.e., it is diencephalic) – as suggested by the referee. This notion is supported by the expression of genes that are typically expressed in pre-thalamic excitatory neurons in jawed-vertebrates (*EBF1*, *EBF2/3*) in the corresponding lamprey cell type (TeExc4) located in the previously denoted “medial pallium” (Fig. 5b-d, f).

Moreover, our data also indicate the presence of a pallial extended amygdala (PEA) in lamprey and show that it is located
dorsal to the sub-pallial amygdala. This region is populated by cells (type: TeExc3) that express markers of the extended
amygdala in mouse^{18,71} (*LMO3*, *PNOC*) (Figs. 5b-d, f and 6b).

We modified the relevant sentence in the results section:

(P12, L386-397) "We identified eight distinct cell types populating four different regions of the lamprey dorsal
telencephalon and anterior diencephalon: i) dorsomedial telencephalic nucleus (DMTN; type: TeExc1), ii) anterior pre-
thalamic eminence (a region previously believed to correspond to the "medial pallium"⁶³ – see also discussion below)
(PThE; type: TeExc4), iii) pallial extended amygdala (PEA; type: TeExc3), and iv) pallium (Pa; types: TeExc2, TeExc5-8)
(Fig. 5a, b, f; Supplementary Table 3; see online atlas). DMTN is a relay nucleus that is innervated by tufted-like cells of
the OB⁶⁴ and is located at the interface between the pallium and OB, of which it constitutes the caudal-most portion.
Like the OB, the DMTN displays a layered structure with outer glutamatergic neurons, which share the same expression
profile with cells of the OB glomerular layer, (e.g., *EBF1*) and inner GABAergic (*PRDM12*) neurons (Extended Data Fig.
8v, p, r-t, v, w). PThE and PEA neurons express the TFs *OTX2* and *NR2F1/6a* and are defined by the expression of *EBF1*,
*SSTc* (TeExc4) and *C1QL3*, *PNOC* (TeExc3) (Fig. 5b; Extended Data Fig. 8a, i, o, s)."

We now also provide a new paragraph regarding this topic in the Discussion section:

(P15, L474-482) "Our analysis of the dorsal telencephalon confirms the hypothesis that the region previously denoted
"medial pallium" in lamprey is actually a rostral enlargement of the prethalamic eminence⁶³ (i.e., it is part of the
diencephalon). This notion is supported by the expression of genes that are typically expressed in pre-thalamic
excitatory neurons in both lampreys and jawed-vertebrates (*EBF1*, *EBF2/3*) in the corresponding lamprey cell type
(TeExc4) located in the previously denoted "medial pallium" (Fig. 5b-d, f). Our data also indicate the presence of a pallial
extended amygdala (PEA) in lamprey, in support of a previous hypothesis⁶³, and show that it is located dorsal to the
sub-pallial amygdala. This region is populated by cells (type: TeExc3) that express markers of the extended amygdala in
mouse^{18,71} (*LMO3*, *PNOC*) (Figs. 5b-d, f and 6b)."

**3. Lateral pallium. The nature of the lamprey lateral pallium is strongly debated. Grillner and colleagues have**
**proposed its homology with the neocortex, on the basis of projections. Striedter and Northcutt (PMID: 34175847)**
**have recently argued that the dorsal pallium might be an amniote innovation! This would imply that the lamprey**
**lateral pallium is homologous to the ventrolateral pallium of amniotes.**
**This controversy could be solved by molecular data. At the very least, the hypothesis of Grillner and colleagues would**
**be supported if similar lamprey and mammalian projection types were to express the same transcription factors.**
**Extended figure 8 does not provide an exhaustive list, but it shows that *etv1/er81*, a marker of corticofugal**
**projections in amniotes (Dugas-Ford et al PMID: 23027930), is not expressed in lateral pallium glutamatergic neurons**
**(how about *fezf2*? *Sox5*? *Satb1/2*?). *Rorb* is uninformative, because it is expressed in both the dorsal and the ventral**
**pallia of amniotes.**

Thanks for this useful comment, which – together with the previous comment – allowed us to clarify the implications of
our data for the understanding of the nature of lamprey pallium and its homology to that of tetrapods.

Within the region previously denoted as "lateral pallium", we identified groups of cell types that are likely homologous
to glutamatergic mammalian cortical neurons, supporting the hypothesis that the core cell types composing
cortical/nuclear circuits across jawed vertebrates emerged in common vertebrate ancestors. These neurons express
genes that are associated with different projection modalities (e.g., input, intratelencephalic, output), but not in the
same combinations as observed in jawed vertebrates. Overall, together with the observations regarding the previously
denoted "medial pallium" (see previous comment), these results suggest that only the evaginated (lateral) portion of
the lamprey telencephalon should be considered a *bona fide* pallium that is homologous to all subdivisions (dorsal,
ventral, lateral, medial) of the pallium of amniotes (Fig.5; Fig. 6b, c). That is, we don't observe any regionalization of the
lamprey pallium based on our neuronal expression data. This suggests that at least in terms of neuronal cell type
expression identities, this regionalization emerged on the jawed vertebrate lineage leading to tetrapods. Indeed, recent
work by Woych et al. (*Science* 2022; PMID: 36048957) demonstrated the presence of four pallial regions in an amphibian
(i.e., a salamander), which is similar to amniotes and implies that such a regionalization was present at least already in
the common tetrapod ancestor, although the authors note that the amphibian dorsal cortex does not express many of
the markers that define the reptilian dorsal cortex (the area typically compared to the mammalian neocortex).

We modified the relevant paragraph in the Results and especially Discussion section according to these new insights:

(P12, L381-405) “The expression programs of excitatory neurons of the lamprey telencephalon are overall highly
 correlated to those of the corresponding cell types in mouse (Fig. 2f). This similarity is confirmed by the expression of
 marker genes typical of mammalian cortical glutamatergic neurons within the lamprey pallium⁶⁴ (previously denoted
 “lateral pallium”⁵⁶ – see also discussion below) and, partially, OB (e.g., *TBR1*, *EMX1/2a*, *EMX1/2b*, *RTN4R*, *LHX2/9*,
 *BCL11B*, *IKZF1/3*; Fig. 5b-d; Extended Data Fig. 7h, i; Extended Data Fig. 8a, g-i, k-l, q, v). We identified eight distinct cell
 types populating four different regions of the lamprey dorsal telencephalon and anterior diencephalon: i) dorsomedial
 telencephalic nucleus (DMTN; type: TeExc1), ii) anterior pre-thalamic eminence (a region previously believed to
 correspond to the “medial pallium”² – see also discussion below) (PThE; type: TeExc4), iii) pallial extended amygdala
 (PEA; type: TeExc3), and iv) pallium (Pal; types: TeExc2, TeExc5-8) (Fig. 5a, b, f; Supplementary Table 3; see online atlas).
 DMTN is a relay nucleus that is innervated by tufted-like cells of the OB⁶⁵ and is located at the interface between the
 pallium and OB, of which it constitutes the caudal-most portion. Like the OB, the DMTN displays a layered structure
 with outer glutamatergic neurons, which share the same expression profile with cells of the OB glomerular layer, (e.g.,
 *EBF1*) and inner GABAergic (*PRDM12*⁺) neurons (Extended Data Fig. 8v, p, r-t, v, w). PThE and PEA neurons express the
 TFs *OTX2* and *NR2F1/6a* and are defined by the expression of *EBF1*, *SSTc* (TeExc4) and *C1QL3*, *PNOO* (TeExc3) (Fig. 5b;
 Extended Data Fig. 8a, i, o, s). We found that pallial neurons form a three-layered cortex with an inner
 GABAergic/glutamatergic layer, a middle glutamatergic layer, and an external molecular, fiber-rich layer, in accord with
 previous work⁶⁶ (Fig. 5g). They all express multiple genes associated with cortical projection neurons in amniotes (e.g.,
 *FOXP1/2/4*, *MEIS2*, *LAMP5*, *RORB*, *TCAP*; Extended Data Fig. 8a). However, contrary to what is known for amniotes –
 and since recently – also for amphibians (i.e., for tetrapods in general)⁶⁷, we did not observe any regional specification
 of gene expression patterns among these neurons (e.g., dorsal, lateral or ventral) that could be related to known,
 functionally distinct areas of the pallium (e.g., somatosensory, visual, motor, olfactory), as previously observed based
 on connectivity data^{68,69}.”

(P15, L484-495) “Within the region previously denoted as “lateral pallium”^{2,56}, we identified groups of cell types that
 are likely homologous to glutamatergic mammalian cortical neurons, supporting the hypothesis that the core cell types
 composing cortical/nuclear circuits across jawed vertebrates emerged in common vertebrate ancestors^{69,73}. These
 neurons express genes that are associated with different projection modalities (e.g., input, intratelencephalic, output)
 (Fig. 5a, b), but not in the same combinations as observed in jawed vertebrates⁷⁴. Altogether, our observations indicate
 that only the evaginated (i.e., lateral) portion of the lamprey telencephalon should be considered a *bona fide* pallium
 (Fig. 6b, c), which – in terms of cell type expression signatures – is homologous to all subdivisions (dorsal, ventral, lateral,
 medial) of the pallium of tetrapods. This suggests that the regional specification of gene expression patterns among
 pallial neurons evolve during gnathostome evolution on the lineage leading to tetrapods^{67,75} – future work may
 illuminate the timing and mechanisms underlying this regionalization.

**Additional comments:**

**4. Is there any evidence from this dataset on the presence or absence of a caudal ganglionic eminence in lamprey?**

Based on the gene expression profiles of the predicted cell types, we could not obtain any evidence for the presence of
 a CGE in the lamprey brain.

**5. The use of the term “lamprey astrocytes” is misleading. The cells described in the paper are ependymoglia cells,
 similar to the ependymoglia cells of other non-mammalian vertebrates. They are quite different from mammalian
 astrocytes, although evolutionarily related in a complex way.**

Thank for raising this point, which helped us to clarify the relevant terminology and also better highlight our novel
 insights regarding ependymoglia cells in lamprey. Indeed, our analyses for the first time demonstrate that there are
 two large families of lamprey ependymoglia cells (section: “Ependymoglia cells and the origin of myelination”, P. 6-7,
 L213-255; Fig. 2, Extended Data Fig. 6): one that is likely homologous to mammalian astrocytes and the other that is
 homologous to mammalian ependymal cells, a conclusion that is based on unbiased and comprehensive transcriptome-
 based comparisons of cell type gene expression profiles between the mouse and lamprey atlases (Fig. 2). Notably,
 lamprey astrocytes are highly comparable to those from mouse in terms of their overall transcriptome signature (Fig.
 2c, e) (P. 7, L231-236). They share key marker genes that are fundamental for the development and function of
 astrocytes, such as *SOXE3* (*Sox9*), *HES5*, and *SLC1A2* (Fig. 3a; Extended Data Fig. 6a). However, like in other anamniotes

(e.g., fishes, amphibians), lamprey astrocytes are mainly localized around the ventricles (Fig.3; Extended Data Fig. 6h),
forming the so-called ependymo-radial glia.

In agreement with the referee's comment and our observations, we now define explicitly the astrocyte-like cells
observed in lamprey as "astrocytes" – a term then used in the rest of the manuscript – in the beginning of the section
"Ependymogial cells and the origin of myelination":

(P15, L214-216) "Our analyses revealed that ependymogial cells (i.e., CNS glia) in lamprey are divided into two main,
developmentally related, cell classes: ependymal cells and astrocyte-like cells, referred to as "ependymal" and
"astrocytes" hereafter, given the observations described below."

We now also extended and refined the corresponding section in the Discussion, to better highlight these findings:

(P14, L432-446) "Our analyses of non-neuronal cells revealed the presence of two distinct cell types within the lamprey
ependymoglia that are likely homologous to ependymal cells and astrocytes of gnathostomes (Fig.1c), suggesting that
two of the three main macroglial cell types (astrocytes, ependyma, oligodendrocytes) were already established in the
common vertebrate ancestor. Notably, however, our work confirms the absence of oligodendrocytes and sheds new
light on their origination. We found that lamprey astrocytes express several oligodendrocyte-specific genes, including
master regulators and effector genes (Fig. 3b). Our observations suggest that key components of the molecular
machinery of oligodendrocytes were present in astrocyte-like cells of the vertebrate ancestor, and indicate that
oligodendrocytes originated from these evolutionary precursors on the gnathostome lineage (Fig. 6a). Our work thus
extends previous studies, which showed that lamprey axons seem to be physically associated with astrocytes²⁷ and that
key aspects of the regulatory program required for oligodendrocyte differentiation in gnathostomes are present during
lamprey gliogenesis³⁴. While our data also confirm the absence of actual myelin from the lamprey PNS³, our study also
lends strong support to the previous hypothesis⁴ that lamprey PNS glia are homologous to both mammalian Schwann
and satellite cells."

6. Is there a GABAergic prethalamus?

We have localized the expression of GABAergic markers (e.g., *DLX1*) in the pre-thalamus (see Fig. 1e and Extended Data
Fig. 7l, 8d), suggesting that there is indeed a GABAergic prethalamus in lamprey, as in jawed vertebrates.

655 7. I was surprised to see foxg1 only in telencephalic GABAergic neurons. Can technical artifacts (e.g. gene annotation, 656 isoforms) be excluded?

*FOXP1* is expressed also in telencephalic excitatory neurons, but in a lower fraction of cells (see online atlas; please also
see the screenshot below, Reviewer Figure 1). We also checked the bam files to look for potential artifacts (e.g., reads
mapping to unannotated 3'UTRs), but we found none.

**Reviewer Figure 1.** Expression of *FOXP1* in telencephalic cell types.

**8.** The website <https://lampreybrain.kaessmannlab.org/adult.html> is great, but having to switch between gene ids
(*MSTRG.10634*) and gene names (*GAD1*) makes data exploration cumbersome.

Thank you for the appreciation of our database. We agree that the need to switch between gene IDs and names was
not ideal and have therefore modified the “Genes” tab, so that it now provides both gene IDs and gene names integrated
into the same table/figure. Additionally, we note that genes can be searched using either their ID (e.g., *MSTRG.10634*)
or name (e.g., *GAD1*).

**Minor suggestion:**

**9.** In Extended Figure 8, it would be helpful to separate larval and adult panels in different rows.

This is a great suggestion that we have implemented in the new **Extended Data Fig. 8**.

**Referee #3 (Remarks to the Author):**

**This is an exceptionally well-written report of a series of scRNAseq done in larval and adult sea lamprey brain and of**
**the separate 4 brain areas: telencephalon, diencephalon, mesencephalon and rhombencephalon. The quality of**
**analysis of these scRNAseq datasets is outstanding. The conclusions are of great interest for brain evolution and the**
**comparisons with published mouse scRNAseq datasets bring new insights in the field. They find conserved cell type**
**'families' building for the first time a framework for the ancestral brain cell type repertoire. They also identify**
**modules of evolution and confirm the absence of oligodendrocytes and thereby absence of myelination. The authors**
**found part of the oligodendrocyte transcriptome signature in astrocytes, supporting the current view of them**
**stemming from astrocytes. The clustering also confirms the absence of a proper cerebellum in the hindbrain and the**

presence of MGE and LGE in the subpallium of the telencephalon. In the
 telencephalon, the analysis deepens in generating findings on the type of projection modalities of the pallial
 excitatory neurons.
 Overall, this is an outstanding study of single cell transcriptomics accompanied by the delivery of a very useful web
 tool for the community. Specific points of improvement are listed below.

 We are very thankful to the referee for being so enthusiastic about our work.

 **Specific comments:**

 **[Lines 90-97] Authors should perform statistical analyses between samples, and comment whether different levels**
 **of expression was observed between the two chemistries at Larval stage.**

 Motivated by the referee's comment and a related comment by referee #1, we have reinvestigated differences between
 v2 and v3 larval data using newly generated datasets and compared the larval and adult data in more detail, adding
 statistical analyses as suggested by the referee.

 Specifically, we generated scRNA-seq data for whole brain samples from three additional ammocoetes (SN580, SN582,
 SN588; **Supplementary Table 1 and 2**; total of 22,950 cells). In parallel, we also found out (based on assessments of data
 also from other lab projects from that time) that during the transition phase from v2 to v3 kits, we had received a few
 suboptimal v3 kit batches, one of which was used to generate the data for the previous SN352 sample, which we hence
 discarded from the analyses.

 Analyses of the new v3 data show that while the number of expressed genes is now similar between adult and larval
 cells (**Extended Data Fig. 4b**), gene expression patterns remain overall significantly more cell type-specific in adults
 compared to ammocoetes ($P < 0.001$ Mann-Whitney U test; **Extended Data Fig. 4c**). This observation was confirmed in
 analyses in which we downsampled all datasets to both the same number of cells and scRNA-seq reads, which rules out
 that the result is due to technical differences (i.e., amounts of data) between samples (**Extended Data Fig. 4d**).

 However, upon our reanalysis, we realized that our data do not directly imply that a larger proportion of neurons (i.e.,
 neurons) in larvae are undifferentiated. We therefore removed this notion from the discussion section and
 condensed/toned down the paragraph describing these observations in the beginning of the paper, which now reads:

 **(P3, L90-93) "Overall, neural cell type compositions are similar between the two stages (Extended Data Figs. 2-4a).**
 **However, we noted a generally higher cell type specificity of gene expression patterns in adults compared to**
 **ammocoetes (Extended Data Fig. 4c) – a result that is robust to controls for technical differences between datasets**
 **(Extended Data Fig. 4b-d; Methods)."**

 In any event, the neural cell type atlases are generally very similar for adult lampreys and larvae and the conclusions
 made in our paper apply to both life stages. We just focus the manuscript more on the adult stage, given the more
 distinct cell type/gene expression data, the overall higher resolution of the adult scRNA-seq (all v3) and ISS data (e.g.,
 larger number of marker genes), and the fact that adults may represent the life stage that is more relevant for
 evolutionary inferences of ancestral traits (Miyashita et al. *Nature* 2021; PMID: 33692547).

 **[Lines 670-674] Clearer description of how the Specificity Indexes was quantified will benefit readers. For instance,**
 **did the authors use raw counts or normalized counts for this quantification, and was this value normalised to the**
 **number of cell types from each species?**

 The method to calculate Specificity Indexes was developed by Tosches and colleagues (Tosches et al. *Science* 2018;
 PMID: 29724907) and is described in detail in their paper, which we cite in this Methods section. However, we agree
 with the referee that a short additional description about the essence of the method is useful and so we now provide a
 brief explanation in the methods when describing the v2 and v3 datasets comparisons, where these indexes were also
 used (see also reply to previous comment; **Extended Data Fig. 4c, d**).

 **(P20, L703-709) "We also compared the distributions of cell type-specific gene expression signals across datasets based**
 **on gene specificity indexes (SI) calculated using the method developed by Tosches and colleagues⁵¹. Briefly, to obtain**

the SI with this method, the mean of normalized scRNA-seq read counts of each gene (g_c) is calculated for each cell type
(C) and then divided by its mean across all cells:

$$s_{g,c} = \frac{g_c}{\frac{1}{N} \sum_{i \in c} g_i}$$

”

The relevant part regarding the lamprey-mouse comparisons now reads:

(P21, L719-723) “The new “meta-gene” count matrices were then normalized using SCTransform, filtered for HVGs, and
averaged across all annotated clusters. Expression levels were finally transformed to SIs (see above), which were then
used for Pearson correlation analyses. Dendrograms relating cell-type families between lamprey and mouse were
constructed using the pvclust¹⁰¹ R package with complete hierarchical clustering and 1,000 replicates.”

**The beautiful and complex set of findings made and highlighted in the Discussion section would deserve a Figure**
**summarising the key findings.**

We thank the referee for the very useful suggestion. Accordingly, we now provide a new figure (Fig. 6a-d) that
summarizes our key observations and inferences and is cited throughout the Discussion section.

Final Decision Letter:

18th July 2023

Dear Henrik,

We are pleased to inform you that your Article entitled "A lamprey neural cell type atlas illuminates the origins of the vertebrate brain", has now been accepted for publication in Nature Ecology & Evolution.

Over the next few weeks, your paper will be copyedited to ensure that it conforms to Nature Ecology and Evolution style. Once your paper is typeset, you will receive an email with a link to choose the appropriate publishing options for your paper and our Author Services team will be in touch regarding any additional information that may be required

Due to the importance of these deadlines, we ask you please us know now whether you will be difficult to contact over the next month. If this is the case, we ask you provide us with the contact information (email, phone and fax) of someone who will be able to check the proofs on your behalf, and who will be available to address any last-minute problems . Once your paper has been scheduled for online publication, the Nature press office will be in touch to confirm the details.

Acceptance of your manuscript is conditional on all authors' agreement with our publication policies (see www.nature.com/authors/policies/index.html). In particular your manuscript must not be published elsewhere and there must be no announcement of the work to any media outlet until the publication date (the day on which it is uploaded onto our web site).

Please note that *Nature Ecology & Evolution* is a Transformative Journal (TJ). Authors may publish their research with us through the traditional subscription access route or make their paper immediately open access through payment of an article-processing charge (APC). Authors will not be required to make a final decision about access to their article until it has been accepted. [Find out more about Transformative Journals](https://www.springernature.com/gp/open-research/transformative-journals)

Authors may need to take specific actions to achieve [compliance](https://www.springernature.com/gp/open-research/funding/policy-compliance-faqs) with funder and institutional open access mandates. If your research is supported by a funder that requires immediate open access (e.g. according to [Plan S principles](https://www.springernature.com/gp/open-research/plan-s-compliance))

31then you should select the gold OA route, and we will direct you to the compliant route where possible. For authors selecting the subscription publication route, the journal's standard licensing terms will need to be accepted, including <https://www.nature.com/nature-portfolio/editorial-policies/self-archiving-and-license-to-publish>. Those licensing terms will supersede any other terms that the author or any third party may assert apply to any version of the manuscript.

We welcome the submission of potential cover material (including a short caption of around 40 words) related to your manuscript; suggestions should be sent to Nature Ecology & Evolution as electronic files (the image should be 300 dpi at 210 x 297 mm in either TIFF or JPEG format). Please note that such pictures should be selected more for their aesthetic appeal than for their scientific content, and that colour images work better than black and white or grayscale images. Please do not try to design a cover with the Nature Ecology & Evolution logo etc., and please do not submit composites of images related to your work. I am sure you will understand that we cannot make any promise as to whether any of your suggestions might be selected for the cover of the journal.

You can generate the link yourself when you receive your article DOI by entering it here: <http://authors.springernature.com/share>.

[REDACTED]

P.S. Click on the following link if you would like to recommend Nature Ecology & Evolution to your librarian <http://www.nature.com/subscriptions/recommend.html#forms>

32** Visit the Springer Nature Editorial and Publishing website at http://editorial-jobs.springernature.com?utm_source=ejp_NEcoE_email&utm_medium=ejp_NEcoE_email&utm_campaign=ejp_NEcoE for more information about our career opportunities. If you have any questions please click [here](mailto:editorial.publishing.jobs@springernature.com).**